# Structural Causal Bandits under Markov Equivalence

**Min Woo Park**[1]     **Andy Arditi**[2]     **Elias Bareinboim**[2*]     **Sanghack Lee**[1*]

[1]Seoul National University     [2]Columbia University
alsdn0110@snu.ac.kr   ava2123@columbia.edu
eb@cs.columbia.edu   sanghack@snu.ac.kr

## Abstract

In decision-making processes, an intelligent agent with causal knowledge can optimize action spaces to avoid unnecessary exploration. A *structural causal bandit* framework provides guidance on how to prune actions that are unable to maximize reward by leveraging prior knowledge of the underlying causal structure among actions. A key assumption of this framework is that the agent has access to a fully-specified causal diagram representing the target system. In this paper, we extend the structural causal bandits to scenarios where the agent leverages a Markov equivalence class. In such cases, the causal structure is provided to the agent in the form of a *partial ancestral graph* (PAG). We propose a generalized framework for identifying potentially optimal actions within this graph structure, thereby broadening the applicability of structural causal bandits.

## 1 Introduction

The multi-armed bandit (MAB) [Robbins, 1952, Lai and Robbins, 1985, Lattimore and Szepesvári, 2020] problem is a central topic in decision-making studies, where an agent aims to maximize cumulative rewards by repeatedly choosing actions based on observed reward, balancing the exploration-exploitation trade-off. Traditionally, MAB problems assume independence among the rewards of different arms, meaning that the reward obtained from one arm provides no information about the others, e.g., KL-UCB [Cappé et al., 2013] and Thompson sampling [Thompson, 1933]. Although this independence assumption simplifies the problem, it limits its applicability to real-world scenarios where dependencies among actions are common, e.g., in a movie recommendation system, a user's positive reaction to one genre may indicate a higher likelihood of a positive reaction to similar genres.

Recent research has increasingly recognized the importance of structured dependencies among arms and reward [Li et al., 2010, Abbasi-Yadkori et al., 2011, Cesa-Bianchi and Lugosi, 2012], leading to the development of structured bandits. Concurrently, the integration of causal inference into the MAB framework has opened new avenues for modeling and solving decision problems with richer dependency structures [Bareinboim et al., 2024]. Causal diagrams [Pearl, 1995] have been employed to represent causal relationships among actions, rewards, and other relevant factors. This approach enables agents to make informed decisions by considering how each action causally influences the reward through causal pathways. Existing studies [Bareinboim et al., 2015, Lattimore et al., 2016, Forney et al., 2017] have shown that causality-aware strategies can significantly outperform MAB algorithms that do not account for such underlying causal relationships. Subsequent work has explored various specialized settings by introducing additional structural assumptions, such as the availability of both observational and experimental distributions, or linear mechanisms [Zhang and Bareinboim, 2017, Lu et al., 2020, Bilodeau et al., 2022, Feng and Chen, 2023, Varici et al., 2023].

---

[*]Corresponding authors

Specifically, Lee and Bareinboim [2018] formalized the *structural causal bandit* (SCM-MAB) without any parametric assumptions, where causal dependencies between arms are modeled using a structural causal model (SCM) [Pearl, 2000]. They proposed a sound and complete graphical characterization to identify *minimal intervention sets* (MISs) and *possibly-optimal minimal intervention sets* (POMISs), where the former includes only the variables that affect the reward, and the latter refers to actions that could be part of an optimal strategy among MISs, thereby guiding the agent to avoid unnecessary exploration without any actual interaction. Lee and Bareinboim [2019] extended this approach to accommodate scenarios involving non-manipulable variables among all the variables in the graph. Lee and Bareinboim [2020] and Everitt et al. [2021] established SCM-MAB with stochastic policies and Carey et al. [2024] studied the completeness of its graphical characterization.

While SCM-MAB has been established as a general framework, these studies assume that the decision-making agent has perfect access to the entire causal structure. From observational data, only a Markov equivalence class of the true causal diagram over observed variables can be inferred without making a substantial assumption about causal mechanisms such as causal sufficiency [Verma and Pearl, 1990, Spirtes et al., 2001b, Chickering, 2002, Tsamardinos et al., 2006] or a functional assumption [Perry et al., 2022, Peters et al., 2016, Ghassami et al., 2017, Heinze-Deml et al., 2018, Huang et al., 2020, Ghassami et al., 2018, Zeng et al., 2021]. A prominent representation of the equivalence class is known as partial ancestral graphs (PAGs), and any causal diagrams can be uniquely represented by a PAG [Richardson and Spirtes, 2002, Zhang, 2006, 2008a,b, Ali, 2005].

**Motivation and Contributions.**    With observational data, we can only learn a PAG, which encodes a super-exponential number of maximal ancestral graphs (MAGs), each of which, in turn, represents an *infinite* number of causal diagrams over supersets of the observed variables. Therefore, considering all causal diagrams consistent with the PAG is computationally prohibitive. Identifying conditions for MIS and POMIS at the level of ancestral graphs *directly* would allow one to circumvent the issue. Our key contributions are as follows:

- We generalize MIS and develop its graphical criteria in ancestral graphs, enabling an agent to identify and exclude variables that have no effect on the reward (Sec. 3).
- We devise POMIS for ancestral graphs along with its graphical characterization, leading to an action space worth exploring (Secs. 4.1 and 4.2).
- We present an efficient algorithm to determine whether a given intervention set can be a POMIS in the Markov equivalence class represented by a PAG (Sec. 4.3).

Experiments in Sec. 5 and additional ones in Appendix D corroborate our findings. All omitted proofs are provided in Appendix H along with auxiliary results in Appendix G.

## 2    Preliminaries

We introduce notation and review relevant prior work. Following conventions, we use a capital letter, such as $X$, to represent a variable, with its corresponding lowercase letter, $x$, denoting a realization of the variable. Boldface is employed to represent a set of variables or values, denoted by $\mathbf{X}$ or $\mathbf{x}$. The domain of $X$ is indicated by $\mathfrak{X}_X$. We use calligraphic letters for graphs and models such as $\mathcal{G}$ and $\mathcal{S}$.

**Graphical notations.**    We consider a graph $\mathcal{G}$ having vertices $\mathbf{V}$ and edges $\mathbf{E}$ composed of directed ($\rightarrow$) and bidirected edges ($\leftrightarrow$). If there is an edge between two vertices $X$ and $Y$ in $\mathcal{G}$, we say that the two vertices are *adjacent* in $\mathcal{G}$, denoted by $Y \in \mathtt{Adj}(X)_{\mathcal{G}}$ or $X \in \mathtt{Adj}(Y)_{\mathcal{G}}$. An ordered sequence of distinct nodes in $\mathcal{G}$ is called a *path* between $X$ and $Y$ in $\mathcal{G}$ if (1) the start node is $X$ and the end node is $Y$, and (2) there is an edge between any two subsequent variables in the sequence. If a path consists of directed edges with the same orientation, we say the path is *directed*. A variable $Z$ is called a *collider* on the path if the path contains two edges having arrowheads toward $Z$. We define a path as a *collider path* if all non-endpoint vertices along the path are colliders. A path is *uncovered (unshielded)* if, for every consecutive triple on the path, its endpoints are not adjacent.

A path is *possibly directed* from $X$ to $Y$ if there is no arrowhead on the path pointing towards $X$. If there is a (possibly) directed path from $X$ to $Y$, then $Y$ is called a *(possible) descendant* of $X$, and $X$ is a *(possible) ancestor* of $Y$. A variable $Y$ is referred to as a *possible child* of $X$, and $X$ is a *possible parent* of $Y$ if they are adjacent and the edge is not directed into $X$. We denote the

ancestors, descendants, parents, and children of a given variable as `An`, `De`, `Pa`, and `Ch`, respectively. Ancestors and descendants include the variable itself. For a set of variables, we define the ancestral set as $\text{An}(\mathbf{X})_{\mathcal{G}} = \bigcup_{X \in \mathbf{X}} \text{An}(X)_{\mathcal{G}}$, and similarly for other relationships. We add the prefix `Poss` when referring to possible relationships, such as `PossAn`.

An *inducing path* relative to $\mathbf{L}$ is defined as a path where every vertex not in $\mathbf{L}$ is a collider on the path, and every collider is an ancestor of an endpoint of the path. A directed edge $X \to Y$ is *visible* if there exists no causal diagram in the corresponding equivalence class where there is an inducing path between $X$ and $Y$ that is into $X$. We refer to any edge that is not visible as *invisible*. The $\mathbf{X}$-lower-manipulation of $\mathcal{G}$ deletes all edges that are visible and are out of variables in $\mathbf{X}$, and replaces all those edges that are out of variables in $\mathbf{X}$ but are invisible in $\mathcal{G}$ with bidirected edges denoted as $\mathcal{G}_{\underline{\mathbf{X}}}$. The $\mathbf{X}$-upper-manipulation of $\mathcal{G}$ deletes all those edges in $\mathcal{G}$ that are into variables in $\mathbf{X}$ denoted as $\mathcal{G}_{\overline{\mathbf{X}}}$. We denote the set of variables in $\mathcal{G}$ by $\mathbf{V}(\mathcal{G})$. A subgraph $\mathcal{G}[\mathbf{V}']$, where $\mathbf{V}' \subseteq \mathbf{V}(\mathcal{G})$ is defined as a vertex-induced subgraph in which all edges among the vertices in $\mathbf{V}'$ preserved. We define $\mathcal{G} \backslash \mathbf{X}$ as $\mathcal{G}[\mathbf{V}(\mathcal{G}) \setminus \mathbf{X}]$ for $\mathbf{X} \subseteq \mathbf{V}(\mathcal{G})$.

**Structural Causal Model.** We use structural causal model (SCM) [Pearl, 2000] as the semantical framework to represent the underlying environment a decision-maker is deployed. An SCM $\mathcal{S}$ is a quadruple $\langle \mathbf{U}, \mathbf{V}, \mathbf{F}, P(\mathbf{U}) \rangle$, where $\mathbf{U}$ is a set of exogenous variables determined by factors outside the model following a joint distribution $P(\mathbf{U})$, and $\mathbf{V}$ is a set of endogenous variables whose values are determined following a collection of functions $\mathbf{F} = \{f_i\}_{V_i \in \mathbf{V}}$ such that $V_i \leftarrow f_i(\mathbf{pa}_i, \mathbf{u}_i)$ where $\mathbf{PA}_i \subseteq \mathbf{V} \setminus \{V_i\}$ and $\mathbf{U}_i \subseteq \mathbf{U}$. The observational probability $P(\mathbf{v})$ is defined as $\sum_{\mathbf{u}} \prod_{V_i \in \mathbf{V}} P(v_i \mid \mathbf{pa}_i, \mathbf{u}_i) P(\mathbf{u})$. Every SCM $\mathcal{S}$ is associated with a *causal diagram* $\mathcal{G} = \langle \mathbf{V}, \mathbf{E} \rangle$ where a directed edge $V_i \to V_j \in \mathbf{E}$ if $V_i \in \mathbf{PA}_j$, and a bidirected edge between $V_i$ and $V_j$ if $\mathbf{U}_i$ and $\mathbf{U}_j$ are correlated. The probability of $\mathbf{V} = \mathbf{v}$ when $\mathbf{X}$ is intervened upon to take the value $\mathbf{x}$ is denoted by $P(\mathbf{v} \setminus \mathbf{x} \mid do(\mathbf{x}))$.

**Ancestral graphical structures.** Ancestral graphs are designed to capture graph structures without explicitly modeling latent variables. While directed edges between vertices in a causal diagram imply a direct causal effect between them, in ancestral graphs, directed edges instead represent ancestral relationships. Similar to the absence of directed cycles in causal diagrams, ancestral graphs do not permit *almost directed cycle*, which occurs when $X \leftrightarrow Y$ is present while $X$ is an ancestor of $Y$.

A mixed graph is called a *maximal ancestral graph* (MAG) if (i) it does not contain any directed or almost directed cycles (i.e., ancestral); and (ii) there is no inducing path between any two non-adjacent vertices (i.e., maximal). In general, a MAG represents a set of causal diagrams with the same set of observed variables

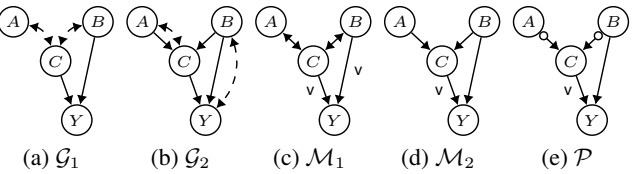

(a) $\mathcal{G}_1$    (b) $\mathcal{G}_2$    (c) $\mathcal{M}_1$    (d) $\mathcal{M}_2$    (e) $\mathcal{P}$

Figure 1: Causal diagrams (a, b) with corresponding (c, d) MAGs and (e) PAG. A visible edge is marked with v.

that entail the same conditional independence and ancestral relations among the observed variables. For each causal diagram, there exists a unique MAG over observed variables which represents its marginal independence relations, as well as its ancestral relations. However, a MAG is not fully testable with observational data since distinct MAGs can encode the same marginal independence relations. To illustrate, consider the causal diagrams $\mathcal{G}_1$ and $\mathcal{G}_2$ in Fig. 1. While they yield the same conditional independence relations, they correspond to distinct MAGs, $\mathcal{M}_1$ and $\mathcal{M}_2$, respectively.

A graph is a partial mixed graph (PMG) if it contains three types of marks: tails ($-$), arrowheads ($>$), and circles ($\circ$). A circle mark implies an uncertain mark that can be either an arrowhead or a tail. In addition, we use an asterisk ($*$) as a wildcard to denote any possible mark. In a PMG, if every edge mark on a path consists of circles, the path is called a *circle path*, and each edge is called a *circle edge* ($\circ\!-\!\circ$). An edge is a *partially directed edge* ($\circ\!\to$) if it has both circle and arrowhead. A *circle component* is a subgraph of a PMG in which every edge is a circle edge. We use the ? mark to emphasize a wildcard that represents either a tail ($-$) or a circle ($\circ$), but not an arrowhead ($>$). Furthermore, $[\mathcal{Q}]$ denotes the set of MAGs represented by the PMG $\mathcal{Q}$, and similarly $[\mathcal{M}]$ denotes the set of causal diagrams conforming to the MAG $\mathcal{M}$.

A *partial ancestral graph* (PAG) denoted by $\mathcal{P}$, is a PMG such that it represents a Markov equivalence class of MAGs. Every MAG $\mathcal{M}$ represented by a PAG has the same skeleton as $\mathcal{P}$, and the non-circle marks in $\mathcal{P}$ are identical to those in $\mathcal{M}$. Every circle in $\mathcal{P}$ corresponds to a variant mark among the

represented MAGs. The PAG $\mathcal{P}$ in Fig. 1e, for instance, encodes every MAG obtained by orienting circle marks incident to $A$ and $B$ as either $>$ or $-$, including both $\mathcal{M}_1$ and $\mathcal{M}_2$. In our work, we assume the absence of selection bias; therefore, there is no undirected edge in PAGs and MAGs. Moreover, we assume access to the *true* PAG that represents the target underlying system. We refer readers unfamiliar with ancestral graphs to Zhang [2006] and Jaber [2022].

**Structural causal bandits.** We follow the *structural causal bandit* (SCM-MAB) problem [Lee and Bareinboim, 2018], where an SCM models the target system with which an agent interacts, including a reward variable $Y \in \mathbf{V}$ where $\mathfrak{X}_Y \subseteq \mathbb{R}$. In the SCM-MAB setting, pulling each arm corresponds to intervening on a set of variables $\{\mathbf{x} \in \mathfrak{X}_\mathbf{X} \mid \mathbf{X} \subseteq \mathbf{V} \setminus \{Y\}\}$. The mean reward of an arm is denoted by $\mu_\mathbf{x} = \mathbb{E}[Y \mid do(\mathbf{x})]$ and the best expected reward by intervening on $\mathbf{X}$ is $\mu_{\mathbf{x}^*} = \max_{\mathbf{x} \in \mathfrak{X}_\mathbf{X}} \mu_\mathbf{x}$. We denote $\mu^*$ as the optimal expected reward. The goal of the agent is to minimize the cumulative regret after $N$ rounds, which is given by $\text{Reg}_N = \sum_{\mathbf{x} \in \mathfrak{X}_\mathbf{X}, \mathbf{X} \subseteq \mathbf{V} \setminus \{Y\}} \Delta_\mathbf{x} \mathbb{E}[T_\mathbf{x}(N)]$ where $T_\mathbf{x}(N)$ denotes the number of times the arm $\mathbf{x}$ was played after $N$ rounds, $\Delta_\mathbf{x} = \mu^* - \mu_\mathbf{x}$ and $\mathfrak{X}_\mathbf{X} = \times_{X \in \mathbf{X}} \mathfrak{X}_X$.

**MIS and POMIS.** A minimal intervention set (MIS) ensures that there is no proper subset that is equivalent to the set with respect to the reward. Lee and Bareinboim [2018] demonstrated that $\mathbf{X}$ is an MIS if and only if $\mathbf{X} \subseteq \text{An}(Y)_{\mathcal{G}_{\overline{\mathbf{X}}}}$. For instance, consider $\mathcal{G}$ in Fig. 2 where $\{A, B\}$ is an MIS since $\{A, B\} \subseteq \text{An}(Y)_{\mathcal{G}_{\overline{\{A,B\}}}}$ holds. In contrast, $\{A, B, C\}$ is *not* an MIS since $A$ is not an ancestor of $Y$ in $\mathcal{G}_{\overline{\{A,B,C\}}}$, as depicted in Fig. 2c.

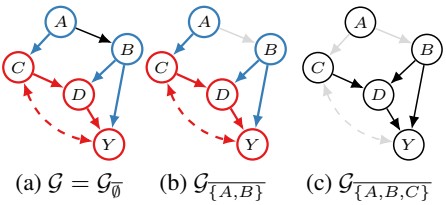

(a) $\mathcal{G} = \mathcal{G}_{\overline{\emptyset}}$    (b) $\mathcal{G}_{\overline{\{A,B\}}}$    (c) $\mathcal{G}_{\overline{\{A,B,C\}}}$

Figure 2: MUCT (red) and IB (blue).

A possibly-optimal minimal intervention set (POMIS) is an MIS such that intervening on any non-POMISs cannot yield a better outcome than the optimal one associated with the POMIS. Therefore, an agent who is aware of POMISs should only explore and exploit actions consistent with those sets. When given a causal diagram $\mathcal{G}$, minimal unobserved confounders' territory (MUCT) and interventional border (IB) [Lee and Bareinboim, 2018] provide a graphical characterization of POMIS. MUCT is the minimal set of variables that (i) contains the reward variable $Y$; and (ii) is closed under descendants and bidirected edge connections; and IB consists of the parents of MUCT, excluding MUCT itself. We defer the formal definitions to Appendix B. Intuitively, MUCT is the minimal closed mechanism that conveys all hidden information from unobserved confounders to the downstream reward, while IB consists of the nodes that directly affect this closed mechanism. Let us denote MUCT and IB with respect to $[\![\mathcal{G}, Y]\!]$ as $\text{MUCT}(\mathcal{G}, Y)$ and $\text{IB}(\mathcal{G}, Y)$, respectively. Leveraging these, the authors showed that $\text{IB}(\mathcal{G}_{\overline{\mathbf{X}}}, Y) = \mathbf{X}$ provides a complete characterization of POMISs. For example, Figs. 2a and 2b show MUCT and IB for the subgraphs $\mathcal{G}_\emptyset$ and $\mathcal{G}_{\overline{\{A,B\}}}$. The do-nothing action ($do(\emptyset)$) is not a POMIS since $\text{MUCT}(\mathcal{G}_{\overline{\emptyset}}, Y) = \{C, D, Y\}$ implies $\text{IB}(\mathcal{G}_{\overline{\emptyset}}, Y) = \{A, B\}$, not $\emptyset$, while the set $\{A, B\}$ is a POMIS since $\text{MUCT}(\mathcal{G}_{\overline{\{A,B\}}}, Y) = \{C, D, Y\}$, implying $\text{IB}(\mathcal{G}_{\overline{\{A,B\}}}, Y) = \{A, B\}$.

## 3 Generalizing Minimal Intervention Sets

We first generalize minimal intervention set (MIS) to cover not only causal diagrams but also ancestral graphs, aiming to identify all sets that do not include variables irrelevant to the reward by ruling them out, referring to MAGs or PAGs. In the following parts, we first provide complete graphical conditions for MIS in terms of MAGs and PAGs. Surprisingly, we then show in Sec. 3.1 that an MIS may include variables irrelevant to reward when dealing with PAGs. To address this issue, in Sec. 3.2, we propose the concept of *definitely* minimal intervention set (DMIS), which ensures that no further variables can be pruned, thereby aligning with the intuitive notion of minimality.

We use $\mathcal{D}$ to refer to either a causal diagram or an ancestral graph (MAG or PAG) over $\mathbf{V}$. We denote by $\mathbf{x}[\mathbf{W}]$ the values of $\mathbf{x}$ restricted to the subset of variables in $\mathbf{W} \cap \mathbf{X}$.

**Definition 1** (Minimal intervention set (MIS)). *Given information $[\![\mathcal{D}, Y]\!]$, a set of variables $\mathbf{X} \subseteq \mathbf{V} \setminus \{Y\}$ is called a minimal intervention set (MIS) relative to $[\![\mathcal{D}, Y]\!]$ if there is no $\mathbf{X}' \subsetneq \mathbf{X}$ such that $\mu_{\mathbf{x}[\mathbf{X}']} = \mu_\mathbf{x}$ for every SCM conforming to $\mathcal{D}$.*

**Proposition 1.** *Let $\mathcal{M}$ be a MAG over $\mathbf{V}$. A set $\mathbf{X} \subseteq \mathbf{V} \setminus \{Y\}$ is an MIS relative to $[\![\mathcal{M}, Y]\!]$ if and only if there exists a causal diagram $\mathcal{G}$ conforming to $\mathcal{M}$ such that $\mathbf{X}$ is an MIS relative to $[\![\mathcal{G}, Y]\!]$.*

The proposition guarantees the existence of a causal diagram $\mathcal{G}$ where $\mathbf{X}$ is an MIS relative to $[\![\mathcal{G}, Y]\!]$, provided that $\mathbf{X}$ is an MIS relative to $[\![\mathcal{M}, Y]\!]$ for the given MAG $\mathcal{M}$. We now proceed to the graphical characterization of MIS for MAGs, in a manner similar to causal diagrams, utilizing the explicit ancestral relations among variables in MAGs and Rule 3 of do-calculus for MAGs, i.e., $\mu_{\mathbf{xz}} = \mu_{\mathbf{x}}$ if $\mathbf{Z}$ and $Y$ are m-separated in $\mathcal{M}_{\overline{\mathbf{X}}}$ [Zhang, 2008b].

**Theorem 1** (Characterization of MIS for MAGs). *Let $\mathcal{M}$ be a MAG over $\mathbf{V}$. Given information $[\![\mathcal{M}, Y]\!]$, a set $\mathbf{X} \subseteq \mathbf{V} \setminus \{Y\}$ is an MIS relative to $[\![\mathcal{M}, Y]\!]$ if and only if $\mathbf{X} \subseteq \mathrm{An}(Y)_{\mathcal{M}_{\overline{\mathbf{X}}}}$ holds.*

For example, consider $\mathcal{G}'$ and $\mathcal{M}$ in Figs. 3a and 3b where $\mathcal{G}' \in [\mathcal{M}]$. A set $\{A, B, C\}$ is an MIS relative to $[\![\mathcal{M}, Y]\!]$ since $\{A, B, C\} \subseteq \mathrm{An}(Y)_{\mathcal{M}_{\overline{\{A,B,C\}}}}$ holds.

**Remark 1.** *Even though a set $\mathbf{X}$ is an MIS with respect to $[\![\mathcal{M}, Y]\!]$, there is no guarantee that $\mathbf{X}$ is an MIS with respect to $[\![\mathcal{G}, Y]\!]$ for every causal diagram $\mathcal{G}$ conforming to $\mathcal{M}$.*

The set $\{A, B, C\}$ is also an MIS relative to $[\![\mathcal{G}', Y]\!]$ in Fig. 3a since $\{A, B, C\} \subseteq \mathrm{An}(Y)_{\mathcal{G}'_{\overline{\{A,B,C\}}}}$ holds. However, while $\mathcal{G}$ in Fig. 2a is also represented by $\mathcal{M}$, it is *not* an MIS with respect to $[\![\mathcal{G}, Y]\!]$.[2]

### 3.1 MIS for PAGs and Its Possible Vacuousness

We proceed to the characterization of MIS for PAGs. Unfortunately, we cannot similarly rely on Rule 3 for PAGs (Jaber et al. [2022]; see Thm. 6 in Appendix B) because the rule is applied when ancestral relations are apparent for *all* represented models, whereas a PAG might involve uncertainty reflected by circle marks—one may easily surmise that $\{X, Z\}$ is not an MIS in a PAG $X \circ\!\!-\!\!\circ Z \circ\!\!-\!\!\circ Y$, but Rule 3 remains silent on this case.

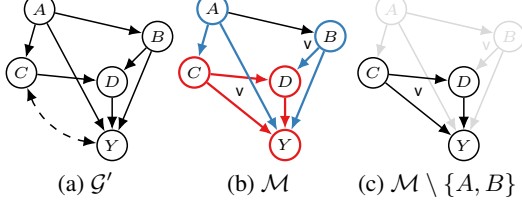

(a) $\mathcal{G}'$      (b) $\mathcal{M}$      (c) $\mathcal{M} \setminus \{A, B\}$

Figure 3: (a) Causal diagram; and (b) MAG representing both causal diagrams $\mathcal{G}'$ and $\mathcal{G}$ in Fig. 2a. (c) Induced graph of $\mathcal{M}$ over $\mathbf{V} \setminus \{A, B\}$.

Hence, we utilize a specific type of path: A *proper possibly-directed path* from $X \in \mathbf{X}$ to $Y$ with respect to $\mathbf{X}$, where only the first node $X$ is in $\mathbf{X}$. This path is not disturbed by other intervening variables, thus aligning with the characterizations of MISs for causal diagrams and MAGs.

**Proposition 2** (Graphical characterization of MIS for PAGs). *Let $\mathcal{P}$ be a PAG over the set of variables $\mathbf{V}$. A set $\mathbf{X} \subseteq \mathbf{V} \setminus \{Y\}$ is an MIS relative to $[\![\mathcal{P}, Y]\!]$ if and only if, for every variable $X \in \mathbf{X}$, there exists a proper possibly-directed path from $X$ to $Y$ with respect to $\mathbf{X}$ in $\mathcal{P}$.*

**Possible vacuousness.** One might expect that if $\mathbf{X}$ is an MIS relative to $[\![\mathcal{P}, Y]\!]$, then it would also be an MIS relative to $[\![\mathcal{M}, Y]\!]$ for *some* MAG $\mathcal{M}$ conforming to the PAG $\mathcal{P}$. However, this is *not* always the case and no SCM may regard $\mathbf{X}$ as an MIS with respect to $[\![\mathcal{M}, Y]\!]$.

For concreteness, consider the PAG $\mathcal{P}$ in Fig. 4 where $\{A, B\}$ is an MIS with respect to $[\![\mathcal{P}, Y]\!]$ since each $A$ and $B$ has proper possibly-directed paths to $Y$ (i.e., $A \circ\!\!-\!\!\circ Y$ and $B \circ\!\!-\!\!\circ Y$, respectively). However, we will demonstrate that at least one of $A$ or $B$ is irrelevant to reward $Y$ in every conforming MAG. To see this, we construct SCMs where the domains of variables are binary and $\forall_{U_V \in \{U_A, U_Y, U_B\}} P(U_V = 1) =$

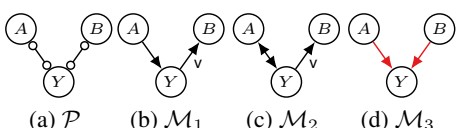

(a) $\mathcal{P}$    (b) $\mathcal{M}_1$    (c) $\mathcal{M}_2$    (d) $\mathcal{M}_3$

Figure 4: $\mathcal{M}_1$ and $\mathcal{M}_2$ are represented by $\mathcal{P}$. In contrast, $\mathcal{M}_3$ is not represented by $\mathcal{P}$.

$\epsilon \approx 0$. For a proper subset $\mathbf{X}' = \{A\}$, we can construct an SCM $\mathcal{S}_1$ following that the mechanism for $Y$ in $\mathcal{S}_1$ is defined as $f_Y = b \oplus u_Y$, and the mechanism for $B$ is $f_B = u_B$ where $\oplus$ denotes the exclusive-or operator. Then, $\mu_a = \mu_\emptyset = 2\epsilon(1 - \epsilon)$ while $\mu_{a,b^*} = \mu_{b^*} = 1 - \epsilon$ with $b^* = 1$. Thus, we find that $\mu_{a,b^*} > \mu_a$ holds in $\mathcal{S}_1$. This construction can be done for each proper subset of $\{A, B\}$, validating $\{A, B\}$ is an MIS relative to $[\![\mathcal{P}, Y]\!]$. However, the remarkable point here is that there is no representative SCM $\mathcal{S}^*$ that satisfies $\mu_{\mathbf{x}[\mathbf{X}']} \neq \mu_{\mathbf{x}}$ for *arbitrary* proper subset $\mathbf{X}' \subsetneq \mathbf{X}$, as doing so would require the mechanism $f_y$ to depend on the values of both $A$ and $B$. This setup would introduce an uncovered collider at $Y$ (i.e., $A \to Y \leftarrow B$ and $A \notin \mathrm{Adj}(B)$) in the underlying graph

---

[2]The inducing path $A \to C \leftrightarrow Y$ in $\mathcal{G}$ appears as $A \to Y$ in $\mathcal{M}$ since $C$ is an ancestor of $Y$ in $\mathcal{G}$.

of $\mathcal{P}$, leading to inconsistency with the structure of $\mathcal{P}$. Therefore, we observe that $\{A, B\}$ *is* an MIS with respect to $[\![\mathcal{P}, Y]\!]$, but at least one of $A$ or $B$ is irrelevant to reward $Y$ in all conforming MAGs.

## 3.2 Definitely MIS and Its Characterization

To address this vacuousness, we propose the concept of *definitely* MIS, which ensures that an MIS in a PAG remains an MIS in some consistent MAG. With the definition of MIS, we first choose $\mathbf{X}' \subsetneq \mathbf{X}$, and then check whether $\mu_{\mathbf{x}[\mathbf{X}']} \neq \mu_{\mathbf{x}}$ holds across all SCMs conforming to $\mathcal{D}$; here, we first choose an SCM $\mathcal{S}^*$ conforming to $\mathcal{D}$, then examine whether the inequality holds across all subsets $\mathbf{X}'$.

**Definition 2** (Definitely minimal intervention set). Given information $[\![\mathcal{D}, Y]\!]$, a set $\mathbf{X} \subseteq \mathbf{V} \setminus \{Y\}$ is called a *definitely minimal intervention set* (DMIS) relative to $[\![\mathcal{D}, Y]\!]$, denoted by $\mathbb{D}_{\mathcal{D}, Y}$ if there exists an SCM compatible with $\mathcal{D}$ such that, for every proper subset $\mathbf{X}' \subsetneq \mathbf{X}$, $\mu_{\mathbf{x}[\mathbf{X}']} \neq \mu_{\mathbf{x}}$ holds.

**Proposition 3.** *If $\mathbf{X}$ is a DMIS with respect to $[\![\mathcal{D}, Y]\!]$, then $\mathbf{X}$ is an MIS with respect to $[\![\mathcal{D}, Y]\!]$.*

*Proof.* Let $\mathcal{S}^*$ be an SCM associated with $\mathcal{D}$ such that $\mu_{\mathbf{x}[\mathbf{X}']} \neq \mu_{\mathbf{x}}$ holds for every $\mathbf{X}' \subsetneq \mathbf{X}$. For all proper subsets, such an $\mathcal{S}^*$ certifies that $\mathbf{X}$ satisfies the definition of MIS. □

**Proposition 4.** *Let $\mathcal{D}$ be either a causal diagram or a MAG (i.e., not a PAG). If $\mathbf{X}$ is an MIS with respect to $[\![\mathcal{D}, Y]\!]$, then $\mathbf{X}$ is a DMIS with respect to $[\![\mathcal{D}, Y]\!]$.*

*Proof sketch.* We can construct an SCM $\mathcal{S}^*$ where all mechanisms consist of the sum of the values of their parents, which ensures that $\mathbf{X}$ is a DMIS. □

This equivalence between MIS and DMIS for a causal diagram or a MAG (Props. 3 and 4) is derived from *determined* ancestral relations, $\mathbf{X} \subseteq \text{An}(Y)_{\mathcal{D}_{\overline{\mathbf{X}}}}$. We now move on to discuss DMIS for PAGs, where ancestral relations are *undetermined*, suggesting a notable gap between MIS and DMIS. Recall the PAG $\mathcal{P}$ in Fig. 4a, where $\{A, B\}$ is an MIS but not a DMIS with respect to $[\![\mathcal{P}, Y]\!]$.

**Proposition 5.** *Let $\mathcal{P}$ be a PAG over $\mathbf{V}$. A set $\mathbf{X} \subseteq \mathbf{V} \setminus \{Y\}$ is a DMIS relative to $[\![\mathcal{P}, Y]\!]$ if and only if there exists a MAG $\mathcal{M}$ conforming to $\mathcal{P}$ such that $\mathbf{X}$ is an MIS relative to $[\![\mathcal{M}, Y]\!]$.*

Hence, DMIS provides a *truly* feasible space for actions associated with intervention sets that no longer contain variables to rule out. According to Props. 3 and 4, we focus on establishing the graphical criterion for DMIS only for PAGs. In Fig. 4a, we have observed that $A \circ\!\!-\!\!\circ Y$ and $B \circ\!\!-\!\!\circ Y$ cannot both be an ancestor of $Y$ at the same time due to the uncovered path $A \circ\!\!-\!\!\circ Y \circ\!\!-\!\!\circ B$. To this end, we devise the notion of *relevance* among edges in a PAG.

**Definition 3** (Relevant edges). Let $\mathcal{P}$ be a PAG. For any edges $e_1(V_1 *\!\!-\!\!* V_2)$ and $e_2(V_{n-1} *\!\!-\!\!* V_n)$, we say that $e_1$ is *relevant* to $e_2$ in $\mathcal{P}$ if there exists an uncovered path $V_1 *\!\!-\!\!\circ V_2 \circ\!\!-\!\!\circ \cdots \circ\!\!-\!\!\circ V_{n-1} \circ\!\!-\!\!* V_n$ with $n \geq 3$ in $\mathcal{P}$.

**Theorem 2** (Graphical characterization of DMIS for PAGs). *Let $\mathcal{P}$ be a PAG over the set of variables $\mathbf{V}$. A set $\mathbf{X} \subseteq \mathbf{V} \setminus \{Y\}$ is a DMIS relative to $[\![\mathcal{P}, Y]\!]$ if and only if, for any pair of vertices $X, Z \in \mathbf{X}$, there exist uncovered proper possibly-directed paths from $X$ and $Z$ to $Y$ with respect to $\mathbf{X}$ such that their starting edges are not relevant.*

Consider the PAG $\mathcal{P}$ shown in Fig. 5, where $A \circ\!\!-\!\!\circ C$ and $D \circ\!\!-\!\!\circ Y$ are relevant in $\mathcal{P}$ because of the path $A \circ\!\!-\!\!\circ C \circ\!\!-\!\!\circ Y \circ\!\!-\!\!\circ D$. The key point here is that all triplets along the path are definite non-colliders so that the end nodes cannot be simultaneously ancestors of non-end nodes. Furthermore, consider any MAGs represented by $\mathcal{P}$ where $A \circ\!\!-\!\!\circ C$ appears as a directed edge out of $A$ (e.g., $\mathcal{M}_1$ and $\mathcal{M}_2$). Clearly, this results in $C \rightarrow Y \rightarrow D$, as the path is of definite status. In contrast, if any MAG contains $D \rightarrow Y$, this

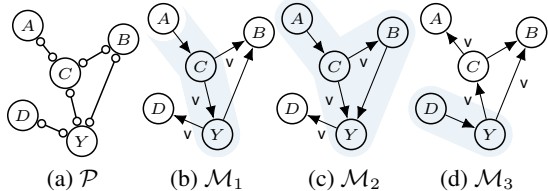

(a) $\mathcal{P}$     (b) $\mathcal{M}_1$     (c) $\mathcal{M}_2$     (d) $\mathcal{M}_3$

Figure 5: The nodes in the light blue region are the ancestors of $Y$. The three MAGs are represented by the PAG $\mathcal{P}$.

leads to $Y \rightarrow C \rightarrow A$, as in $\mathcal{M}_3$. The important observation is that $A \rightarrow C$ ensures $D \notin \text{An}(Y)_{\mathcal{M}_1}$, and $D \rightarrow Y$ ensures $A \notin \text{An}(Y)_{\mathcal{M}}$ for all MAGs $\mathcal{M}$ represented by $\mathcal{P}$. This indicates that $A$ and $D$ cannot simultaneously be ancestors of $Y$ in $\mathcal{M}$, thus $\{A, D\}$ is not a DMIS relative to $[\![\mathcal{P}, Y]\!]$.

# 4 Possibly Optimal Minimal Intervention Sets

We now refine the *possibly-optimal minimal intervention set* over DMISs rather than MISs. This refinement guarantees the existence of an underlying SCM compatible with a PAG and implies the following proposition. Note that the refined POMIS exactly matches established studies and serves as a natural extension, as supported by Props. 3 and 4, which state that MIS and DMIS coincide in causal diagrams and MAGs.

**Definition 4** (Possibly-optimal minimal intervention set (POMIS)). Let $\mathbf{X} \subseteq \mathbf{V} \setminus \{Y\}$ be a DMIS relative to $[\![\mathcal{D}, Y]\!]$. If there exists an SCM conforming to $\mathcal{D}$ such that $\mu_{\mathbf{x}^*} > \forall_{\mathbf{W} \in \mathbb{D}_{\mathcal{D},Y} \setminus \{\mathbf{X}\}} \mu_{\mathbf{w}^*}$, then $\mathbf{X}$ is a *possibly-optimal minimal intervention set* (POMIS) relative to $[\![\mathcal{D}, Y]\!]$.

**Proposition 6.** *Let $\mathcal{P}$ be a PAG over $\mathbf{V}$. A set $\mathbf{X} \subseteq \mathbf{V} \setminus \{Y\}$ is a POMIS relative to $[\![\mathcal{P}, Y]\!]$ if and only if there exists a MAG $\mathcal{M}$ conforming to $\mathcal{P}$ such that $\mathbf{X}$ is a POMIS relative to $[\![\mathcal{M}, Y]\!]$.*

We investigate into the graphical characterization of POMISs for MAGs and PAGs. The main challenge in characterizing POMIS for ancestral graphs lies in the fact that induced paths by latent variables (or UCs) do not explicitly appear, which makes it impossible to directly identify the unobserved confounders' territory as for causal diagrams. Instead, we leverage edge's *visibility* which indicates that the edge is not confounded in any underlying causal diagram (Lem. 2 in Appendix B for details). To generalize the UC-territory, we introduce a *possible c-component*, which provides a necessary condition for nodes to belong to the same c-component in an underlying causal diagram.

**Definition 5** (pc-component [Jaber et al., 2018]). Two nodes are in the same *possible c-component* (pc-component) if there is a path between them such that (i) all non-endpoint nodes along the path are colliders, and (ii) none of the edges are visible.

We denote the pc-component of a partial mixed graph (PMG) $\mathcal{Q}$ containing $X$ as $\mathsf{PC}(X)_{\mathcal{Q}}$ and $\mathsf{PC}(\mathbf{X})_{\mathcal{Q}} \triangleq \bigcup_{X \in \mathbf{X}} \mathsf{PC}(X)_{\mathcal{Q}}$. For example, $A$ and $B$ are in the same pc-component in $\mathcal{P}$ of Fig. 1e because they are connected through an invisible colliding path $A \circ\!\!\rightarrow C \leftarrow\!\!\circ B$, i.e., $\mathsf{PC}(A)_{\mathcal{P}} = \{A, B, C\}$. Furthermore, due to $A \notin \mathsf{PC}(Y)_{\mathcal{P}} = \{B, Y\}$, $A$ and $Y$ cannot belong to the same c-component in any causal diagrams conforming to $\mathcal{P}$. We now generalize MUCT and IB for PMGs.

**Definition 6** (Unobserved-confounders' territory for PMGs). Given information $[\![\mathcal{Q}, Y]\!]$ and intervention set $\mathbf{X} \subseteq \mathbf{V} \setminus \{Y\}$, let $\mathcal{H} = \mathcal{Q}[\mathtt{PossAn}(Y)_{\mathcal{Q}} \setminus \mathbf{X}]$. A set of variables $\mathbf{T} \subseteq \mathtt{PossAn}(Y)_{\mathcal{Q}} \setminus \mathbf{X}$ containing $Y$ is called a *UC-territory* on $\mathcal{Q}$ with respect to $Y$ if $\mathtt{PossDe}(\mathbf{T})_{\mathcal{H}} = \mathbf{T}$ and $\mathsf{PC}(\mathbf{T})_{\mathcal{H}} = \mathbf{T}$. A UC-territory $\mathbf{T}$ is called a *minimal* UC-territory (MUCT) if no $\mathbf{T}' \subsetneq \mathbf{T}$ is a UC-territory.

**Definition 7** (Interventional border for PMGs). Let $\mathbf{T}$ be a minimal UC-territory with respect to $[\![\mathcal{Q}, Y, \mathbf{X}]\!]$. Then $\mathtt{Pa}(\mathbf{T})_{\mathcal{Q}} \setminus \mathbf{T}$ is called an *interventional border* (IB) with respect to $[\![\mathcal{Q}, Y, \mathbf{X}]\!]$.

For concreteness, consider $\mathcal{M}$ and $\mathbf{X} = \{A, B\}$ in Fig. 3. Here, we omit $\mathtt{Poss}$, as we discuss in the context of a MAG. Let $\mathcal{H}$ be the induced graph $\mathcal{M}[\mathtt{An}(Y)_{\mathcal{M}} \setminus \mathbf{X}]$. In $\mathcal{H}$, the nodes $C$ and $Y$ are in the same pc-component, and $D$ is a descendant of $C$. This implies that $\mathbf{T} = \{C, D, Y\}$ is the minimal closed set for $\mathtt{De}_{\mathcal{H}}$ and $\mathsf{PC}_{\mathcal{H}}$, leading to $\mathsf{IB}(\mathcal{M}, Y, \mathbf{X}) = \{A, B\}$[3], derived from $\mathtt{Pa}(\mathbf{T})_{\mathcal{M}} \setminus \mathbf{T}$.

## 4.1 Characterization of POMIS for MAGs

With the MUCT and IB for PMGs established, we are now ready to characterize POMISs for MAGs.

**Theorem 3** (Graphical characterization of POMIS for MAGs). *Let $\mathcal{M}$ be a MAG over the set of variable $\mathbf{V}$. A set $\mathbf{X} \subseteq \mathbf{V} \setminus \{Y\}$ is a POMIS relative to $[\![\mathcal{M}, Y]\!]$ if and only if $\mathbf{X} = \mathsf{IB}(\mathcal{M}, Y, \mathbf{X})$.*

For example, consider the MAG $\mathcal{M}$ in Fig. 3c where $\mathsf{IB}(\mathcal{M}, Y, \{A, B\}) = \{A, B\}$ holds. Therefore, we get that $\{A, B\}$ is a POMIS with respect to $[\![\mathcal{M}, Y]\!]$. Indeed, as previously shown, $\mathcal{G}$ in Fig. 2 represented by $\mathcal{M}$ and $\{A, B\}$ is a POMIS with respect to $[\![\mathcal{G}, Y]\!]$.

## 4.2 Characterization of POMIS for PAGs.

The remainder of the main body focuses on characterizing POMIS for PAGs. We first present necessary conditions for a PMG to represent MAGs $\mathcal{M}$ in which $\mathbf{X}$ is a POMIS relative to $[\![\mathcal{M}, Y]\!]$.

---

[3]We denote MUCT and IB with respect to $[\![\mathcal{Q}, Y, \mathbf{X}]\!]$ as $\mathsf{MUCT}(\mathcal{Q}, Y, \mathbf{X})$ and $\mathsf{IB}(\mathcal{Q}, Y, \mathbf{X})$, respectively.

---

**Algorithm 1:** Identify whether a given set is a POMIS for PAG.

---

1 **function** IsPOMIS($\mathcal{P}, Y, \mathbf{X}$)
   **Input:** $\mathcal{P}$: PAG, $Y$: reward, $\mathbf{X}$: Intervention set
2    **if** given $\mathbf{X}$ does not satisfy Thm. 2 **then return** False
3    Let $\mathcal{Q}_{\mathbf{X}}$ be a PMG oriented from $\mathcal{P}$ with $\mathbf{X}$ according to Prop. 7.
4    **return** subIsPOMIS($\mathcal{Q}_{\mathbf{X}}, \mathbf{X} \cup \{Y\}, Y, \mathbf{X}$)

5 **function** subIsPOMIS($\mathcal{Q}, \mathbf{A}, Y, \mathbf{X}$)
6    **if** $\mathbf{A}$ is empty **then return** IB($\mathcal{Q}, Y, \mathbf{X}$) $= \mathbf{X}$
7    $A \leftarrow$ Pick a node from $\mathbf{A}$.
8    **for** each set $\mathbf{C}_A^{\mathcal{Q}} \subseteq \{V \in \text{Adj}(A)_{\mathcal{Q}} \mid A \circ\!\!-\!\!* V\}$ **do**
9       **if** $\mathbf{C}_A^{\mathcal{Q}}$ satisfies Thm. 7 (i.e., check validity of local transformation) and
          $Y \in \text{PossDe}(A)_{\mathcal{Q}\setminus\mathbf{C}_A^{\mathcal{Q}}}$ **then**
10          Let $\mathcal{Q}'$ be the PMG obtained by orienting the circle marks around $A$ following $\mathbf{C}_A^{\mathcal{Q}}$
             and completing the orientation rules from $\mathcal{Q}$.
11          **if** subIsPOMIS($\mathcal{Q}', \mathbf{A} \setminus \{A\}, Y, \mathbf{X}$) **then return** True
12    **return** False

---

**Proposition 7.** *Let $\mathcal{Q}_{\mathbf{X}}$ be a PMG representing MAGs where $\mathbf{X}$ is a POMIS with respect to $Y$. Then, the following properties hold in $\mathcal{Q}_{\mathbf{X}}$, for every $X \in \mathbf{X}$:*

1. *Every uncovered proper possibly-directed path from $X$ to $Y$ relative to $\mathbf{X}$ ends with an arrowhead ($>$).*

2. *If $X$ is adjacent to $Y$, then the edge between $X$ and $Y$ is a directed edge ($X \to Y$).*

Put simply, violating these conditions introduces an almost directed cycle or directed cycle. To characterize POMIS for PAGs, we partition $[\mathcal{Q}_{\mathbf{X}}]$ based on the orientation of circle marks incident on $\mathbf{X} \cup \{Y\}$. We refer to a *local transformation* [Wang et al., 2023b] $\mathbf{C}_A^{\mathcal{Q}} \subseteq \{V \in \text{Adj}(A)_{\mathcal{Q}} \mid A \circ\!\!-\!\!* V\}$ as the vertices whose edges with a circle at $A$ (i.e., $A \circ\!\!-\!\!* V$ in $\mathcal{Q}$) will be oriented with arrowheads at $A$ (i.e., $A \leftarrow\!\!* V$); all remaining edges $A \circ\!\!-\!\!* V'$ will be oriented as $A \to\!\!* V'$.

**Proposition 8.** *For every MAG $\mathcal{M} \in [\mathcal{Q}_{\mathbf{X}}]$, if $\mathbf{X}$ is a POMIS relative to $[\![\mathcal{M}, Y]\!]$, then there exists a PMG $\mathcal{Q}_{\mathbf{X}}^i$ representing $\mathcal{M}$ such that the following conditions are satisfied:*

1. *Every circle mark around $\mathbf{X} \cup \{Y\}$ in $\mathcal{Q}_{\mathbf{X}}$ is oriented as either a tail ($-$) or an arrowhead ($>$) in $\mathcal{Q}_{\mathbf{X}}^i$ according to valid local transformations[4].*

2. *Every $X \in \mathbf{X}$ is an ancestor of $Y$ in $\mathcal{Q}_{\mathbf{X}}^i$.*

3. *$\mathcal{Q}_{\mathbf{X}}^i$ is closed under orientation rules.[5]*

In words, $\mathcal{Q}_{\mathbf{X}}^i$ is a *more* oriented PMG instance derived from $\mathcal{Q}_{\mathbf{X}}$ by applying the valid local transformations for circle marks around $\mathbf{X} \cup \{Y\}$, along with the orientation rules, while confirming that $\mathbf{X}$ is an ancestor of $Y$ in all MAGs $\mathcal{M} \in [\mathcal{Q}_{\mathbf{X}}^i]$. For clarity, recall the PAG $\mathcal{P}$ in Fig. 5a with a DMIS $\mathbf{X} = \{C\}$. Here, every MAG $\mathcal{M} \in [\mathcal{P}]$ satisfying that $\{C\}$ is a POMIS conforms to $\mathcal{Q}_{\{C\}}$ in Fig. 6a. Each $\mathcal{Q}_{\{C\}}^1$ (Fig. 6b; corresponding to $\mathbf{C}_Y^{\mathcal{Q}_{\{C\}}} = \{B\}$ and $\mathbf{C}_C^{\mathcal{Q}_{\{C\}}} = \{B\}$) and $\mathcal{Q}_{\{C\}}^2$ (Fig. 6c;

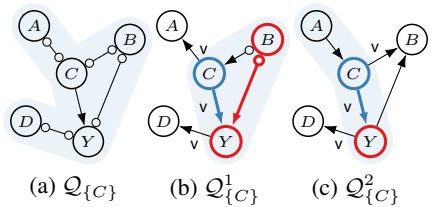

(a) $\mathcal{Q}_{\{C\}}$     (b) $\mathcal{Q}_{\{C\}}^1$     (c) $\mathcal{Q}_{\{C\}}^2$

Figure 6: The light blue region indicates possible ancestors of $Y$.

corresponding to $\mathbf{C}_Y^{\mathcal{Q}_{\{C\}}} = \emptyset$ and $\mathbf{C}_C^{\mathcal{Q}_{\{C\}}} = \{A\}$) illustrates a PMG where local transformations for $C$ and $Y$ are oriented, and both graphs are closed under the orientation rules.

---

[4]For example, $\mathbf{C}_Y^{\mathcal{Q}_{\{C\}}} = \emptyset$ with $\mathbf{C}_C^{\mathcal{Q}_{\{C\}}} = \{B\}$ is invalid local transformation, as it implies $Y \to B \circ\!\to C \to Y$ which introduces either an directed or almost directed cycle. The complete graphical criterion of the validity of proposed by Wang et al. [2023b] is presented in Thm. 7 of Appendix B.

[5]The orientation rules refer to $\mathcal{R}_1-\mathcal{R}_3, \mathcal{R}_4', \mathcal{R}_8-\mathcal{R}_{10}$ and $\mathcal{R}_{\text{SB}}$. $\mathcal{R}_5-\mathcal{R}_7$ are not considered since we assume no selection bias. Further details regarding the orientation rules are provided in Appendix B.

**Theorem 4** (Characterization of POMIS for PAGs). *A set $\mathbf{X} \subseteq \mathbf{V} \setminus \{Y\}$ is a POMIS relative to $[\![\mathcal{P}, Y]\!]$ if and only if there exists $\mathcal{Q}_{\mathbf{X}}^i$ satisfying Props. 7 and 8 such that $\mathsf{IB}(\mathcal{Q}_{\mathbf{X}}^i, Y, \mathbf{X}) = \mathbf{X}$.*

The key observation is that local transformations restricted to $\mathbf{X} \cup \{Y\}$ are sufficient for this determination, thereby circumventing the need to enumerate all MAGs represented by the given PAG. To witness, consider $\mathcal{Q}_{\{C\}}^1$ with $\mathbf{X} = \{C\}$ in Fig. 6b where $\mathsf{IB}(\mathcal{Q}_{\{C\}}^1, Y, \mathbf{X}) = \mathbf{X}$ holds, and it follows that $\{C\}$ is a POMIS with respect to $[\![\mathcal{P}, Y]\!]$. Indeed, we can find a MAG instance $\mathcal{M}$ by orienting the circle marks around $B$ in $\mathcal{Q}_{\{C\}}^1$ as tails, in which $\mathsf{IB}(\mathcal{M}, Y, \mathbf{X}) = \mathbf{X}$ (Thm. 3) also holds.

### 4.3 Algorithmic Approach: Enumerating POMISs

We now present an algorithm IsPOMIS (Alg. 1), through which we can determine whether a given set $\mathbf{X}$ is a POMIS relative to $[\![\mathcal{P}, Y]\!]$ based on our theoretical results (Thms. 2 and 4).

**Theorem 5** (Soundness and completeness). *The algorithm IsPOMIS (Alg. 1) returns True if and only if there exists a MAG $\mathcal{M}$ conforming to $\mathcal{P}$ such that $\mathbf{X}$ is a POMIS relative to $[\![\mathcal{M}, Y]\!]$.*

The algorithm begins by checking whether the given set $\mathbf{X}$ is a DMIS (Line 2). If $\mathbf{X}$ is identified as a DMIS, it then infuses the necessary condition in Prop. 7 (Line 3). Subsequently, the local transformations for $\mathbf{X} \cup \{Y\}$ are oriented recursively within subIsPOMIS. During each recursion, it evaluates the validity of a local transformation around a vertex and the ancestral relations between the vertex and the reward $Y$ (Line 9). The PMG is updated based on local transformations and orientation rules (Lines 10–11). Finally, in the base case (Line 6), the algorithm checks whether the fully-oriented PMG $\mathcal{Q}_{\mathbf{X}}^i$ satisfies $\mathsf{IB}(\mathcal{Q}_{\mathbf{X}}^i, Y, \mathbf{X}) = \mathbf{X}$ based on Thm. 4.

**Runtime analysis.** In the algorithm, Line 2 runs in $\mathcal{O}(|\mathbf{V}|^5)$ time, using standard reachability algorithm. Each local transformation (Line 8) requires $\mathcal{O}(2^p)$ space, where $p < |\mathbf{V}|$ denotes the number of circle marks around the current vertex. Both the validation of local transformations and the check for ancestral relations (Line 9) take $\mathcal{O}(|\mathbf{V}|^3)$ [Wang et al., 2023b].

Identifying all POMIS sets requires checking all subsets of $\mathbf{V} \setminus \{Y\}$ using IsPOMIS (Alg. 1), and thus the size of the search space grows exponentially. However, since all non-DMIS sets are filtered out in Line 2, the enumeration process effectively depends only on the number of DMISs[6].

A naive approach is to enumerate all possible MAGs $\mathcal{M}$ that conform to a given PAG, and verify whether $\mathsf{IB}(\mathcal{M}, Y, \mathbf{X}) = \mathbf{X}$ holds for each $\mathcal{M}$. However, this method presents analytical challenges in terms of complexity—the enumeration process may generate many duplicate MAGs, and it is difficult to determine when the transformation should terminate. Even under optimistic assumptions—namely, that no duplicate MAGs are produced and that an oracle informs us when all MAGs consistent with the PAG have been generated—the number of such MAGs remains super-exponentially large [Zhang, 2012, Wang et al., 2023a]. Even when adopting MAGLIST [Wang et al., 2024a], which systematically enumerates MAGs via local transformations, its worst-case complexity remains higher than ours, as it performs transformations over the entire set $\mathbf{V}$, whereas IsPOMIS constructs only distinct PMGs $\mathcal{Q}_{\mathbf{X}}^i$ oriented through local transformations around $\mathbf{X} \cup \{Y\} \subseteq \mathbf{V}$ for a DMIS $\mathbf{X}$.

## 5 Experiments

We evaluate the cumulative regrets (CR) of SCM-MAB under different strategies to assess the effect of employing POMIS for PAGs (Fig. 7). The number of trials is set to 10,000 for Tasks 1 and 2, and 5,000 for Task 3, which is sufficient to observe performance differences. Each simulation is repeated 1,000 times to obtain consistent results. We compare three arm-selection strategies: POMISs (pink), DMISs (purple), and Brute-force (BF; green), each combined with two prominent solvers: Thompson Sampling (TS) and KL-UCB. In the Brute-force strategy (i.e., without causal knowledge), all possible combinations of arms are evaluated; that is, the number of possible intervention sets of BF is $2^{|\mathbf{V}|-1}$ and the total number of corresponding arms is $\sum_{i=0}^{|\mathbf{V}|-1} \binom{|\mathbf{V}|-1}{i} K^i = (K+1)^{|\mathbf{V}|-1}$ where $K$ denotes the cardinality. We assume that all variables are binary for simplicity ($K = 2$).

---

[6]Although Lee and Bareinboim [2018] provided an efficient algorithm for enumerating all POMISs in a causal diagram by leveraging a topological order, such an approach is *not* applicable to PAGs, where the topological order is not determined.

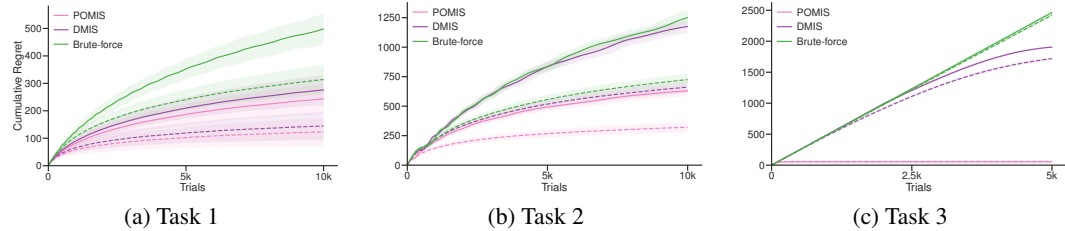

| (a) Task 1 | (b) Task 2 | (c) Task 3 |

Figure 7: Cumulative regrets for the corresponding KL-UCB (solid) and TS (dashed) under distinct strategies. We plot the average cumulative regrets along with their standard deviations.

The underlying model mechanisms are randomly generated by combining binary logical operations, and the exogenous variables are set to follow Bernoulli distributions whose parameters are randomly selected over $(0, 1)$. Additional details and experiments are provided in Appendix D.

**Task 1.** The deployed agent can only access the PAG $\mathcal{P}$ in Fig. 5a to obtain DMISs and POMISs. The environment in which an agent interacts is consistent with $\mathcal{P}$. Using three strategies (BF: 81 arms, DMIS: 25 arms, POMIS: 19 arms), the POMIS-based TS and KL-UCB achieve CRs of 123.4 and 243.4, which correspond to **39.3**% and **48.9**%, respectively, of CR for BF.

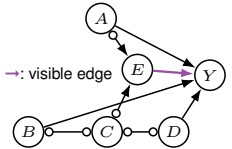

Figure 8: Six variables.

**Task 2.** We consider the PAG in Fig. 8 to validate our result. Using three strategies (BF: 243 arms, DMIS: 195 arms, POMIS: 85 arms), the POMIS-based TS and KL-UCB achieve CRs of 320.9 and 629.9, which correspond to **44.3**% and **50.4**%, respectively, of CR for BF.

**Task 3.** We consider a more involved scenario (Fig. 9) to validate our result. Using three strategies (BF: 19683 arms, DMIS: 2025 arms, POMIS: 54 arms), the POMIS-based TS and KL-UCB achieve CRs of 60.3 and 52.0, which correspond to only **2.5**% and **2.1**%, respectively, of CR for BF. Notably, the size of the POMIS arms accounts for only $\frac{54}{19683}$=**0.27**% of that of the BF. We observe that the superiority of POMIS remains consistent regardless of the solvers used. All CRs and the numbers of sets and arms are provided in Tables 1 and 2 in Appendix D. These results demonstrate that refining arms by taking into account the Markov equivalence class represented by a PAG enhances the efficiency of agents.

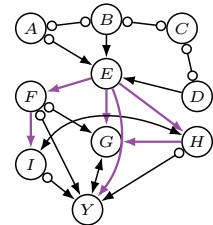

Figure 9: Ten variables.

## 6 Conclusion

We proposed a novel structured causal bandit strategy (SCM-MAB) in the context of ancestral graphs. We first provided a graphical characterization of MIS for MAGs and PAGs. We then demonstrated the *vacuousness* of MIS, i.e., that some MISs for a PAG are not MISs of any MAG consistent with that PAG. To address this, we introduced the notion of a *definitely* minimal intervention set (DMIS), which guarantees the existence of an underlying SCM, thereby aligning with the general intuition behind the concept of MIS. Finally, we refined the concept of POMIS over DMIS from MIS and provided a complete characterization of POMIS for MAGs and PAGs, along with an algorithm for enumerating all POMISs given a PAG. We believe these results have practical implications for the design of intelligent agents, providing a foundation for optimizing the action space when the environment is not fully accessible but is abstracted as a Markov equivalence class.

## Acknowledgments and Disclosure of Funding

This work was supported in part by the NSF, ONR, AFOR, DOE, Amazon, JP Morgan, and The Alfred P. Sloan Foundation. Min Woo Park and Sanghack Lee were partly supported by the IITP (RS-2022-II220953/25%, RS-2025-02263754/25%) and NRF (RS-2023-00211904/25%, RS-2023-00222663/25%) grant funded by the Korean government.

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

# Contents

# Appendix for "Structural Causal Bandits under Markov Equivalence"

## A    Related Works

The integration of causal inference into the MAB framework has opened new avenues for modeling and solving decision problems with richer dependency structures [Bareinboim et al., 2024]. Causal diagrams [Pearl, 1995] have been employed to represent causal relationships among actions, rewards, and other relevant factors. This approach enables agents to make informed decisions by considering how each action causally influences the reward through causal pathways. Existing studies [Bareinboim et al., 2015, Lattimore et al., 2016, Forney et al., 2017] have shown that causality-aware strategies can significantly outperform MAB algorithms that do not account for such underlying causal relationships. Subsequent work has explored various specialized settings by introducing additional structural assumptions, such as the availability of both observational and experimental distributions, or linear mechanisms [Zhang and Bareinboim, 2017, Lu et al., 2020, Bilodeau et al., 2022, De Kroon et al., 2022, Feng and Chen, 2023, Varici et al., 2023].

Lu et al. [2021] were the first to study causal bandits without assuming access to the full causal diagram. Their approach targets the atomic setting in which the reward variable has only a single parent, reducing the problem to identifying that parent for optimal intervention. They further assume that the agent instead observes the skeleton of the true causal diagram. Extending this line of work, Konobeev et al. [2023] eliminated the need for prior knowledge of the graph skeleton. However, their setting remains restricted to the same atomic case. More recently, Feng et al. [2023] considered causal bandits in which each action corresponds to an intervention on a set of variables. Yan and Tajer [2024] considered actions as *soft* interventions on variables, i.e., changing the conditional distribution $P(v_i \mid \mathbf{pa}_i)$ to $Q(v_i \mid \mathbf{pa}_i)$. Despite this generalization, all these approaches assumed *causal sufficiency* and thus do not account for the presence of latent variables. Malek et al. [2023] provided some results for settings with unknown graph structures, the authors initially highlight the challenge posed by the exponentially large number of arms in causal bandit problems under unknown graphs, and assumed that no confounding exists between the reward variable and its ancestors.

Lee and Bareinboim [2018] formalized the *structural causal bandit* (SCM-MAB) framework, in which a bandit instance is structured by an SCM, and each action corresponds to an intervention on a subset of variables. They proposed a sound and complete graphical characterization to identify *minimal intervention sets* (MISs) and *possibly-optimal minimal intervention sets* (POMISs), where the former includes only the variables that affect the reward, and the latter refers to actions that could be part of an optimal strategy among MISs, thereby guiding the agent to avoid unnecessary exploration, without any actual interaction. Lee and Bareinboim [2019] extended this approach to accommodate scenarios involving non-manipulable variables among all the variables in the graph. Lee and Bareinboim [2020] established the framework under stochastic policies and demonstrated the informativeness of such policies. Everitt et al. [2021] and Carey et al. [2024] further investigated the completeness of the graphical characterization of optimal policy spaces, although the general completeness remains an open problem. Wei et al. [2023] proposed a parameterization-based approach to incorporate shared information among possibly-optimal actions. Elahi et al. [2024a] extended the SCM-MAB framework to settings where no causal graph is assumed to be accessible, requiring their algorithm to perform causal discovery—i.e., constructing the causal structure—during online interaction. In contrast, our work investigates a graphical approach that eliminates unnecessary actions *a priori*, given a partial ancestral graph, before the interaction begins. A detailed comparison between our work and Elahi et al. [2024a] is presented in Appendix E.3.

Building on this line of work, causal Bayesian optimization (CBO; Aglietti et al. [2020]) leverages the systematic characterization of MIS and POMIS for structural pruning in continuous action spaces, and Bhatija et al. [2025] extend it to a multi-outcome variant incorporating Pareto optimality.

# B Additional Preliminaries and Background Results

In this section, we provide additional preliminaries from previous works (B.1) and background results relevant to our study (B.2).

## B.1 Additional Preliminaries

**Definite status.** Let $p$ be any path in a PMG, and $\langle X, Z, Y \rangle$ be any consecutive triple along $p$. We say that $Z$ is a *definite collider* on $p$ if both edges are directed into $Z$. If one of the edges is out of $Z$, or both edges have a circle mark at $Z$ (i.e., $X \ast\!\!-\!\!\circ Z \circ\!\!-\!\!\ast Y$) and there is no edge between $X$ and $Z$, then we say that $Z$ is a *definite non-collider* on $p$. A path is said to have a *definite status* if every non-endpoint node along it is either a definite collider or a definite non-collider.

**Markov equivalence class.** Multiple MAGs can entail the same m-separation[7] relationships. Such MAGs constitute a Markov equivalence class (MEC). The Markov equivalence class of MAGs can be uniquely represented by a PMG which we refer to as a PAG.

**Definition 8** (Markov equivalence [Zhang, 2012]). *Two MAGs $\mathcal{M}_1$, $\mathcal{M}_2$ with $\mathbf{V}(\mathcal{M}_1) = \mathbf{V}(\mathcal{M}_2)$ are Markov equivalent if for any three disjoint sets of vertices $\mathbf{X}, \mathbf{Y}, \mathbf{Z}$, $\mathbf{X}$ and $\mathbf{Y}$ are m-separated by $\mathbf{Z}$ in $\mathcal{M}_1$ if and only if $\mathbf{X}$ and $\mathbf{Y}$ are m-separated by $\mathbf{Z}$ in $\mathcal{M}_2$.*

A path between $X$ and $Y$, $p = \langle X, \cdots, W, Z, Y \rangle$, is a *discriminating path* for $Z$ if (i) $p$ includes at least three edges; (ii) $Z$ is a non-endpoint vertex on $p$, and is adjacent to $Y$ on $p$; and (iii) $X$ is not adjacent to $Y$, and every vertex between $X$ and $Z$ is a collider on $p$ and is a parent of $Y$.

For two MAGs to be in the same Markov equivalence class, discriminating paths must either be present in both graphs or none of the graphs, as well as the same skeleton and unobserved colliders.

**Lemma 1** (Graphical characterization of MEC [Spirtes and Richardson, 1997, Zhang, 2012]). *Two MAGs $\mathcal{M}_1$ and $\mathcal{M}_2$ with $\mathbf{V}(\mathcal{M}_1) = \mathbf{V}(\mathcal{M}_2)$ are Markov equivalent if and only if*

*(i) they have the same adjacencies;*

*(ii) they have the same uncovered colliders; and*

*(iii) if some path is a discriminating path for a vertex $V$ in both graphs $\mathcal{M}_1$ and $\mathcal{M}_2$, then $V$ is a collider on the path in $\mathcal{M}_1$ if and only if it is a collider on the path in $\mathcal{M}_2$.*

A collider path $\langle V_1, \cdots, V_k \rangle$ is called a *minimal collider path* if $V_1$ is not adjacent to $V_k$, and no subsequence of the path is also a collider path.

The two conditions (ii) and (iii) can be expressed as a condition for two MAGs to share the same minimal colliding paths [Zhao et al., 2005]. Identifying Markov equivalence of a pair of MAGs is tractable with worst-case runtime $\mathcal{O}(|\mathbf{V}|^3)$ [Wienöbst et al., 2022].

**Visible edges.** A directed edge $X \to Y$ is *visible* if there exists no causal diagram in the corresponding equivalence class where there is an inducing path between $X$ and $Y$ that is into $X$. We refer to any edge that is not visible as *invisible*.

**Lemma 2** (Graphical characterization of visibility [Zhang, 2006, Maathuis and Colombo, 2015]). *A directed edge $X \to Y$ is visible if*

*(i) there is a vertex $Z$ not adjacent to $Y$, such that there is an edge between $Z$ and $X$ that is into $X$ ($Z \ast\!\!\to X$); or*

*(ii) there is a collider path between $Z$ and $X$ that is into $X$ ($Z \ast\!\!\to \cdot \leftrightarrow \cdots \leftrightarrow X$) and every vertex on the path except $Z$ is a parent of $Y$.*

---

[7]M-separation [Richardson and Spirtes, 2002] refers to an extension of d-separation [Pearl and Robins, 1995b] for ancestral graphs.

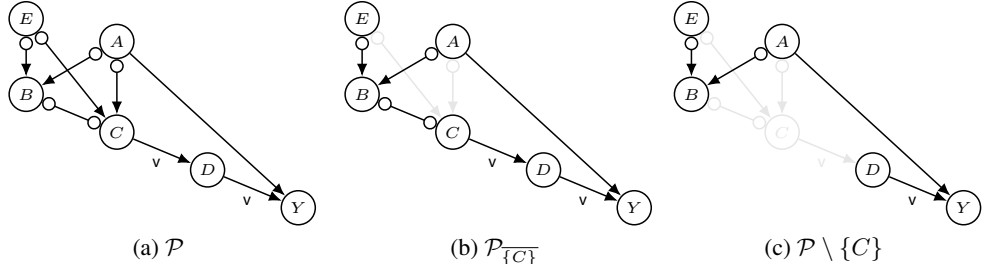

$$\text{(a) } \mathcal{P} \qquad\qquad \text{(b) } \mathcal{P}_{\overline{\{C\}}} \qquad\qquad \text{(c) } \mathcal{P} \setminus \{C\}$$

Figure 10: (a) PAG $\mathcal{P}$, (b) $\{C\}$-upper-manipulated graph, and (c) induced graph over $\mathbf{V}(\mathcal{P}) \setminus \{C\}$. In MAGs and PAGs, the visibility is preserved from $\mathcal{P}$ (see Lem. 15). For example, although there is no edge oriented into $D$ in $\mathcal{P} \setminus \{C\}$, the directed edge $D \to Y$ remains visible.

It is important to note that (i) an invisible edge $X \to Y$ does not necessarily imply that $X$ and $Y$ are confounded in every underlying causal graph; and (ii) invisible edges should not be considered independently. To witness, consider a scenario where we have $X \leftarrow Y \to Z$ in a MAG $\mathcal{M}$, and $X$ and $Z$ are not adjacent. Since both edges, $X \leftarrow Y$ and $Y \to Z$, are invisible, causal diagrams can include at most one of the following structures added to $\mathcal{M}$: $X \leftarrow L_1 \leftarrow \cdots \to L_n \to Y$ or $Y \leftarrow L_1 \leftarrow \cdots \to L_n \to Z$ ($X \leftrightarrow Y$, or $Y \leftrightarrow Z$). Adding any one of these does not introduce a new collider between $X$ and $Z$, thereby maintaining conformity with $\mathcal{M}$. However, if both are added simultaneously, a new collider is introduced at $Y$, resulting in a causal diagram that is not represented by $\mathcal{M}$.

**Manipulations.** Given a causal diagram $\mathcal{G}$ and a set of variables $\mathbf{X}$ therein, the $\mathbf{X}$-lower-manipulation of $\mathcal{G}$ deletes all edges in $\mathcal{G}$ that are out of the variables in $\mathbf{X}$. The resulting graph is denoted by $\mathcal{G}_{\underline{\mathbf{X}}}$. The $\mathbf{X}$-upper-manipulation of $\mathcal{G}$ deletes all edges in $\mathcal{G}$ that are into variables in $\mathbf{X}$. The resulting graph is denoted by $\mathcal{G}_{\overline{\mathbf{X}}}$.

Given a PMG $\mathcal{Q}$ and a set of variables $\mathbf{X}$ therein, the $\mathbf{X}$-lower-manipulation of $\mathcal{Q}$ deletes all those edges that are visible in $\mathcal{Q}$ and are out of variables in $\mathbf{X}$ and replaces all those edges that are out of variables in $\mathbf{X}$ but are invisible in $\mathcal{Q}$ with bidirected edges. The resulting graph is denoted as $\mathcal{Q}_{\underline{\mathbf{X}}}$. The $\mathbf{X}$-upper-manipulation of $\mathcal{Q}$ deletes all edges in $\mathcal{Q}$ that are into variables in $\mathbf{X}$, and otherwise keeps $\mathcal{Q}$ as it is.

The manipulated graphs play a crucial role in the derivation of do-calculus.

**Do-calculus.** Pearl [1995] devised *do-calculus* which acts as a bridge between observational and interventional distributions from a causal diagram without relying on any parametric assumptions. Zhang [2008b] proposed the do-calculus for MAGs and PAGs (also known as Zhang's calculus). Jaber et al. [2022] noted that there are cases where Pearl's do-calculus rules are applicable to every causal diagram within a given PAG, but Zhang's calculus cannot be applied to the same PAG. To address this, Jaber et al. [2022] proposed a refined version of do-calculus for PAGs and demonstrated that whenever the proposed rule is not applicable given a PAG, then the corresponding rule in Pearl's calculus is not applicable for some causal diagram in the Markov equivalence class represented by the PAG.

Here, we present do-calculus for PAGs, which encompasses that for MAGs.

**Definition 9** (Definite m-connecting path [Jaber et al., 2022]). In a PAG, a path $p$ between $X$ and $Y$ is a *definite m-connecting path* relative to a set of nodes $\mathbf{Z}$ if $p$ is definite status, every definite non-collider on $p$ is not a member of $\mathbf{Z}$, and every collider on $p$ is a ancestor of some member of $\mathbf{Z}$. $X$ and $Y$ are *m-separated* by $\mathbf{Z}$ if there is no definite m-connecting path between them relative to $\mathbf{Z}$.

**Theorem 6** (Do-calculus for PAGs [Jaber et al., 2022]). *Let $\mathcal{P}$ be the PAG over $\mathbf{V}$, and $\mathbf{X}$, $\mathbf{Y}$, $\mathbf{W}$, $\mathbf{Z}$ be disjoint subsets of $\mathbf{V}$. The following rules are valid, in the sense that if the antecedent of the rule holds, then the consequent holds in every MAG and consequently every causal diagrams represented by $\mathcal{P}$.*

***Rule 1.*** $P(\mathbf{y} \mid do(\mathbf{w}), \mathbf{x}, \mathbf{z}) = P(\mathbf{y} \mid do(\mathbf{w}), \mathbf{z}) \qquad\qquad$ *if $\mathbf{X}$ and $\mathbf{Y}$ are m-separated by $\mathbf{W} \cup \mathbf{Z}$ in $\mathcal{P}_{\overline{\mathbf{W}}}$*

***Rule 2.*** $P(\mathbf{y} \mid do(\mathbf{w}), do(\mathbf{x}), \mathbf{z}) = P(\mathbf{y} \mid do(\mathbf{w}), \mathbf{x}, \mathbf{z})$     *if* $\mathbf{X}$ *and* $\mathbf{Y}$ *are m-separated by* $\mathbf{W} \cup \mathbf{Z}$ *in* $\mathcal{P}_{\overline{\mathbf{W}}, \underline{\mathbf{X}}}$

***Rule 3.*** $P(\mathbf{y} \mid do(\mathbf{w}), do(\mathbf{x}), \mathbf{z}) = P(\mathbf{y} \mid do(\mathbf{w}), \mathbf{z})$     *if* $\mathbf{X}$ *and* $\mathbf{Y}$ *are m-separated by* $\mathbf{W} \cup \mathbf{Z}$ *in* $\mathcal{P}_{\overline{\mathbf{W}}, \overline{\mathbf{X}(\mathbf{Z})}}$

*where* $\mathbf{X}(\mathbf{Z}) \triangleq \mathbf{X} \setminus \texttt{PossAn}(\mathbf{Z})_{\mathcal{P}[\mathbf{V} \setminus \mathbf{W}]}$.

**Induced graph.** A subgraph $\mathcal{Q}[\mathbf{A}]$ is defined as a vertex-induced subgraph in which all the edges among the vertices in $\mathbf{A} \subseteq \mathbf{V}(\mathcal{Q})$ are preserved while maintaining the visibility from $\mathcal{Q}$ (see Fig. 10).

**Chordal graph.** We also introduce some useful graph theory and terminology, excerpted from Maathuis et al. [2009] and Wang et al. [2023a]. A graph is *chordal* if any cycle of length four or more has a chord, which refers to an edge joining two vertices that are not adjacent in the cycle. If a graph $\mathcal{G} = \langle \mathbf{V}, \mathbf{E} \rangle$ is chordal, then its subgraphs are also chordal. A vertex $Z$ in $\mathbf{V}$ is called *simplicial* if $\mathcal{G}[\texttt{Adj}(Z)_{\mathcal{G}}]$ induces a complete graph. As shown by Dirac [1961] and Golumbic [2004], there are at least two non-adjacent simplicial vertices in any non-complete chordal graph with more than one vertex. A *perfect elimination order* of a graph $\mathcal{G}$ is an ordering $\sigma = (V_1, \cdots, V_{|\mathbf{V}|})$ of its vertices, so that each vertex $V_i$ is a simplicial vertex in the subgraph $\mathcal{G} \setminus \{V_1, \cdots, V_{i-1}\}$. It is always possible to transform any circle component in a PAG into a *directed acyclic graph* (DAG) without introducing new unshielded colliders, as the circle component is chordal and every chordal graph has a perfect elimination order [Rose et al., 1976, Habib et al., 2000].

**Orientation rules.** *Fast Causal Inference* (FCI) [Spirtes et al., 2001a] is a causal discovery algorithm for identifying PAGs from conditional independence relationships derived from an observable distribution that follows underlying model. We present the complete orientation rules proposed by Zhang [2008a], omitting rules $\mathcal{R}_5 - \mathcal{R}_7$ due to the absence of selection bias.

$\mathcal{R}_0$ *For each uncovered triple* $\langle X, Z, Y \rangle$ *in* $\mathcal{P}$, *orient it as a collider* $X \ast\!\!\to Z \leftarrow\!\!\ast Y$ *if and only if* $Z$ *is not in* $\mathsf{Sepset}(X, Y)$[8].

$\mathcal{R}_1$ *If* $X \ast\!\!\to Z \circ\!\!-\!\!\ast Y$, *and* $X$ *and* $Y$ *are not adjacent, then orient* $Z \circ\!\!-\!\!\ast Y$ *as* $Z \to Y$.

$\mathcal{R}_2$ *If* $X \to Z \ast\!\!\to Y$ *or* $X \ast\!\!\to Z \to Y$, *and* $X \ast\!\!-\!\!\circ Y$, *then orient* $X \ast\!\!-\!\!\circ Y$ *as* $X \ast\!\!\to Y$.

$\mathcal{R}_3$ *If* $X \ast\!\!\to Z \leftarrow\!\!\ast Y$, $X \ast\!\!-\!\!\circ W \circ\!\!-\!\!\ast Y$, $X$ *and* $Y$ *are not adjacent, and* $W \ast\!\!-\!\!\circ Z$, *then orient* $W \ast\!\!-\!\!\circ Z$ *as* $W \ast\!\!\to Z$.

$\mathcal{R}_4$ *If* $\langle X, \cdots, W, Z, Y \rangle$ *is a discriminating path between* $X$ *and* $Y$ *for* $Z$, *and* $Z \circ\!\!-\!\!\ast Y$; *then if* $Z \in \mathsf{Sepset}(X, Y)$, *orient* $Z \circ\!\!-\!\!\ast Y$ *as* $Z \to Y$; *Otherwise orient the triple* $\langle W, Z, Y \rangle$ *as* $W \leftrightarrow Z \leftrightarrow Y$.

$\mathcal{R}_8$ *If* $X \to Z \to Y$, *and* $X \circ\!\!\to Y$, *orient* $X \circ\!\!\to Y$ *as* $X \to Y$.

$\mathcal{R}_9$ *If* $X \circ\!\!\to Y$, *and* $p = \langle X, Z, W, \cdots, Y \rangle$ *is an uncovered possibly directed path from* $X$ *to* $Y$ *such that* $Z$ *and* $Y$ *are not adjacent, then orient* $X \circ\!\!\to Y$ *as* $X \to Y$.

$\mathcal{R}_{10}$ *Suppose* $X \circ\!\!\to Y$, $Z \to Y \leftarrow W$, $p_1$ *is an uncovered possibly directed path from* $X$ *to* $Z$, *and* $p_2$ *is an uncovered possibly directed path from* $X$ *to* $W$. *Let* $U$ *be the vertex adjacent to* $X$ *on* $p_1$ *(U could be* $Z$*), and* $V$ *be the vertex adjacent to* $X$ *on* $p_2$ *(V could be* $W$*). If* $U$ *and* $V$ *are distinct, and not adjacent, then orient* $X \circ\!\!\to Y$ *as* $X \to Y$.

**Incorporating background knowledge.** Andrews et al. [2020] demonstrated that the ten rules $\mathcal{R}_1 - \mathcal{R}_{10}$ are complete for incorporating *tiered background knowledge*, which refers to background knowledge where the variables in a PAG can be partitioned into distinct groups with an explicit causal order defined among them.

Wang et al. [2022, 2023b] proposed that the rules $\mathcal{R}_1 - \mathcal{R}_3, \mathcal{R}'_4, \mathcal{R}_8 - \mathcal{R}_{10}$ and $\mathcal{R}_{\text{SB}}$ are complete for orienting a PAG when *local background knowledge* (i.e., *all* marks around a vertex) is available. The second additional rule $\mathcal{R}_{\text{SB}}$ naturally follows from the absence of selection bias.[9]

---

[8]A set $\mathbf{Z} \in \mathsf{Sepset}(X, Y)$ if $X$ and $Y$ are independent given $\mathbf{Z}$.

[9]Wang et al. [2024a] proved that rules $\mathcal{R}_1 - \mathcal{R}_{10}$ with one additional rule are sound and complete to incorporate local background knowledge to scenarios where selection bias is present.

$\mathcal{R}_4'$ If $\langle X, \cdots, W, Z, Y \rangle$ is a discriminating path between $X$ and $Y$ for $Z$, and $Z \circ\!\!-\!\!* Y$; then orient $Z \circ\!\!-\!\!* Y$ as $Z \to Y$.

$\mathcal{R}_{\text{SB}}$ If $X \multimap Y$, then orient $X \multimap Y$ as $X \to Y$.

Furthermore, they built the necessary and sufficient conditions for validating local background knowledge (referred to here as *local transformation* in the context of our paper), which can be determined in $\mathcal{O}(|\mathbf{V}|^3)$.

**Theorem 7** (Theorem 3 in Wang et al. [2023b]). *Denote $\mathcal{Q}$ the obtained PMG after some valid local transformations from a PAG $\mathcal{P}$ with orientation rules $\mathcal{R}_1 - \mathcal{R}_3, \mathcal{R}_4', \mathcal{R}_8 - \mathcal{R}_{10}$ and $\mathcal{R}_{SB}$. Given a set $\mathbf{C}_X^Q \subseteq \{V \in \mathtt{Adj}(X)_\mathcal{Q} \mid X \circ\!\!-\!\!* V\}$, there exists a MAG $\mathcal{M}$ consistent to $\mathcal{Q}$ with $X \leftarrow\!\!* V$ for all $V \in \mathbf{C}_X^Q$, and $X \to V$ for all $V \in \{V \in \mathtt{Adj}(X)_\mathcal{Q} \mid X \circ\!\!-\!\!* V\} \setminus \mathbf{C}_X^Q$ if and only if $\mathbf{C}_X^Q$ satisfies the following conditions:*

1. *$\mathtt{PossDe}(X)_{\mathcal{Q} \setminus \mathbf{C}_X^Q} \cap \mathtt{Pa}(\mathbf{C}_X^Q)_\mathcal{Q} = \emptyset$;*

2. *$\mathcal{Q}[\mathbf{C}_X^Q]$ is a complete graph;*

3. *Orient the subgraph $\mathcal{Q}[\mathtt{PossDe}(X)_{\mathcal{Q} \setminus \mathbf{C}_X^Q}]$ as follows until no feasible updates:*
   *For any vertices $V_l$ and $V_j$ such that $V_l \circ\!\!-\!\!\circ V_j$, orient it as $V_l \circ\!\!\to V_j$ if*

   *(i)  $\mathcal{F}_{V_l} \setminus \mathcal{F}_{V_j} \neq \emptyset$, or;*
   *(ii) $\mathcal{F}_{V_l} = \mathcal{F}_{V_j}$ as well as there is a vertex $V_m \in \mathtt{PossDe}(X)_{\mathcal{Q} \setminus \mathbf{C}_X^Q}$ not adjacent to $V_j$ such that $V_m \to V_l \circ\!\!-\!\!\circ V_j$*

   *where $\mathcal{F}_{V_l} = \{V \in \mathbf{C}_X^Q \cup \{X\} \mid V *\!\!-\!\!\circ V_l \in \mathcal{Q}\}$. Then, no new uncovered colliders are introduced.*

The PMG incorporating local transformations satisfies desirable properties as follows.

**Theorem 8** (Theorem 1 in Wang et al. [2023b]). *Let $\mathcal{Q}$ be a PMG obtained from some valid local transformations from a PAG $\mathcal{P}$ and orientation rules $\mathcal{R}_1 - \mathcal{R}_3, \mathcal{R}_4', \mathcal{R}_8 - \mathcal{R}_{10}$ and $\mathcal{R}_{SB}$. Then $\mathcal{Q}$ satisfies the following properties.*

*(Closed). $\mathcal{Q}$ is closed under the orientation rules.*

*(Invariant). The arrowheads ($>$) and tails ($-$) in $\mathcal{Q}$ are invariant in all the MAGs consistent with $\mathcal{Q}$.*

*(Chordal). The circle component in $\mathcal{Q}$ is chordal.*

*(Balanced). For any three nodes $A, B, C$ in $\mathcal{Q}$, if $A *\!\!\to B \circ\!\!-\!\!* C$, then there is an edge between $A$ and $C$ with an arrowhead at $C$, namely, $A *\!\!\to C$. Furthermore, if the edge between $A$ and $B$ is $A \to B$, then the edge between $A$ and $C$ is either $A \to C$ or $A \circ\!\!\to C$ (i.e., it is not $A \leftrightarrow C$).*

*(Complete). For each circle at vertex $A$ on any edge $A \circ\!\!-\!\!* B$ in $\mathcal{Q}$, there exist MAGs $\mathcal{M}_1$ and $\mathcal{M}_2$ consistent with $\mathcal{Q}$ such that $A \leftarrow\!\!* B$ in $\mathcal{M}_1$ and $A \to B$ in $\mathcal{M}_2$.*

Recently, Venkateswaran and Perković [2024], Wang et al. [2024b, 2025a] devised additional rules for more general type of background knowledge. However, the completeness of the orientations in the resulting PMG after applying these rules remains an open problem.

Wang et al. [2023a] leveraged the PMG incorporating local background knowledge to determine whether a given set of variables can be an adjustment set in some MAG consistent with the PMG, and Wang et al. [2024b, 2025b] demonstrated that the additional rules can improve this process.

**Soundness and completeness of orientations.**  To eliminate ambiguity, we provide a formal description of soundness and completeness in the context of orientation within a PMG. Let $\mathcal{Q}$ be a PMG. We say that orientations in $\mathcal{Q}$ are *sound* if there is at least one MAG $\mathcal{M}$ conforming to $\mathcal{Q}$ such

that invariant edge marks in $\mathcal{Q}$ are a subset of edge marks in $\mathcal{M}$. We say that the orientations in $\mathcal{Q}$ are *complete* if for every $A \circ\!\!-\!\!* B$ edge in $\mathcal{H}$, there are two MAGs $\mathcal{M}_1$ and $\mathcal{M}_2$ represented by $\mathcal{Q}$ containing the edges $A \to B$ and $A \leftarrow\!\!* B$, respectively, such that $\mathcal{M}_1$ and $\mathcal{M}_2$ conforming to $\mathcal{Q}$.

**Structural causal bandit.** We review the notion of minimal intervention set (MIS) and possibly optimal minimal intervention set (POMIS) as well as their graphical characterizations for causal diagram by Lee and Bareinboim [2018]. Let $\mathcal{G}$ be a causal diagram and $\mathsf{CC}(X)_{\mathcal{G}}$ be the *c-component* [Tian and Pearl, 2002] of $\mathcal{G}$ that contains $X$ where a c-component is a maximal set of vertices connected with bidirected edges. We denote $\mathsf{CC}(\mathbf{X})_{\mathcal{G}} = \bigcup_{X \in \mathbf{X}} \mathsf{CC}(X)_{\mathcal{G}}$. Let $\mathsf{MUCT}(\mathcal{G}, Y)$ and $\mathsf{IB}(\mathcal{G}, Y)$ be the MUCT and IB given $[\![\mathcal{G}, Y]\!]$, respectively.

**Definition 10** (MIS [Lee and Bareinboim, 2018]). Given information $[\![\mathcal{G}, Y]\!]$, a set of variables $\mathbf{X} \subseteq \mathbf{V} \setminus \{Y\}$ is said to be a *minimal intervention set* (MIS) with respect to $[\![\mathcal{G}, Y]\!]$, denoted by $\mathbb{M}_{\mathcal{G}, Y}$ if there is no $\mathbf{X}' \subsetneq \mathbf{X}$ such that $\mu_{\mathbf{x}[\mathbf{X}']} = \mu_{\mathbf{x}}$ for every SCM conforming to the causal diagram $\mathcal{G}$.

**Proposition 9** (Proposition 1 in Lee and Bareinboim [2018]). *Let $\mathcal{G}$ be a causal diagram over the set of variables $\mathbf{V}$. A set $\mathbf{X} \subseteq \mathbf{V} \setminus \{Y\}$ is an MIS relative to $[\![\mathcal{G}, Y]\!]$ if and only if $\mathbf{X} \subseteq \mathtt{An}(Y)_{\mathcal{G}_{\overline{\mathbf{X}}}}$.*

MIS leverages Rule 3 of do-calculus [Pearl, 1995] to eliminate variables that are irrelevant to the reward. Intuitively, an MIS can be understood as a set $\mathbf{X}$ in which there exists a directed path from any variable $X \in \mathbf{X}$ to $Y$, ensuring that each $X$ can influence $Y$.

**Definition 11** (POMIS [Lee and Bareinboim, 2018]). Let $\mathbf{X} \subseteq \mathbf{V} \setminus \{Y\}$ be an MIS with respect to $[\![\mathcal{G}, Y]\!]$. If there exists an SCM conforming to $\mathcal{G}$ such that $\mu_{\mathbf{x}^*} > \forall_{\mathbf{W} \in \mathbb{M}_{\mathcal{G}, Y} \setminus \{\mathbf{X}\}} \mu_{\mathbf{w}^*}$, then $\mathbf{X}$ is a *possibly-optimal minimal intervention set* (POMIS) with respect to $[\![\mathcal{G}, Y]\!]$.

**Definition 12** (Unobserved-confounders' territory). Given information $[\![\mathcal{G}, Y]\!]$, let $\mathcal{H} = \mathcal{G}[\mathtt{An}(Y)_{\mathcal{G}}]$. A set of variables $\mathbf{T} \subseteq \mathbf{V}(\mathcal{H})$ containing $Y$ is called a *UC-territory* on $\mathcal{G}$ with respect to $Y$ if $\mathtt{De}(\mathbf{T})_{\mathcal{H}} = \mathbf{T}$ and $\mathsf{CC}(\mathbf{T})_{\mathcal{H}} = \mathbf{T}$. A UC-territory $\mathbf{T}$ is said to be *minimal* if no $\mathbf{T}' \subsetneq \mathbf{T}$ is a UC-territory (MUCT).

**Definition 13** (Interventional border). Let $\mathbf{T}$ be a minimal UC-territory on causal diagram $\mathcal{G}$ with respect to $Y$. Then $\mathbf{W} = \mathtt{Pa}(\mathbf{T})_{\mathcal{G}} \setminus \mathbf{T}$ is called an *interventional border* (IB) for $\mathcal{G}$ with respect to $Y$.

When given a causal diagram $\mathcal{G}$, MUCT and IB provide a graphical characterization of POMIS. In words, MUCT is the minimal set of variables that is closed under descendants and connected by a bidirected edge; and IB consists of the parents of MUCT, excluding MUCT itself. Intuitively, MUCT is the minimal closed mechanism that conveys all hidden information from unobserved confounders to the downstream reward, while IB consists of the nodes that directly affect this closed mechanism.

**Theorem 9** (Theorem 6 in Lee and Bareinboim [2018]). *Let $\mathcal{G}$ be a causal diagram over the set of variables $\mathbf{V}$. A set $\mathbf{X} \subseteq \mathbf{V} \setminus \{Y\}$ is a POMIS if and only if it holds $\mathsf{IB}(\mathcal{G}_{\overline{\mathbf{X}}}, Y) = \mathbf{X}$.*

## B.2 Background Results

We present useful results established in existing works.

### B.2.1 Background Results in Zhang [2006, 2008a]

**Lemma 3** (Lemma 0, as used in the proof of Lemma 5.1.7 in Zhang [2006]). *Let $X$ and $Y$ be distinct nodes in a MAG $\mathcal{M}$. If $p = \langle X, \cdots, Z, V, Y \rangle$ is a discriminating path from $X$ to $Y$ for $V$ in a MAG $\mathcal{M}$, and the corresponding subpath between $X$ and $V$ in $\mathcal{P}$ is (also) a collider path, then the path corresponding to $p$ in $\mathcal{Q}$ is also a discriminating path for $V$.*

**Lemma 4** (Lemma A.1 in Zhang [2008a] & Lemma 5 in Jaber et al. [2018]). *Let $\mathcal{P}$ be a PAG over $\mathbf{V}$, and let $\mathcal{P}[\mathbf{A}]$ be the subgraph of $\mathcal{P}$ induced by $\mathbf{A} \subseteq \mathbf{V}$. For any three nodes $A, B, C$, if $A *\!\!\to B \circ\!\!-\!\!* C$, then there is an edge between $A$ and $C$ with an arrowhead at $C$, namely, $A *\!\!\to C$. Furthermore, if the edge between $A$ and $B$ is $A \to B$, then the edge between $A$ and $C$ is either $A \to C$ or $A \circ\!\!\to C$ (i.e., it is not $A \leftrightarrow C$).*

**Lemma 5** (Lemma 3.3.2 in Zhang [2006]). *In a PAG $\mathcal{P}$, for any two nodes $A$ and $B$, if there is a circle path, then following holds:*

1. *If there is an edge between $A$ and $B$, the edge is not into $A$ or $B$;*

2. *For any other node $C$, $C \ast\!\!\rightarrow A$ if and only if $C \ast\!\!\rightarrow B$. Furthermore, $C \leftrightarrow A$ if and only if $C \leftrightarrow B$.*

**Lemma 6** (Theorem 2 in Zhang [2008a])**.** *Let $\mathcal{P}$ be a PAG. Let $\mathcal{M}$ be the graph resulting from the following procedure applied to a $\mathcal{P}$.*

    *Step 1.* *Replace all partially directed edges ($\circ\!\!\rightarrow$) in $\mathcal{P}$ with directed edges ($\rightarrow$).*

    *Step 2.* *Orient the circle component of $\mathcal{P}$ into a DAG with no unshielded colliders.*

*Then, the result graph $\mathcal{M}$ conforms to $\mathcal{P}$.*

**Lemma 7** (Lemma B.1 in Zhang [2008a])**.** *Let $A$ and $B$ be two distinct nodes in a PAG $\mathcal{P}$. If $p$ is a possibly directed path from $A$ to $B$ in a PAG $\mathcal{P}$, then some subsequence of $p$ forms an uncovered possibly directed path from $A$ to $B$ in $\mathcal{P}$.*

**Lemma 8** (Lemma B.2 in Zhang [2008a])**.** *Let $A$ and $B$ be two distinct nodes in a PAG $\mathcal{P}$. If $p = \langle V_0(= A), \cdots V_n(= B)\rangle, n \geq 2$, is an uncovered possibly directed path from $A$ to $B$ in $\mathcal{P}$, and $V_{i-1} \ast\!\!\rightarrow V_i$ for some $i \in \{1, \cdots, n\}$, then $V_{j-1} \rightarrow V_j$ for all $j \in \{i+1, \cdots, n\}$.*

**Lemma 9** (Lemma B.4 in Zhang [2008a])**.** *In a PAG $\mathcal{P}$, if there is a possibly directed path from $A$ to $B$, then the edge between $A$ and $B$, if any, is not into $A$.*

**Lemma 10** (Lemma B.5 in Zhang [2008a])**.** *In a PAG $\mathcal{P}$, let $A$ and $B$ be two distinct nodes in a PAG $\mathcal{P}$. If there is a possibly directed path from $A$ to $B$ that is into $B$, then every uncovered possibly directed path from $A$ to $B$ is into $B$.*

**Lemma 11** (Lemma B.7 in Zhang [2008a])**.** *In a PAG $\mathcal{P}$, if there is a circle path between two adjacent vertices in $\mathcal{P}$, then the edge between the two vertices is a circle edge ($\circ\!\!-\!\!\circ$).*

### B.2.2   Background Results in Maathuis and Colombo [2015]

**Lemma 12** (Lemma 7.6 in Maathuis and Colombo [2015])**.** *Let $\mathcal{P}$ be a PAG with $k$ edges into $X$, $k \geq 0$. Then there exists at least one MAG $\mathcal{M}$ in the Markov equivalence class represented by $\mathcal{P}$ that has $k$ edges into $X$.*

### B.2.3   Background Results in Perkovic et al. [2018]

**Lemma 13** (Lemma 48 in Perkovic et al. [2018])**.** *Let $X$ be a node in a PAG $\mathcal{P}$. Let $\mathcal{M}$ be a MAG conforming $\mathcal{P}$ that satisfies Lem. 6. Then any edge that is either $X \circ\!\!-\!\!\circ Y$, $X \circ\!\!\rightarrow Y$, or invisible $X \rightarrow Y$ in $\mathcal{P}$ is invisible $X \rightarrow Y$ in $\mathcal{M}$.*

### B.2.4   Background Results in Jaber et al. [2018, 2022]

**Lemma 14** (Proposition 1 in Jaber et al. [2018])**.** *Let $\mathcal{P}$ be a PAG over $\mathbf{V}$, and $\mathcal{G}$ be any causal diagram in the equivalence class represented by $\mathcal{P}$. Let $X \neq Y$ be two nodes in $\mathbf{A} \subseteq \mathbf{V}$. If $X$ is an ancestor of $Y$ in $\mathcal{G}[\mathbf{A}]$, then $X$ is a possible ancestor of $Y$ in $\mathcal{P}[\mathbf{A}]$.*

**Lemma 15** (Lemma 4 in Jaber et al. [2018])**.** *Let $\mathcal{P}$ be a PAG over $\mathbf{V}$. For every directed edge $X \rightarrow Y$ in induced subgraph $\mathcal{P}[\mathbf{A}]$ with $\mathbf{A} \subseteq \mathbf{V}$, if it is visible in $\mathcal{P}$, then it is also visible in $\mathcal{P}[\mathbf{A}]$.*

**Lemma 16** (Proposition 2 in Jaber et al. [2018])**.** *Let $\mathcal{P}$ be a PAG over $\mathbf{V}$, and $\mathcal{G}$ be any causal diagram in the equivalence class represented by $\mathcal{P}$. Let $X \neq Y$ be two nodes in $\mathbf{A} \subseteq \mathbf{V}$. If $X$ and $Y$ are in the same c-component in $\mathcal{G}[\mathbf{A}]$, then $X$ and $Y$ are in the same pc-component in $\mathcal{P}[\mathbf{A}]$.*

**Algorithm 2:** Partial Topological Order PTO [Jaber et al., 2018]

**Input:** $\mathcal{P}$, $\mathbf{A} \subseteq \mathbf{V}(\mathcal{P})$
**Output:** Partial Topological Order over $\mathcal{P}[\mathbf{A}]$

1 **while** there exists a bucket $\mathbf{B}$ in $\mathcal{P}[\mathbf{A}]$ with only arrowheads incident on it **do**
2      Extract $\mathbf{B}$ from $\mathcal{P}[\mathbf{A}]$
3      $\mathbf{A} \leftarrow \mathbf{A} \setminus \mathbf{B}$
4 **end**
5 The partial order is $\mathbf{B}^1 \prec \cdots \prec \mathbf{B}^m$ in reverse order of the bucket extraction, i.e., $\mathbf{B}^1$ is the last bucket extracted and $\mathbf{B}^m$ is the first.

**Lemma 17** (Proposition 4 in Jaber et al. [2018]). *Let $\mathcal{P}$ be a PAG over $\mathbf{V}$, and let $\mathcal{P}[\mathbf{A}]$ be the subgraph of $\mathcal{P}$ induced by $\mathbf{A} \subseteq \mathbf{V}$. Then, Alg. 2 is sound over $\mathcal{P}[\mathbf{A}]$, in the sense that the partial order is valid with respect to $\mathcal{G}[\mathbf{A}]$, for every causal diagram $\mathcal{G}$ in the equivalence class represented by $\mathcal{P}$.*[10]

**Lemma 18** (Lemma 6 in Jaber et al. [2018]). *In $\mathcal{M}[\mathbf{A}]$, where $\mathcal{M}$ is a MAG over $\mathbf{V}$ and $\mathbf{A} \subseteq \mathbf{V}$, the following property holds:*

> *For any three vertices $A, B, C$, if $A \ast\!\!\rightarrow B \rightarrow C$ and both edges are invisible, then we have $A \ast\!\!\rightarrow C$ and the edge is invisible.*

**Lemma 19** (Lemma 18 in Jaber et al. [2022]). *Let $\mathcal{P}$ be a PAG over $\mathbf{V}$, and let $\mathcal{P}[\mathbf{A}]$ be the subgraph of $\mathcal{P}$ induced by $\mathbf{A} \subseteq \mathbf{V}$. In $\mathcal{P}[\mathbf{A}]$, the following property holds:*

> *For any three vertices $A, B, C$, if $A \ast\!\!\rightarrow B \;^{\circ}\!\!\!\rightarrow C$ and both edges are invisible, then we have $A \ast\!\!\rightarrow C$ and the edge is invisible.*

### B.2.5 Background Results in Wang et al. [2023b, 2024a]

**Lemma 20** (Lemma 2 in Wang et al. [2023b]). *Let $\mathcal{Q}$ be a PMG obtained from some valid local transformations from a PAG $\mathcal{P}$ and the orientation rules. If $p$ is a possibly directed path from $A$ to $B$ in $\mathcal{Q}$, then some subsequence of $p$ is an uncovered possibly directed path from $A$ to $B$ in $\mathcal{Q}$.*

**Lemma 21** (Lemma 3 in Wang et al. [2023b]). *Let $\mathcal{Q}$ be a PMG obtained from some valid local transformations from a PAG $\mathcal{P}$ and the orientation rules. In a PMG $\mathcal{Q}$, for any two nodes $A$ and $B$, if there is a circle path, then following holds:*

1. *If there is an edge between $A$ and $B$, the edge is not into $A$ or $B$;*

2. *For any other node $C$, $C \ast\!\!\rightarrow A$ if and only if $C \ast\!\!\rightarrow B$. Furthermore, $C \leftrightarrow A$ if and only if $C \leftrightarrow B$.*

**Lemma 22** (Lemma 4 in Wang et al. [2023b]). *Let $\mathcal{Q}$ be a PMG obtained from some valid local transformations from a PAG $\mathcal{P}$ and the orientation rules. Suppose a MAG $\mathcal{M}$ consistent to $\mathcal{Q}$ and the local transformation $\mathbf{C}_X^{\mathcal{Q}}$. Then $Y \in \texttt{PossDe}(X)_{\mathcal{Q} \setminus \mathbf{C}_X^{\mathcal{Q}}}$ if and only if $Y \in \texttt{De}(X)_{\mathcal{M}}$.*

**Lemma 23** (Lemma 16.1 in Wang et al. [2023b]). *Let $\mathcal{Q}$ be a PMG obtained from some valid local transformations from a PAG $\mathcal{P}$ and the orientation rules. The MAG oriented according to Lem. 6 conforms to $\mathcal{Q}$.*

**Lemma 24** (Lemma 2 in Wang et al. [2024a]). *Let $\mathcal{Q}$ be a PMG obtained from some valid local transformations from a PAG $\mathcal{P}$ and the orientation rules. If there is an uncovered circle path $p = \langle V_1, V_2, \cdots, V_n \rangle, n \geq 3$ in $\mathcal{Q}$, then any two non-consecutive vertices are not adjacent (minimal circle path).*

---

[10]A *bucket* refers to the closure of nodes connected with circle paths.

| | Total trials | Task 1 10k | Task 2 10k | Task 3 5k | Task 4 * 10k | Task 5 * 2k | Task 6 * 2k |
|---|---|---|---|---|---|---|---|
| **TS** | POMIS | **123.4** ± 52.2(39.3%) | **320.9** ± 43.7(44.3%) | **60.3** ± 3.9(2.5%) | **85.9** ± 43.1(9.7%) | **51.9** ± 2.3(14.5%) | **203.3** ± 3.9(23.8%) |
| | DMIS | 144.9 ± 51.9 | 661.1 ± 50.6 | 1719.6 ± 23.1 | 335.9 ± 48.3 | 108.2 ± 4.7 | 805.6 ± 20.2 |
| | BF | 314.0 ± 54.1 | 724.8 ± 50.3 | 2421.3 ± 35.5 | 889.1 ± 59.5 | 357.9 ± 16.2 | 854.5 ± 20.3 |
| **KL-UCB** | POMIS | **243.4** ± 55.5(48.9%) | **629.9** ± 45.1(50.4%) | **52.0** ± 0.1(2.1%) | **195.1** ± 45.7(12.9%) | **54.0** ± 0.2(12.7%) | **202.9** ± 0.3(17.9%) |
| | DMIS | 275.9 ± 54.9 | 1175.3 ± 52.3 | 1905.6 ± 9.9 | 705.5 ± 55.6 | 123.4 ± 1.7 | 1043.3 ± 13.7 |
| | BF | 497.9 ± 55.7 | 1250.9 ± 60.5 | 2463.6 ± 33.7 | 1518.4 ± 71.0 | 431.1 ± 16.0 | 1130.7 ± 13.9 |

Table 1: Mean and standard deviation of cumulative regret (CR). The asterisk ($*$) indicates additional experiments. The percentages (red) represent the ratio $\frac{\text{CR for POMIS}}{\text{CR for BF}} \times 100(\%)$.

## C  Assumptions

In this paper, we assume that there is *no selection bias* in the SCM-MAB system; that is, the PAG representing our causal diagrams of interest contains no undirected edges. Since our work focuses on a graphical perspective of the structured bandit system in terms of PAGs, we assume access to the *true* PAG representing the causal diagram corresponding to the target bandit instance.

## D  Experimental Details and Additional Results

This section provides details on the specific SCMs used in all bandit instances presented in the experiments (Sec. 5) and additional experiments. Simulations are repeated 1,000 times to obtain consistent results. The simulations were conducted on a Linux server equipped with an Intel Xeon Gold 5317 processor running at 3.0 GHz and 64 GB of RAM. No GPUs were used during the simulations.

We consider three strategies for selecting arms: POMISs, DMISs, and Brute-force (BF), combined with two prominent MAB solvers: Thompson Sampling (TS) [Thompson, 1933, Chapelle and Li, 2011, Agrawal and Goyal, 2012, Kaufmann et al., 2012] and KL-UCB [Garivier and Cappé, 2011, Cappé et al., 2013]. In the Brute-force strategy, all possible combinations of arms $\bigcup_{\mathbf{X} \subseteq \mathbf{V} \setminus \{Y\}} \mathfrak{X}_{\mathbf{X}}$ are evaluated. The number of trials is set to 10,000 for Tasks 1, 2, and 4; 5,000 for Task 3; and 2,000 for Tasks 5 and 6, which is sufficient to observe performance differences among action spaces. The number of trials is selected such that the cumulative regret with respect to POMIS stabilizes across 1000 repeated runs. Our experimental setup closely follows those of Lee and Bareinboim [2018] and Wei et al. [2023]. Tables 1 and 2 summarize our simulation results.

These results demonstrate that refining arms by considering the Markov equivalence class into account enhances the efficiency of agents when interacting with the underlying environment.

### Details of the Causal Models for Bandit Instances

We denote the exclusive-or operation by $\oplus$, and use `Bern` to represent a Bernoulli distribution. We *randomly* generate structural functions $\mathbf{F}$ using binary logical operations ($\wedge, \vee, \oplus, \neg$), and the parameters of the exogenous variable distributions are also *randomly* selected.

**Task 1.**  The bandit instance is associated with an SCM $\mathcal{S}_1$ where

$$
\mathcal{S}_1 = \begin{cases} \mathbf{U} & = \{U_A, U_B, U_C, U_D, U_Y, U_{BY}, U_{AC}\} \\ \mathbf{V} & = \{A, B, C, D, Y\} \\ \mathbf{F} & = \begin{cases} f_A = u_A \wedge u_{AC}, f_B = c \vee ((1 - u_B) \wedge u_{BY}), \\ f_C = a \vee ((1 - u_C) \wedge u_{AC}), f_D = y \wedge u_D, \\ f_Y = \{(1 - b) \vee \{(1 - c) \wedge (1 - u_{BY})\}\} \wedge u_Y \end{cases} \\ P(\mathbf{U}) & = \begin{cases} U_A \sim \texttt{Bern}(0.44), U_B \sim \texttt{Bern}(0.7), U_C \sim \texttt{Bern}(0.4), \\ U_D \sim \texttt{Bern}(0.59), U_Y \sim \texttt{Bern}(0.66), U_{BY} \sim \texttt{Bern}(0.28), \\ U_{AC} \sim \texttt{Bern}(0.77). \end{cases} \end{cases}
\tag{1}
$$

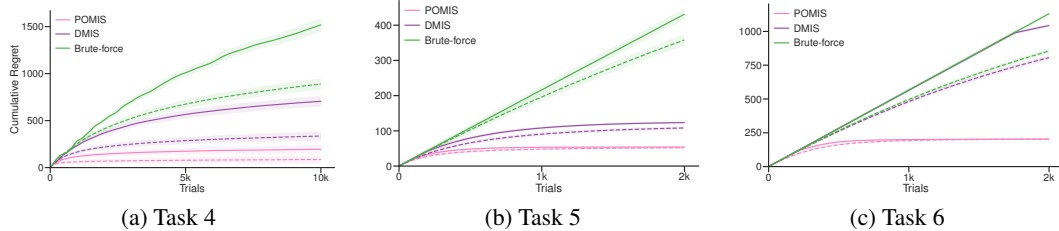

| (a) Task 4 | (b) Task 5 | (c) Task 6 |

Figure 11: Cumulative regrets for the corresponding KL-UCB (solid) and TS (dashed) for additional experiments (Task 4–6) under distinct strategies. We plot the average cumulative regrets along with their standard deviations.

**Task 2.** The bandit instance is associated with an SCM $\mathcal{S}_2$ where

$$
\mathcal{S}_2 = \begin{cases}
\mathbf{U} & = \{U_A, U_B, U_C, U_D, U_E, U_Y, U_{AY}, U_{BY}\} \\
\mathbf{V} & = \{A, B, C, D, E, Y\} \\
\mathbf{F} & = \begin{cases}
f_A = u_A \oplus u_{AY}, f_B = u_b \oplus u_{BY}, f_C = u_C \oplus b, \\
f_D = u_D \oplus c, f_E = \{(1 - u_E) \oplus (1 - a)\} \wedge c, \\
f_Y = u_Y \vee \{\{(1 - e) \oplus (u_{AY} \wedge u_{BY})\} \wedge d\}
\end{cases} \\
P(\mathbf{U}) & = \begin{cases}
U_A \sim \texttt{Bern}(0.47), U_B \sim \texttt{Bern}(0.59), U_C \sim \texttt{Bern}(0.37), \\
U_D \sim \texttt{Bern}(0.61), U_E \sim \texttt{Bern}(0.55), U_Y \sim \texttt{Bern}(0.21), \\
U_{AY} \sim \texttt{Bern}(0.54), U_{BY} \sim \texttt{Bern}(0.36).
\end{cases}
\end{cases}
\tag{2}
$$

**Task 3.** The bandit instance is associated with an SCM $\mathcal{S}_3$ where

$$
\mathcal{S}_3 = \begin{cases}
\mathbf{U} & = \{U_A, U_B, U_C, U_D, U_E, U_F, U_G, U_H, U_I, U_Y, U_{AB}, U_{BE}, U_{FG}, U_{HI}, U_{IY}\} \\
\mathbf{V} & = \{A, B, C, D, E, F, G, H, I, Y\} \\
\mathbf{F} & = \begin{cases}
f_A = u_A \oplus U_{AB}, f_B = (u_{BE} \oplus u_{AB}) \wedge (a \vee u_b), \\
f_C = u_C \oplus b, f_D = u_D \oplus c, \\
f_E = ((1 - u_E) \oplus (1 - d)) \wedge u_{BE}, \\
f_F = ((1 - u_F) \oplus (1 - e)) \wedge u_{FG}, \\
f_G = (u_{FG} \oplus h) \wedge (e \vee u_G), \\
f_H = ((1 - u_H) \oplus (1 - e)) \wedge u_{HI}, \\
f_I = (u_{IY} \oplus u_{HI}) \wedge (f \vee u_I), \\
f_Y = u_Y \vee (((1 - h) \oplus (i \wedge u_{IY})) \wedge f)
\end{cases} \\
P(\mathbf{U}) & = \begin{cases}
U_A \sim \texttt{Bern}(0.47), U_B \sim \texttt{Bern}(0.59), U_C \sim \texttt{Bern}(0.37), \\
U_D \sim \texttt{Bern}(0.61), U_E \sim \texttt{Bern}(0.55), U_F \sim \texttt{Bern}(0.21), \\
U_G \sim \texttt{Bern}(0.54), U_H \sim \texttt{Bern}(0.36), U_I \sim \texttt{Bern}(0.45), \\
U_Y \sim \texttt{Bern}(0.37), U_{AB} \sim \texttt{Bern}(0.29), U_{BE} \sim \texttt{Bern}(0.53), \\
U_{FG} \sim \texttt{Bern}(0.62), U_{HI} \sim \texttt{Bern}(0.46), U_{IY} \sim \texttt{Bern}(0.67).
\end{cases}
\end{cases}
\tag{3}
$$

As an additional experiment, we evaluate the cumulative regrets (CR) of SCM-MAB using the PAGs illustrated in Fig. 12. The corresponding plots are shown in Fig. 11.

**Task 4.** We consider the PAG in Fig. 12a to validate our result. Using three strategies, the POMIS-based TS and KL-UCB achieve CRs of 85.9 and 195.1, which correspond to **9.7%** and **12.9%**,

| | | Task 1 | Task 2 | Task 3 | Task 4* | Task 5* | Task 6* |
|---|---|---|---|---|---|---|---|
| | POMIS | **7** (43.75%) | **18** (56.3%) | **8** (1.56%) | **6** (37.5%) | **16** (12.5%) | **40** (31.3%) |
| IS | DMIS | 9 | 30 | 152 | 14 | 32 | 120 |
| | BF | 16 | 32 | 512 | 16 | 128 | 128 |
| | POMIS | **19** (23.5%) | **89** (36.6%) | **54** (0.27%) | **15** (6.17%) | **81** (3.70%) | **231** (10.7%) |
| Arms | DMIS | 25 | 195 | 2025 | 57 | 189 | 1755 |
| | BF | 81 | 243 | 19683 | 243 | 2187 | 2187 |

Table 2: For each task, the number of intervention sets (IS; shown above) and the corresponding number of arms (shown below) are reported. The percentages (red) indicate the ratio $\frac{\# \text{POMIS}}{\# \text{BF}} \times 100(\%)$, and the corresponding ratio for the number of arms.

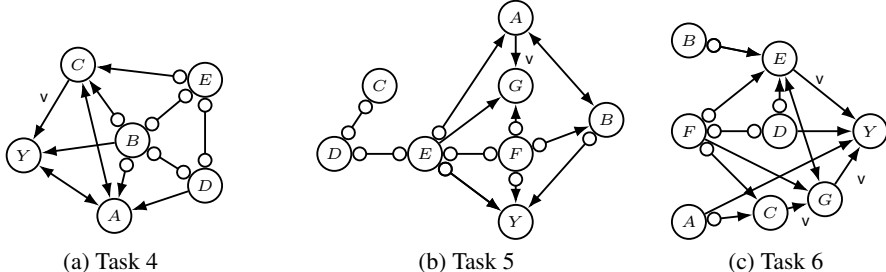

(a) Task 4          (b) Task 5          (c) Task 6

Figure 12: Each PAG represents a target bandit mechanism that the deployment agent interacts with.

respectively, of CR for BF. The bandit instance is associated with an SCM $\mathcal{S}_4$ where

$$
\mathcal{S}_4 = \begin{cases}
\mathbf{U} & = \{U_A, U_B, U_C, U_D, U_E, U_Y, U_{AC}, U_{AY}, U_{BY}\} \\
\mathbf{V} & = \{A, B, C, D, E, Y\} \\
\mathbf{F} & = \begin{cases}
f_A = u_A \vee \{\{(1-d) \oplus (u_{AY} \wedge u_{AC})\} \wedge b\}, \\
f_B = u_B \oplus u_{BY}, \\
f_C = (u_{AC} \oplus E) \wedge (B \vee u_C), f_D = u_D \oplus b, \\
f_E = \{(1 - u_E) \oplus (1 - b)\} \wedge d, \\
f_Y = u_Y \vee \{\{(1-c) \oplus (u_{AY} \wedge u_{BY})\} \wedge b
\end{cases} \\
P(\mathbf{U}) & = \begin{cases}
U_A \sim \texttt{Bern}(0.47), \{U_B \sim \texttt{Bern}(0.59), U_C \sim \texttt{Bern}(0.37), \\
U_D \sim \texttt{Bern}(0.61), U_E \sim \texttt{Bern}(0.55), U_Y \sim \texttt{Bern}(0.21), \\
U_{AC} \sim \texttt{Bern}(0.36), U_{AY} \sim \texttt{Bern}(0.54), U_{BY} \sim \texttt{Bern}(0.45).
\end{cases}
\end{cases}
\tag{4}
$$

**Task 5.** We consider the PAG in Fig. 12b to validate our result. Using three strategies, the POMIS-based TS and KL-UCB achieve CRs of 51.9 and 54.1, which correspond to **14.5%** and **12.7%**, respectively, of CR for BF. The bandit instance is associated with an SCM $\mathcal{S}_5$ where

$$
\mathcal{S}_5 = \begin{cases}
\mathbf{U} & = \{U_A, U_B, U_C, U_D, U_E, U_F, U_G, U_Y, U_{AB}, U_{BY}, U_{CD}, U_{FG}\} \\
\mathbf{V} & = \{A, B, C, D, E, F, G, Y\} \\
\mathbf{F} & = \begin{cases}
f_A = (1 - u_A) \vee \{(1 - u_{AB}) \oplus e\}, \\
f_B = \{1 - \{u_{BY} \vee (1 - u_B)\}\} \oplus (u_{AB} \oplus f), \\
f_C = u_{CD} \oplus u_C, f_D = (1 - u_D) \vee \{(1 - u_{CD}) \oplus c\}, \\
f_E = d \oplus u_E, f_F = (1 - u_F) \vee \{(1 - u_{FG}) \oplus e\}, \\
f_G = \{1 - \{u_{FG} \vee (1 - u_G)\}\} \oplus (e \oplus a), \\
f_Y = f \oplus \{\{(1 - u_{BY}) \oplus u_Y\}\} \oplus (b \oplus a)
\end{cases} \\
P(\mathbf{U}) & = \begin{cases}
U_A \sim \texttt{Bern}(0.29), U_B \sim \texttt{Bern}(0.73), U_C \sim \texttt{Bern}(0.36), \\
U_D \sim \texttt{Bern}(0.45), U_E \sim \texttt{Bern}(0.38), U_F \sim \texttt{Bern}(0.58), \\
U_G \sim \texttt{Bern}(0.55), U_Y \sim \texttt{Bern}(0.57), U_{FG} \sim \texttt{Bern}(0.36), \\
U_{AB} \sim \texttt{Bern}(0.37), U_{BY} \sim \texttt{Bern}(0.35), U_{CD} \sim \texttt{Bern}(0.4).
\end{cases}
\end{cases}
\tag{5}
$$

**Task 6.** We consider the PAG in Fig. 12c to validate our result. Using three strategies, the POMIS-based TS and KL-UCB achieve CRs of 203.3 and 202.9, which correspond to **23.8%** and **17.9%**, respectively, of CR for BF. The bandit instance is associated with an SCM $\mathcal{S}_6$ where

$$
\mathcal{S}_6 = \begin{cases}
\mathbf{U} & = \{U_A, U_B, U_C, U_D, U_E, U_F, U_G, U_Y, U_{AY}, U_{EF}, U_{FG}\} \\
\mathbf{V} & = \{A, B, C, D, E, F, G, Y\} \\
\mathbf{F} & = \begin{cases}
f_A = u_A \oplus u_{AY}, f_B = u_B, \\
f_C = ((1 - u_C) \wedge f) \vee a, f_D = f \wedge u_D, \\
f_E = ((1 - b) \vee ((1 - u_{EF}) \wedge (1 - d))) \wedge u_E, \\
f_F = ((1 - u_F) \wedge u_{FG}) \vee u_{EF}, \\
f_G = ((1 - u_G) \wedge u_{FG}) \vee g, \\
f_Y = e \wedge ((1 - d) \vee (((1 - u_Y) \wedge g) \vee u_{AY}))
\end{cases} \\
P(\mathbf{U}) & = \begin{cases}
U_A \sim \texttt{Bern}(0.59), U_B \sim \texttt{Bern}(0.42), U_C \sim \texttt{Bern}(0.77), \\
U_D \sim \texttt{Bern}(0.28), U_E \sim \texttt{Bern}(0.72), U_F \sim \texttt{Bern}(0.48), \\
U_G \sim \texttt{Bern}(0.68), U_Y \sim \texttt{Bern}(0.51), U_{AY} \sim \texttt{Bern}(0.61), \\
U_{EF} \sim \texttt{Bern}(0.63), U_{FG} \sim \texttt{Bern}(0.55).
\end{cases}
\end{cases} \tag{6}
$$

# E   Discussions

In this section, we discuss circle mark transformations from the perspective of orientation completeness and complexity of enumerating all POMISs for PAGs.

## E.1   Partial Mixed Graphs Obtained from Local Transformation

Let $\widetilde{\mathcal{Q}_{\mathbf{X}}}$ be a PMG that satisfies (1) the two conditions in Prop. 7 and (2) is closed under orientation rules $\mathcal{R}_1 - \mathcal{R}_3, \widetilde{\mathcal{R}}_4, \mathcal{R}_8 - \mathcal{R}_{10}$, and $\mathcal{R}_{\text{SB}}$ with additional Rules provided by Wang et al. [2024b], Venkateswaran and Perković [2024], Wang et al. [2025a]. It is important to note that the completeness of $\widetilde{\mathcal{Q}_{\mathbf{X}}}$ remains an open problem. Therefore, $\widetilde{\mathcal{Q}_{\mathbf{X}}}$ is inadequate to completely characterize POMIS for PAGs.

**Remark 2.** *Every $\mathcal{Q}_{\mathbf{X}}^i$ is complete for orientations; for any $A \circ\!\!-\!\!* B$ in $\mathcal{Q}_{\mathbf{X}}^i$, there are two MAGs $\mathcal{M}_1$ and $\mathcal{M}_2$ represented by $\mathcal{Q}_{\mathbf{X}}^i$ containing $A \to B$ and $A \leftarrow\!\!* B$ respectively.*

Moreover, even though we have access to $\mathcal{Q}_{\mathbf{X}}^*$—a PMG that satisfies (1) the two conditions in Prop. 7 and (2) the orientation completeness—$\mathcal{Q}_{\mathbf{X}}^*$ is still insufficient to ensure $\mathbf{X} \subseteq \texttt{An}(Y)_{\mathcal{Q}_{\mathbf{X}}^*}$. To witness, consider a PAG $\mathcal{P}$ in Fig. 5 with $\mathbf{X} = \{A\}$.

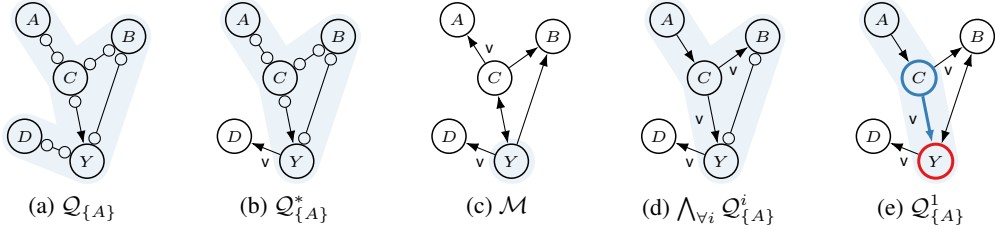

|  |  |  |  |  |
|---|---|---|---|---|
| (a) $\mathcal{Q}_{\{A\}}$ | (b) $\mathcal{Q}_{\{A\}}^*$ | (c) $\mathcal{M}$ | (d) $\bigwedge_{\forall i} \mathcal{Q}_{\{A\}}^i$ | (e) $\mathcal{Q}_{\{A\}}^1$ |

Figure 13: The light blue region indicates possible ancestors of $Y$. (a) PMG incorporating necessary conditions (Prop. 7) and (b) the PMG with orientation completeness. (c) MAG represented by $\mathcal{Q}_{\{A\}}$ while $A \notin \texttt{An}(Y)_{\mathcal{M}}$. (d) PMG representing sound and complete orientations over MAGs satisfying that $\{A\}$ is an MIS. (e) PMG with $\mathbf{C}_{\{A\}}^{\mathcal{Q}_{\{A\}}} = \emptyset$ and $\mathbf{C}_{\{Y\}}^{\mathcal{Q}_{\{A\}}} = \{B\}$.

Then $C \circ\!\!-\!\!\circ Y$ in $\mathcal{P}$ corresponds to $C \circ\!\!\to Y$ in $\mathcal{Q}_{\mathbf{X}}^*$, according to the first condition in Prop. 7 supported by the uncovered proper possibly directed path $A \circ\!\!-\!\!\circ C \circ\!\!-\!\!\circ Y$. Moreover, $Y \to D$ is oriented by $\mathcal{R}_1$, and all remaining circle marks can vary across the underlying MAGs represented by $\mathcal{Q}_{\mathbf{X}}^*$. Here, we can find a MAG $\mathcal{M}$ where $\mathbf{X} \notin \texttt{An}(Y)_{\mathcal{M}}$ by orienting $C \circ\!\!\to Y$ as $C \leftrightarrow Y$, suggesting that additional information (orientation) is necessary.

Furthermore, neither $\mathcal{Q}_{\mathbf{X}}^*$ nor $\widetilde{\mathcal{Q}_{\mathbf{X}}}$ guarantees the *balanced property* (Lems 4 and 31). To witness, refer to $\mathcal{Q}_{\{C\}}^*$ (identical to $\widetilde{\mathcal{Q}_{\{C\}}}$). We can observe that there is $C \to Y \circ\!\!-\!\!\circ B$ while $C \circ\!\!-\!\!\circ B$, which violates the balanced property.

One might surmise that $\mathbf{X} = \mathsf{IB}(\mathcal{P}, Y, \mathbf{X})$ is an appropriate characterization of POMIS for PAGs. However, this approach does *not* hold. For illustration, consider the PAG $\mathcal{P}$ in Fig. 5a and a set $\mathbf{X} = \{A\}$, which is a DMIS with respect to $[\![\mathcal{P}, Y]\!]$. Moreover, we can simply derive $\mathsf{IB}(\mathcal{P}, Y, \mathbf{X}) = \{A\}$, and thus $\mathbf{X} = \mathsf{IB}(\mathcal{P}, Y, \mathbf{X})$ holds. For $\mathbf{X}$ to be an MIS for a MAG $\mathcal{M}$ represented by $\mathcal{P}$, the edge $A \circ\!\!-\!\!\circ C$ should correspond to $A \to C$ in $\mathcal{M}$, implying the visible edges $C \to B$ and $C \to Y$, as these are non-definite colliders (see $\mathcal{M}_1$ and $\mathcal{M}_2$ in Figs. 5b and 5c with Fig. 13d). Regardless of the edge orientation of $B \circ\!\!-\!\!\circ Y$, we find $\mathsf{IB}(\mathcal{M}, Y, \mathbf{X}) = \{C\}$, as in Fig. 13e. Thus, $\mathbf{X} = \{A\}$ is not a POMIS with respect to $[\![\mathcal{M}, Y]\!]$ for all $\mathcal{M} \in [\mathcal{P}]$. Therefore, $\mathsf{IB}(\mathcal{P}, Y, \mathbf{X})$ fails to characterize POMIS.

### E.2 Adaptive Learning: Simultaneous Discovery and Regret Minimization

A natural question is why we do not pursue adaptive discovery from online information. We address this point, beginning with relevant literature on *causal discovery with interventions*.

**Offline discovery from interventions.** In offline or non-adaptive setting, interventions are predetermined before algorithm execution. Hauser and Bühlmann [2012] studied the problem of learning graph structures from interventions under the assumption of no unobserved confounders, while Kocaoglu et al. [2017] explored experimental design for learning causal diagrams from interventions. Recently, Zhou et al. [2025] investigated learning PAGs from interventions.

**Online discovery from interventions.** While those offline causal discovery researches require access to infinite interventions, there has been intensive works that adaptively selects interventions from an online learning. Squires et al. [2020] and Choo and Shiragur [2023] applied interventions sequentially, with adaptively chosen targets at each step, still necessitating access to interventional distributions. Although Greenewald et al. [2019] and Elahi et al. [2024b] worked with finite interventions, it is applicable only when the underlying causal structure has no unobserved confounders. Notably, designing adaptive discovery algorithms that work with finite interventions and allow for unobserved confounders remains an open problem.

It may possible to incorporate online causal discovery into the decision-making process. For example, at each step, an agent can choose interventions aimed at improving structural knowledge, while also expecting that those arms could be valuable for minimizing regret. However, designing algorithms that effectively balance exploration for structure learning and for regret minimization poses substantial additional challenges, as these two objectives—structure discovery and regret minimization—are not naturally aligned.

Furthermore, Wang et al. [2022, 2023b] adaptively refine a PAG by resolving circle marks through targeted interventions. In this sense, interventions on nodes involved in circle marks can be useful for structural refinement. However, as noted in Wang et al. [2024a], obtaining a closed-form characterization of the number of MAGs compatible with a PAG—given a particular orientation of a circle mark—remains an open problem, implying that determining which circle marks to prioritize for learning is itself a challenging problem. As such, designing reliable algorithms that exploit structural uncertainty during learning involves solving nontrivial structure learning problems and remains an active area of research.

### E.3 Comparison with Elahi et al. [2024a]

Elahi et al. [2024a] demonstrated that it is not necessary to learn the full causal diagram to identify all POMISs, and specified the extent of graphical structure that must be discovered to do so. Building on this insight, their work flow proceeds as follows: In the first phase, the method learns the induced subgraph of the ancestors of the reward node in an online manner, through interventions; Using the learned graph, they identify POMISs following the method of Lee and Bareinboim [2018]; Finally, they run standard independent MAB solvers with the identified POMISs.

In contrast, our work does not focus on causal discovery, but instead assumes access to a PAG (e.g., obtained via FCI [Spirtes et al., 2001b, Zhang, 2008a]) and aims to identify POMISs directly from this PAG. That is, given a PAG derived from purely observations, our algorithm prunes suboptimal arms *a priori*, without requiring any interventions for causal discovery. From a practical perspective, interventional data is often more costly and risk-prone than observational data. This suggests that our approach first discovering a PAG from observational data and then identifying POMISs may offer substantial advantages in resource-constrained or high-risk domains, compared to methods that rely on extensive intervention for graph discovery.

Since both our method and Elahi et al. [2024a] ultimately reduce the problem to standard bandits over a set of POMISs, their regret bounds in both approaches depend critically on the size of the resulting POMIS set (each denoted by $\mathcal{I}_{\mathcal{P}}$ and $\mathcal{I}_{\mathcal{G}}$). In Elahi et al. [2024a], the regret bound consists of a discovery term and a minimization: $\mathcal{O}(f(d_{\max}, \delta, \varepsilon)) + \mathcal{O}(\sum_{\mathbf{x} \in \mathcal{I}_{\mathcal{G}}} \Delta_{\mathbf{x}}(1 + \frac{\log T}{\Delta_{\mathbf{x}}^2}))$[11], whereas our regret bound contains only the minimization term: $\mathcal{O}(\sum_{\mathbf{x} \in \mathcal{I}_{\mathcal{P}}} \Delta_{\mathbf{x}}(1 + \frac{\log T}{\Delta_{\mathbf{x}}^2}))$. Since our approach avoids online causal discovery, our regret does not include the additional discovery term. However, because PAGs contain structural uncertainty (e.g., circle marks), the number of POMISs derived from a PAG is typically larger than that from a fully specified causal diagram. As a result, the bandit term of Elahi et al. [2024a] is usually smaller than ours. Therefore, although a direct comparison is difficult due to the different settings, the trade-off can be summarized as eliminating the online discovery cost at the expense of starting with a larger initial action space.

## F  Limitations and Future Works

In this section, we present limitations of our work and outline promising directions.

**Modeling bandit instances in the form of SCMs.**  Structural Causal Models (SCMs) are a versatile and expressive framework that provides a principled way to represent and reason about causal relationships. Their generality makes them applicable across a wide range of domains. However, SCMs come with certain limitations, such as the assumption of a well-defined set of variables and a fixed causal structure, which may not adequately capture the complexity of dynamic, high-dimensional, or partially observed systems. Nonetheless, our work addresses a fundamental problem within the SCM framework. We believe it provides a solid foundation for future research, such as extending causal bandits to more complex or less structured environments.

**Known partial ancestral graphs.**  We make the standard assumption that the deployment-phase learner has access to the true PAG representing the underlying causal diagram. In practice, while several causal discovery methods in the presence of latent confounders have been proposed [Spirtes et al., 2001a, Zhang, 2008a, Colombo et al., 2012, Rohekar et al., 2021, 2023], these techniques typically rely on accurate estimation of conditional independence (CI) relations. This would be especially true for constrained-based algorithms like FCI, where the exact PAG recovery would require many empirical conditional-independence tests to work perfectly. Therefore, our work implicitly assumes that the decision-maker possesses sufficient domain knowledge and statistical capability for reliable CI testing.

**PAG misspecification.**  It is of great practical interest to study how MIS, DMIS and POMIS are affected given PAG misspecification. Notably, the edges in a PAG are governed by structural constraints and logical dependencies such as the balanced property and chordality. Due to the structured entanglements, it is indeed difficult to expect that computing POMISs on an incorrect PAG would yield robust results. Although our work is primarily theoretical and assumes access to the true PAG, we acknowledge that developing robust methodologies that account for such issues is a promising direction for future research.

---

[11]where $d_{\max}$ denotes a constant greater than the maximal in-degree in the true causal diagram, and $\delta$, $\varepsilon$ represent some parameters. See Appendix A.13 of Elahi et al. [2024a] for further details.

**Future work.** In future research, given the availability of an observational distribution, it becomes possible to identify specific causal effects and eliminate suboptimal arms [Jaber et al., 2022]. Moreover, integrating this approach with partial identification [Balke and Pearl, 1995, Richardson et al., 2014, Zhang and Bareinboim, 2020, 2021, Zhang et al., 2022, Bellot, 2024], enables the exclusion of arms where the upper bound is less than the lower bound of another arm, as proposed by Zhang and Bareinboim [2017]. One can account for uncertainty in identification or bounds caused by a finite sample, which will lead to more robust analyzes [Bellot and Chiappa, 2024, Jung and Bellot, 2024]. Beyond causal bandits, we believe that ancestral graphical modeling offers practical value by integrating with causal reinforcement learning [Zhang and Bareinboim, 2022, Hwang et al., 2024, Bareinboim et al., 2024], rehearsal learning [Qin et al., 2023, 2025, Du et al., 2024, 2025, Tao et al., 2025] and sequential planning [Pearl and Robins, 1995a, Jung et al., 2024].

# G   Auxiliary Results

In this section, we provide auxiliary results utilized throughout the paper.

**Lemma 25.** *Let $\mathcal{P}$ be a PAG over $\mathbf{V}$, and let $\mathcal{P}[\mathbf{A}]$ be the subgraph of $\mathcal{P}$ induced by $\mathbf{A} \subseteq$ $\texttt{PossAn}(Y)_{\mathcal{P}} \subseteq \mathbf{V}$. If $X$ and $Z$ belong to different buckets over $\mathcal{P}[\mathbf{A}]$, then the starting edges of any uncovered proper possibly directed paths from $X$ and $Z$ to $Y$ with respect to $\mathbf{X}$ are not relevant.*

*Proof.* Since $X$ and $Z$ are not in the same bucket, there is no circle path connecting the two nodes. Consequently, $X$ and $Z$ are not relevant. $\square$

**Lemma 26.** *Let $\mathcal{P}$ be a PAG over the set of variables $\mathbf{V}$. If a set $\mathbf{X} \subseteq \mathbf{V} \setminus \{Y\}$ is a DMIS relative to $[\![\mathcal{P}, Y]\!]$, then there exists a MAG $\mathcal{M}$ such that every $X \in \mathbf{X}$ has a proper directed path to $Y$ with respect to $\mathbf{X}$ in $\mathcal{M}$.*

*Proof.* According to Prop. 5 and thm. 1, there exists a MAG $\mathcal{M}$ such that $\mathbf{X} \subseteq \texttt{An}(Y)_{\mathcal{M}_{\overline{\mathbf{X}}}}$. For the sake of contradiction, suppose that $\mathbf{X} \subseteq \texttt{An}(Y)_{\mathcal{M}_{\overline{\mathbf{X}}}}$ holds while there is no proper directed path from $X \in \mathbf{X}$ to $Y$ with respect to $\mathbf{X}$ in $\mathcal{M}$. This implies that every directed path from $X$ to $Y$ must contain some node $Z \in \mathbf{X} \setminus \{X\}$. Consequently, such paths would be cut by the $\mathbf{X}$-lower manipulation, resulting in $X \notin \texttt{An}(Y)_{\mathcal{M}_{\overline{\mathbf{X}}}}$. This contradicts the assumption that $\mathbf{X} \subseteq \texttt{An}(Y)_{\mathcal{M}_{\overline{\mathbf{X}}}}$. $\square$

**Lemma 27.** *Let $\mathcal{Q}$ be a PMG obtained from some valid local transformations from a PAG $\mathcal{P}$ and the orientation rules. In $\mathcal{Q}$, the following property holds:*

> *If $A \rightarrow B$ is visible, then every $A \rightarrow C$ is also visible for every $C$ connected as circle path with $B$.*

*Proof.* For the sake of contradiction, assume that there exists a node $C$ such that $A \rightarrow C$ is invisible while connected as circle path with $B$.

First, let $D \mathbin{*}{\rightarrow} A$ be an arbitrary edge that makes $A \rightarrow B$ visible. Since $A \rightarrow C$ is invisible, $D$ and $C$ must be adjacent and the edge is into $C$ by the orientation rule $\mathcal{R}_2$ (i.e., $D \mathbin{*}{\rightarrow} C$). According to Lem. 31, this implies the existence of $D \mathbin{*}{\rightarrow} B$, which contradicts the assumption that $A \rightarrow B$ is visible.

Next, consider the path $D \mathbin{*}{\rightarrow} V_1 \leftrightarrow \cdots \leftrightarrow V_n \leftrightarrow A$ with $n \geq 1$ where $V_i$ is a parent of $B$. By Lem. 31, we get that there exist edges $V_i \mathbin{?}{\rightarrow} C$ for all $V_i$. Furthermore, these edges must take the form $V_i \rightarrow C$, because if any edges $V_i \mathbin{\circ}{\rightarrow} C$ existed, $\mathcal{R}'_4$ would be triggered, resulting in $V_i \rightarrow C$. Therefore, $A \rightarrow C$ is also visible, leading to a contradiction for the assumption that $A \rightarrow C$ is invisible. This concludes the proof. $\square$

**Lemma 28.** *Let $\mathcal{Q}$ be a PMG obtained from some valid local transformations from a PAG $\mathcal{P}$ and the orientation rules, and $\mathcal{G}$ be any causal diagram in the equivalence class represented by $\mathcal{Q}$. Let $X \neq Y$ be two nodes in $\mathbf{A} \subseteq \mathbf{V}(\mathcal{Q})$. If $X$ is an ancestor of $Y$ in $\mathcal{G}[\mathbf{A}]$, then $X$ is a possible ancestor of $Y$ in $\mathcal{Q}[\mathbf{A}]$.*

*Proof.* The lemma follows the proof of Lem. 14 (Prop. 1 in Jaber et al. [2018]). If $X$ is an ancestor of $Y$ in $\mathcal{G}[\mathbf{A}]$, then there exists a directed path $X \rightarrow \cdots \rightarrow Y$ in $\mathcal{G}[\mathbf{A}]$. This path is also present in $\mathcal{G}$, and consequently in the corresponding MAG $\mathcal{M}$. Hence, the path corresponds to a possibly directed path in $\mathcal{Q}$. Since all nodes along the path are in $\mathbf{A}$, they are also present in $\mathcal{Q}[\mathbf{A}]$, implying $X$ is a possible ancestor of $Y$ in $\mathcal{Q}[\mathbf{A}]$. □

**Lemma 29.** *Let $\mathcal{Q}$ be a PMG obtained from some valid local transformations from a PAG $\mathcal{P}$ and the orientation rules, and $\mathcal{G}$ be any causal diagram in the equivalence class represented by $\mathcal{Q}$. Let $X \neq Y$ be two nodes in $\mathbf{A} \subseteq \mathbf{V}(\mathcal{Q})$. For every $X \rightarrow Y$ in $\mathcal{Q}[\mathbf{A}]$, if it is visible in $\mathcal{Q}$, then it remains visible in $\mathcal{Q}[\mathbf{A}]$.*

*Proof.* The proof follows the argument of Lem. 15 (Lem 4. in Jaber et al. [2018]). Let $\mathcal{G}$ defined over $\mathbf{V}(\mathcal{Q}) \cup \mathbf{L}$. Let $X \rightarrow Y$ be a visible edge in $\mathcal{Q}$ where $X$ and $Y$ are in $\mathbf{A}$. Then, there is no inducing path between $X$ and $Y$ relative to $\mathbf{L}$ that is into $X$ in $\mathcal{G}$. It follows that no such inducing path (relative to the latent nodes in $\mathcal{G}[\mathbf{A}]$) exists in the subgraph $\mathcal{G}[\mathbf{A}]$. □

**Lemma 30.** *Let $\mathcal{Q}$ be a PMG obtained from some valid local transformations from a PAG $\mathcal{P}$ and the orientation rules, and $\mathcal{G}$ be any causal diagram in the equivalence class represented by $\mathcal{Q}$. Let $X \neq Y$ be two nodes in $\mathbf{A} \subseteq \mathbf{V}(\mathcal{Q})$. If $X$ and $Y$ are in the same c-component in $\mathcal{G}[\mathbf{A}]$, then $X$ and $Y$ are in the same pc-component in $\mathcal{Q}[\mathbf{A}]$.*

*Proof.* The proof follows the argument of Lem. 16 (Prop. 2 in Jaber et al. [2018]). If $X$ and $Y$ are in the same c-component in $\mathcal{G}[\mathbf{A}]$, then there is a bidirected path $p$ in $\mathcal{G}[\mathbf{A}]$.

> **Lemma I** (Lemma 6 in Jaber et al. [2018]). *Let $\mathcal{M}$ be a MAG over $\mathbf{V}$ and $\mathcal{G}$ be a causal diagram represented by $\mathcal{M}$. For any $X$ and $Y$ in $\mathbf{V}$, if there is a bidirected path $p$ between $X$ and $Y$ in $\mathcal{G}$, then there is a path $p'$ between $X$ and $Y$ in $\mathcal{M}$ over a subsequence of $p$ such that (1) all the non-endpoint nodes are colliders, and (2) all directed edges on $p'$ are invisible.*

> **Lemma II** (Lemma 7 in Jaber et al. [2018]). *Let $\mathcal{M}$ be a MAG over $\mathbf{V}$ and $\mathcal{P}$ be a PAG representing $\mathcal{M}$. For any $X$ and $Y$ in $\mathbf{V}$, if there is a path $p$ between $X$ and $Y$ in $\mathcal{M}$ such that (1) all non-endpoint nodes are colliders and (2) all directed edges, if any, are not visible, then there is a path $p^*$ between $X$ and $Y$ in $\mathcal{P}$ over a subsequence of $p$ such that (1) all non-endpoint nodes along the path are definite colliders, and (2) none of the edges are visible.*

According to Lemma I, we choose a path $p'$, which is the shortest subsequence of $p$ between $X$ and $Y$ in $\mathcal{M}$, corresponding to $p^*$ in $\mathcal{P}$, such that (1) all non-endpoint nodes along the path are colliders, and (2) none of the directed edges are visible. By Lemma II, the path $p^*$ is a definite colliding path between $X$ and $Y$, and none of the directed edges along the path are visible in $\mathcal{P}$. For contradiction, assume that $p^\dagger$ in $\mathcal{Q}$, which is corresponding to $p^*$ in $\mathcal{P}$, includes a visible edge out of $X$. Then, the visible edge would have to appear in all MAGs represented by $\mathcal{Q}$. However, the edge along $p'$ is invisible in $\mathcal{M}$, leading to a contradiction. Therefore, $p^\dagger$ is also of definite status, containing no visible edges, which implies that $X$ and $Y$ are in the same pc-component in $\mathcal{Q}$. Since all nodes along $p^\dagger$ are in $\mathbf{A}$, $p^\dagger$ is also present in $\mathcal{Q}[\mathbf{A}]$, ensuring that $X$ and $Y$ are in the same pc-component in $\mathcal{Q}[\mathbf{A}]$. □

**Lemma 31.** *Let $\mathcal{Q}$ be a PMG obtained from some valid local transformations from a PAG $\mathcal{P}$ and the orientation rules, and $\mathcal{Q}[\mathbf{A}]$ be the induced graph over $\mathbf{A} \subseteq \mathbf{V}(\mathcal{Q})$. For any three nodes $A, B, C$ in $\mathcal{Q}$, if $A \ast\!\!\rightarrow B \circ\!\!-\!\!\ast C$, then there is an edge between $A$ and $C$ with an arrowhead at $C$, namely, $A \ast\!\!\rightarrow C$. Furthermore, if the edge between $A$ and $B$ is $A \rightarrow B$, then the edge between $A$ and $C$ is either $A \rightarrow C$ or $A \circ\!\!\rightarrow C$ (i.e., it is not $A \leftrightarrow C$).*

*Proof.* The balanced property holds in the PMG with local transformations as shown in Thm. 8 (Theorem 1 in Wang et al. [2023b]). By the definition of an induced graph, this property is preserved in $\mathcal{Q}[\mathbf{A}]$. □

**Lemma 32.** *Let $\mathcal{Q}$ be a PMG obtained from some valid local transformations from a PAG $\mathcal{P}$ and the orientation rules. In a PMG $\mathcal{Q}$, for any two nodes $A$ and $B$, if there is a circle path, then following holds:*

1. *If there is an edge between $A$ and $B$, the edge is not into $A$ or $B$;*

2. *For any other node $C$, $C \ast\!\!\rightarrow A$ if and only if $C \ast\!\!\rightarrow B$. Furthermore, $C \leftrightarrow A$ if and only if $C \leftrightarrow B$.*

*Proof.* The proof follows the argument of Lem. 5 (Lem 3.3.2 in Zhang [2006]). The properties depend on the balanced property in Lem. 4, which holds in $\mathcal{Q}$ as demonstrated in Thm. 8 and lem. 31. □

**Lemma 33.** *Let $\mathcal{Q}$ be a PMG obtained from some valid local transformations from a PAG $\mathcal{P}$ and the orientation rules. PTO (Alg. 2) is also sound over $\mathcal{Q}[\mathbf{A}]$, in the sense that the partial order is valid with respect to $\mathcal{G}[\mathbf{A}]$, for every causal diagram $\mathcal{G}$ in the equivalence class represented by $\mathcal{Q}$.*

*Proof.* The proof follows the argument of Lem. 17 (Prop. 4 in Jaber et al. [2018]). By Lem. 28, the possible-ancestral relations in $\mathcal{Q}[\mathbf{A}]$ subsume those in $\mathcal{G}[\mathbf{A}]$. Hence, a partial topological order that is valid with respect to $\mathcal{Q}[\mathbf{A}]$ is also valid with respect to $\mathcal{G}[\mathbf{A}]$. The correctness of Alg. 2 relies solely on the balanced property, which is satisfied in the PMG with local transformations as per Thm. 8 and lem. 31. Thus, the algorithm is also sound with respect to $\mathcal{Q}[\mathbf{A}]$. □

# H   Proofs

In this section, we provide detailed proofs of the propositions and theorems presented in the main body of the paper. For readability, we restate all of them.

**Theorem 1** (Characterization of MIS for MAGs)**.** *Let $\mathcal{M}$ be a MAG over $\mathbf{V}$. Given information $[\![\mathcal{M}, Y]\!]$, a set $\mathbf{X} \subseteq \mathbf{V} \setminus \{Y\}$ is an MIS relative to $[\![\mathcal{M}, Y]\!]$ if and only if $\mathbf{X} \subseteq \mathtt{An}(Y)_{\mathcal{M}_{\overline{\mathbf{X}}}}$ holds.*

*Proof.* **(If)** Suppose that $\mathbf{X}$ is *not* an MIS relative to $[\![\mathcal{M}, Y]\!]$. This implies that there exists some $\mathbf{X}' \subsetneq \mathbf{X}$ such that $\mu_{\mathbf{x}[\mathbf{X}']} = \mu_{\mathbf{x}}$ for every SCM conforming to the MAG $\mathcal{M}$. For the sake of contradiction, assume that $\mathbf{X} \subseteq \mathtt{An}(Y)_{\mathcal{M}_{\overline{\mathbf{X}}}}$. To derive a contradiction, it suffices to construct a SCM such that $\mu_{\mathbf{x}[\mathbf{X}']} \neq \mu_{\mathbf{x}}$. Consider the causal diagram $\mathcal{G}$ generated by the following procedure:

*Step 1. If $A \rightarrow B$ in $\mathcal{M}$, then add a directed edge $A \rightarrow B$ to $\mathcal{G}$.*

*Step 2. If $A \leftrightarrow B$ in $\mathcal{M}$, then add a bidirected edge $A \leftrightarrow B$ to $\mathcal{G}$.*

From this construction, it is clear that the causal diagram $\mathcal{G}$ corresponds to $\mathcal{M}$. Furthermore, we have $\mathbf{X} \subseteq \mathtt{An}(Y)_{\mathcal{G}_{\overline{\mathbf{X}}}}$ since $\mathcal{G}$ and $\mathcal{M}$ have the exact same edges.

Now consider the following SCM associated with $\mathcal{G}$: Each variable in $V_i \in \mathbf{V}(\mathcal{G})$ is associated with a unique latent variable $U_i$ and the function of each endogenous variable in $\mathbf{V}(\mathcal{G})$ is the sum of the

value of its parents. Since $\mathbf{X} \subseteq \text{An}(Y)_{\mathcal{G}_{\overline{\mathbf{X}}}}$ holds, there exist directed paths from $\mathbf{X} \setminus \mathbf{X}'$ to $Y$ without passing through $\mathbf{X}'$. Let $\mathbf{W} = \mathbf{X} \setminus \mathbf{X}'$. Then, setting $\mathbf{W}$ to $\mathbb{E}[\mathbf{W} \mid do(\mathbf{x}')] + 1$ results in a larger outcome value for $Y$, i.e., $\mu_{\mathbf{x}} = \mu_{\mathbf{w},\mathbf{x}'} > \mu_{\mathbf{x}[\mathbf{X}']}$, which leads to a contradiction.

(**Only if**) Suppose that $\mathbf{X} \not\subseteq \text{An}(Y)_{\mathcal{M}_{\overline{\mathbf{X}}}}$ holds. This indicates that there exists a nonempty subset $\mathbf{Z} \triangleq \mathbf{X} \setminus \text{An}(Y)_{\mathcal{M}_{\overline{\mathbf{X}}}}$. Let $\mathbf{X}' = \mathbf{X} \setminus \mathbf{Z}$. Our goal is to show that $Y$ and $\mathbf{Z}$ are m-separated by $\mathbf{X}'$ in $\mathcal{M}_{\overline{\mathbf{X}}}$. Once established, we can apply Rule 3 of do-calculus for MAGs [Zhang, 2008b] to derive $\mu_{\mathbf{x}'} = \mu_{\mathbf{x}',\mathbf{z}}$.

For contradiction, assume that there exists some variable $Z \in \mathbf{Z}$ such that $Z$ and $Y$ are m-connected conditioning on $\mathbf{X}'$ in $\mathcal{M}_{\overline{\mathbf{X}}}$. This means the existence of a m-connected path $p$ between $Z$ and $Y$. Since $Z$ has its incoming edges removed, $p$ must start with an edge outgoing from $Z$. If there were any collider along the path, it would be m-separated, as the collider cannot be an ancestor of a conditioned node $\mathbf{X}'$. However, if the path $p$ begins with an outgoing edge from $Z$ and has no colliders, then it must be a directed path from $Z$ to $Y$. This implies that $Z \in \text{An}(Y)_{\mathcal{M}_{\overline{\mathbf{X}}}}$ holds, thus $\mathbf{Z}$ and $Y$ are not m-separated by $\mathbf{X}'$ in $\mathcal{M}_{\overline{\mathbf{X}}}$, leading to a contradiction. Consequently, we have that $\mathbf{X}$ is not an MIS relative to $[\![\mathcal{M}, Y]\!]$. $\qquad\square$

**Proposition 1.** *Let $\mathcal{M}$ be a MAG over $\mathbf{V}$. A set $\mathbf{X} \subseteq \mathbf{V} \setminus \{Y\}$ is an MIS relative to $[\![\mathcal{M}, Y]\!]$ if and only if there exists a causal diagram $\mathcal{G}$ conforming to $\mathcal{M}$ such that $\mathbf{X}$ is an MIS relative to $[\![\mathcal{G}, Y]\!]$.*

*Proof.* (**If**) Let $\mathbf{X}$ be an MIS relative to $[\![\mathcal{G}, Y]\!]$ for some causal diagram $\mathcal{G}$ conforming to $\mathcal{M}$. By the definition of MIS for causal diagrams in Def. 10, there is no $\mathbf{X}' \subsetneq \mathbf{X}$ such that for all SCM conforming to $\mathcal{G}$, $\mu_{\mathbf{x}[\mathbf{X}']} = \mu_{\mathbf{x}}$. In other words, for every $\mathbf{X}' \subsetneq \mathbf{X}$, there exists an SCM $\mathcal{S}$ conforming to $\mathcal{G}$ such that $\mu_{\mathbf{x}[\mathbf{X}']} \neq \mu_{\mathbf{x}}$. Since any SCM conforming to $\mathcal{G}$ also conforms to $\mathcal{M}$, we know that $\mathcal{S}$ also conforms to $\mathcal{M}$. Thus, for any proper subset $\mathbf{X}' \subsetneq \mathbf{X}$, there exists an SCM associated with $\mathcal{M}$ in which $\mu_{\mathbf{x}[\mathbf{X}']} = \mu_{\mathbf{x}}$ holds.

(**Only if**) Let $\mathbf{X}$ be an MIS relative to $[\![\mathcal{M}, Y]\!]$. The causal diagram $\mathcal{G}$ constructed in the same manner as in the proof of thm. 1 conforms to $\mathcal{M}$ and satisfies $\mathbf{X} \subseteq \text{An}(Y)_{\mathcal{G}_{\overline{\mathbf{X}}}}$. Therefore, we can conclude that $\mathbf{X}$ is an MIS relative to $[\![\mathcal{G}, Y]\!]$ supported by Prop. 9. $\qquad\square$

**Proposition 2** (Graphical characterization of MIS for PAGs)**.** *Let $\mathcal{P}$ be a PAG over the set of variables $\mathbf{V}$. A set $\mathbf{X} \subseteq \mathbf{V} \setminus \{Y\}$ is an MIS relative to $[\![\mathcal{P}, Y]\!]$ if and only if, for every variable $X \in \mathbf{X}$, there exists a proper possibly-directed path from $X$ to $Y$ with respect to $\mathbf{X}$ in $\mathcal{P}$.*

*Proof.* (**If**) Suppose that $\mathbf{X}$ is *not* an MIS relative to $[\![\mathcal{P}, Y]\!]$, which implies that there exists some proper subset $\mathbf{X}' \subsetneq \mathbf{X}$ such that $\mu_{\mathbf{x}[\mathbf{X}']} = \mu_{\mathbf{x}}$ for every SCM conforming to $\mathcal{P}$. For contradiction, suppose that for all $X \in \mathbf{X}$, there exist proper possibly-directed paths from $X$ to $Y$ with respect to $\mathbf{X}$ in $\mathcal{P}$. Let $\mathbf{W} = \mathbf{X} \setminus \mathbf{X}'$ and $W$ be a vertex in $\mathbf{W}$. Suppose that $p$ is an uncovered proper possibly-directed path from $W$ to $Y$ with respect to $\mathbf{X}$ in $\mathcal{P}$. Let $\mathcal{M} \in [\mathcal{P}]$ be a MAG constructed by the following procedure:

    *Step 1. Orient all edges along $p$ as directed edges.*

    *Step 2. Orient the remaining edges according to Lem. 6.*

Then, $p$ corresponds to a proper directed path from $W$ to $Y$ with respect to $\mathbf{X}$ in $\mathcal{M}$. Thus, $W \in \text{An}(Y)_{\mathcal{M}_{\overline{\mathbf{X}}}}$ holds. We can then use the same construction in the proof of Thm. 1. In the constructed causal diagram $\mathcal{G}$, $W \in \text{An}(Y)_{\mathcal{G}_{\overline{\mathbf{X}}}}$ holds. Furthermore, we know there exists an SCM $\mathcal{S}$ in which $W$ has a positive causal effect on $Y$ which is not mediated by any variable in $\mathbf{X}$. Thus, setting $\mathbf{W}$ to $\mathbb{E}[\mathbf{W} \mid do(\mathbf{x}')] + 1$ will result in a larger outcome for $Y$, i.e., $\mu_{\mathbf{x}} = \mu_{\mathbf{w},\mathbf{x}'} > \mu_{\mathbf{x}[\mathbf{X}']}$, meaning $\mu_{\mathbf{x}} \neq \mu_{\mathbf{x}[\mathbf{X}']}$, which contradicts the statement: $\mu_{\mathbf{x}[\mathbf{X}']} = \mu_{\mathbf{x}}$ for every SCM conforming to $\mathcal{P}$.

(**Only if**) Suppose that for some $Z \in \mathbf{X}$, there is no proper possibly directed path from $Z$ to $Y$ with respect to $\mathbf{X}$ in $\mathcal{P}$. Let $\mathbf{X}' = \mathbf{X} \setminus \{Z\}$. We aim to show that $P(y \mid do(\mathbf{x}')) = P(y \mid do(\mathbf{x}', z))$,

which would imply $\mu_{\mathbf{x}'} = \mu_{\mathbf{x}',z}$. Unfortunately, we cannot apply Rule 3 of do-calculus for PAGs, since it is not guaranteed that $X$ and $Y$ are definitely m-separated by $\mathbf{X}'$ in $\mathcal{P}_{\overline{\mathbf{X}}}$. However, we can reason over the MAGs in the Markov equivalence class represented by $\mathcal{P}$.

All paths from $Z$ to $Y$ in $\mathcal{P}$ which do not pass through $\mathbf{X}$ must not be a directed path due to our assumption, i.e., they all contain an arrowhead pointing towards $Z$. Let $\mathcal{M}$ be a MAG conforming to $\mathcal{P}$. Then, all paths from $Z$ to $Y$ in $\mathcal{M}$ which do not pass through $\mathbf{X}$ must also be non-directed. Thus, using similar reasoning as in the proof of Thm. 1, $Z$ and $Y$ are m-separated by $\mathbf{X}'$ in $\mathcal{M}_{\overline{\mathbf{X}}}$. This is because any path out of $Z$ to $Y$ must contain a collider node, which must be blocked, since it cannot be an ancestor of any conditioned node. Therefore, we conclude that $P(y \mid do(\mathbf{x}')) = P(y \mid do(\mathbf{x}', z))$. Since this argument holds for every MAG conforming to $\mathcal{P}$, it holds for all SCMs conforming to $\mathcal{P}$. $\qquad\square$

**Proposition 4.** *Let $\mathcal{D}$ be either a causal diagram or a MAG (i.e., not a PAG). If $\mathbf{X}$ is an MIS with respect to $[\![\mathcal{D}, Y]\!]$, then $\mathbf{X}$ is a DMIS with respect to $[\![\mathcal{D}, Y]\!]$.*

*Proof.* Without loss of generality, assume that all nodes in $\mathcal{D}$ are ancestors of $Y$. For contradiction, assume that $\mathbf{X}$ is an MIS but *not* a DMIS relative to $[\![\mathcal{D}, Y]\!]$. By Thm. 1 and prop. 9, we have $\mathbf{X} \subseteq \texttt{An}(Y)_{\mathcal{D}_{\overline{\mathbf{X}}}}$. Then, we can consider an SCM $\mathcal{S}^*$ compatible with $\mathcal{D}$, where all mechanisms consist of the sum of the values of their parents, i.e., $f_V = \sum_{|\mathbf{pa}_V|} \mathbf{pa}_V + \mathbf{u}_V$. Let $\mathbf{X}'$ be an arbitrary proper subset of $\mathbf{X}$, and $\mathbf{W}$ denote $\mathbf{X} \setminus \mathbf{X}'$. Such a model $\mathcal{S}^*$ always ensures that setting $\mathbf{W}$ as $\mathbb{E}[\mathbf{W} \mid do(\mathbf{x}')] + 1$ results in $\mu_{\mathbf{x}} = \mu_{\mathbf{w},\mathbf{x}'} > \mu_{\mathbf{x}[\mathbf{X}']}$ for any proper subset $\mathbf{X}'$ since there exist directed paths from each $W \in \mathbf{W}$ to $Y$ without passing through $\mathbf{X}'$. The existence of $\mathcal{S}^*$ leads to a contradiction. $\qquad\square$

**Proposition 5.** *Let $\mathcal{P}$ be a PAG over $\mathbf{V}$. A set $\mathbf{X} \subseteq \mathbf{V} \setminus \{Y\}$ is a DMIS relative to $[\![\mathcal{P}, Y]\!]$ if and only if there exists a MAG $\mathcal{M}$ conforming to $\mathcal{P}$ such that $\mathbf{X}$ is an MIS relative to $[\![\mathcal{M}, Y]\!]$.*

*Proof.* (**If**) Suppose $\mathbf{X} \subseteq \mathbf{V} \setminus \{Y\}$ be an MIS relative to $[\![\mathcal{P}, Y]\!]$, and there exists a MAG $\mathcal{M}$ conforming to $\mathcal{P}$ where $\mathbf{X}$ is an MIS relative to $[\![\mathcal{M}, Y]\!]$. By Prop. 4, $\mathbf{X}$ is a DMIS relative to $[\![\mathcal{M}, Y]\!]$. Hence, there exists an SCM $\mathcal{S}$ such that for any proper subset $\mathbf{X}'$, $\mu_{\mathbf{x}[\mathbf{X}']} \neq \mu_{\mathbf{x}}$ holds. Since $\mathcal{S}$ conforms to $\mathcal{M}$, it also conforms to $\mathcal{P}$, thus concluding proof for this direction.

(**Only if**) Suppose $\mathbf{X} \subseteq \mathbf{V} \setminus \{Y\}$ be a DMIS relative to $[\![\mathcal{P}, Y]\!]$. By the definition of DMIS (2), there exists an SCM $\mathcal{S}$ associated with $\mathcal{P}$ such that, for every $\mathbf{X}' \subsetneq \mathbf{X}$, $\mu_{\mathbf{x}[\mathbf{X}']} \neq \mu_{\mathbf{x}}$ holds. Therefore, $\mathbf{X}$ is an MIS, since for any proper subset $\mathbf{X}'$, $\mu_{\mathbf{x}[\mathbf{X}']} \neq \mu_{\mathbf{x}}$ holds under the SCM $\mathcal{S}$. $\qquad\square$

**Theorem 2** (Graphical characterization of DMIS for PAGs). *Let $\mathcal{P}$ be a PAG over the set of variables $\mathbf{V}$. A set $\mathbf{X} \subseteq \mathbf{V} \setminus \{Y\}$ is a DMIS relative to $[\![\mathcal{P}, Y]\!]$ if and only if, for any pair of vertices $X, Z \in \mathbf{X}$, there exist uncovered proper possibly-directed paths from $X$ and $Z$ to $Y$ with respect to $\mathbf{X}$ such that their starting edges are not relevant.*

*Proof.* (**If**) Let $p_X$ denote an uncovered proper possibly-directed path from $X$ to $Y$ with respect to $\mathbf{X}$ in $\mathcal{P}$. Suppose that $\mathbf{X}$ is *not* a DMIS, implying that, for all MAGs $\mathcal{M} \in [\mathcal{P}]$, it holds that $Z \notin \texttt{An}(Y)_{\mathcal{M}_{\overline{\mathbf{X}}}}$ and $X \in \texttt{An}(Y)_{\mathcal{M}_{\overline{\mathbf{X}}}}$ without loss of generality. In other words, if orienting $p_X$ as $X \to \cdots \to Y$ is valid, it follows that orienting any possibly directed path from $Z$ to $Y$ as $Z \to \cdots \to Y$ is invalid in all MAGs conforming to $\mathcal{P}$. We will show that the starting edge of $p_X$ is relevant to the starting edge of any uncovered possibly-directed path from $Z$ to $Y$ in $\mathcal{P}$.

Let $p_Z$ be an arbitrary uncovered proper possibly-directed path from $Z$ to $Y$ with respect to $\mathbf{X}$ in $\mathcal{P}$. Note that such a path always exists, as established by Lem. 7. We know that the path $p_Z$ must begin with one of the following edges: $\circ\!\!-\!\!\circ$, $\circ\!\!\to$, or $\to$. We will show that $p_Z$ can *only* start with a circle edge ($\circ\!\!-\!\!\circ$).

($p_Z$ **only starts with a circle edge** ($\circ\!\!-\!\!\circ$)). Suppose $p_Z$ starts with $^?\!\!\to$. Then, the path must take the form $Z \stackrel{?}{\to} \cdot \to \cdots \to Y$ in $\mathcal{P}$ by Lem. 8. In this case, we can construct a valid $\mathcal{M}$ by orienting any

circle marks (○) along the path as tails (−) following Lem. 6. This contradicts the assumption that there is no MAG conforming $\mathcal{P}$ in which $p_Z$ is a directed path from $Z$ to $Y$. Therefore, we conclude that $p_Z$ only can be $Z \circ\!\!-\!\!\circ \cdots \ast\!\!-\!\!\ast Y$.

For the sake of contradiction, assume $e_X(X \ast\!\!-\!\!\ast X')$ is not relevant to $e_Z(Z' \circ\!\!-\!\!\circ Z)$ where each denotes the starting edges of $p_X$ and $p_Z$ respectively; Then, we consider the following two cases separately: ① $X$ and $Z$ are *not* in the same bucket, or ② they are in the same bucket, and every circle path including $e_X$ and $e_Z$ is not uncovered, i.e., they are *not* relevant.

(① $X$ **and** $Z$ **do not belong to the same bucket**). Consider the orientation according to Lem. 6. In the second step of the construction, we always have a MAG $\mathcal{M}$ containing $Z \to Z'$ by the completeness of orientation in PAGs, which indicates $p_Z$ corresponds to a directed path from $Z$ to $Y$ in $\mathcal{M}$, as it is uncovered. Therefore, we can construct a valid $\mathcal{M}$ according to Lem. 6, contradicting the assumption that $Z \notin \texttt{An}(Y)_{\mathcal{M}_{\overline{\mathbf{X}}}}$ for all MAGs $\mathcal{M} \in [\mathcal{P}]$.

(② $X$ **and** $Z$ **are in the same bucket**). Suppose that $X$ and $Z$ are in the same bucket. Let $V_1(= X) \circ\!\!-\!\!\circ V_2(= X') \circ\!\!-\!\!\circ \cdots \circ\!\!-\!\!\circ V_{n-1}(= Z') \circ\!\!-\!\!\circ V_n(= Z)$ be an arbitrary non-uncovered circle path between $X$ and $Z$ in $\mathcal{P}$. By the definition of an uncovered circle path, such a path must include at least one non-uncovered triple $\langle V_i, V_{i+1}, V_{i+2} \rangle$ on the circle path. The existence of an edge between $V_i \circ\!\!-\!\!\circ V_{i+2}$ would induce an uncovered circle path $V_1 \circ\!\!-\!\!\circ \cdots \circ\!\!-\!\!\circ V_i \circ\!\!-\!\!\circ V_{i+2} \circ\!\!-\!\!\circ \cdots \circ\!\!-\!\!\circ V_n$. To avoid this, $X$ and $Z$ must be adjacent, and furthermore, the edge connecting $X$ and $Z$ must appear as a circle edge $X \circ\!\!-\!\!\circ Z$ by Lem. 11.

The existence of the edge $X \circ\!\!-\!\!\circ Z$ implies that there must be edges $X \circ\!\!-\!\!\circ V_i$ for all $3 \leq i \leq n-1$, or $Z \circ\!\!-\!\!\circ V_i$ for all $2 \leq i \leq n-2$ by chordality. In the former case, we orient the subgraph of $\mathcal{P}$ over $\{V_1, \cdots, V_n\}$ following a similar approach to the proof of Lemma 7.6 in Maathuis and Colombo [2015]. We begin by selecting a vertex $V_2$ and orient all edges incident to $V_2$ as directed into $V_2$. Since the subgraph is chordal and $V_2$ is simplicial, this orientation does not create any uncovered colliders in the subgraph. We then remove $V_2$ and the oriented edges from the subgraph. The resulting graph remains chordal and therefore again choose a vertex $V_3$, and orient any edges incident to $V_3$ into $V_3$. We continue this procedure until all edges are oriented. The constructed subgraph does not create any directed cycle, almost directed cycle, or uncovered collider, thus it is valid orientations. Since $X \to X' \to \cdots \to Y$ is valid, we have a directed path $Z \to Z' \to \cdots \to X' \to \cdots \to Y$ which leads to a contradiction.

In the latter case, we can similarly orient the edges, starting from $V_{n-1}$ and proceeding to $V_2$. Furthermore, this procedure can also be extended to cases where the graph takes on a superimposed form.

(**Only if**) Suppose that $e_X$ is relevant to $e_Z$ in $\mathcal{P}$. It follows that $V_1(= X) \circ\!\!-\!\!\circ V_2(= X') \circ\!\!-\!\!\circ \cdots \circ\!\!-\!\!\circ V_{n-1}(= Z') \circ\!\!-\!\!\circ V_n(= Z)$ is an uncovered circle path. For the sake of contradiction, assume that $\mathbf{X}$ is a DMIS relative to $[\![\mathcal{P}, Y, ]\!]$. Then, there exists a MAG $\mathcal{M}$ conforming to $\mathcal{P}$ such that both $p_X$ and $p_Z$ are proper directed paths with respect to $\mathbf{X}$ in $\mathcal{M}$. Therefore, we can orient $V_1 \circ\!\!-\!\!\circ V_2$ as $V_1 \to V_2$, and $V_n \circ\!\!-\!\!\circ V_{n-1}$ as $V_n \to V_{n-1}$ to construct $\mathcal{M}$ from $\mathcal{P}$. Furthermore, since the circle path is uncovered, $V_i \circ\!\!-\!\!\circ V_{i+1}$ must be oriented $V_i \to V_{i+1}$ for $i = 2, \cdots, n-2$. However, this orientation introduces a new uncovered collider $V_{n-2} \to V_{n-1} \leftarrow V_n$, which leads to a contradiction. $\qquad \square$

**Theorem 3** (Graphical characterization of POMIS for MAGs). *Let $\mathcal{M}$ be a MAG over the set of variable $\mathbf{V}$. A set $\mathbf{X} \subseteq \mathbf{V} \setminus \{Y\}$ is a POMIS relative to $[\![\mathcal{M}, Y]\!]$ if and only if $\mathbf{X} = \textsf{IB}(\mathcal{M}, Y, \mathbf{X})$.*

*Proof.* (**Only if**) We will show contrapositive, i.e., if $\mathbf{X} = \textsf{IB}(\mathcal{M}, Y, \mathbf{X})$ does not hold, then $\mathbf{X}$ is not a POMIS relative to $[\![\mathcal{M}, Y]\!]$. We denote $\mathbf{W} = \textsf{IB}(\mathcal{M}, Y, \mathbf{X})$ and $\mathbf{T} = \textsf{MUCT}(\mathcal{M}, Y, \mathbf{X})$, assuming $\mathbf{X} \neq \mathbf{W}$. Let $\mathbf{W}' \triangleq \mathbf{W} \setminus \mathbf{X}$. Before proceeding with the main proof, we first establish that the following conditional independence statement holds:

    **Claim 1.** $(Y \perp\!\!\!\perp \mathbf{W}' \mid \mathbf{X})$ *holds in* $\mathcal{M}_{\overline{\mathbf{X}}\underline{\mathbf{W}'}}$[12].

---

[12] Note that lower-manipulation has a higher priority than upper-manipulation so that $\mathcal{Q}_{\overline{\mathbf{X}}\underline{Y}}$ or $\mathcal{Q}_{\underline{Y}\overline{\mathbf{X}}}$ denotes the graph resulting from applying the $\mathbf{X}$-upper-manipulation to the $Y$-lower-manipulated graph of $\mathcal{Q}$.

*Proof.* Suppose that the negation of this statement holds: $(Y \not\perp\!\!\!\perp \mathbf{W}' \mid \mathbf{X})$ in $\mathcal{M}_{\overline{\mathbf{X}}\underline{\mathbf{W}'}}$. This would imply that there exists an m-connected path from some $W \in \mathbf{W}'$ to $Y$ given $\mathbf{X}$ in $\mathcal{M}_{\overline{\mathbf{X}}\underline{\mathbf{W}'}}$. For the m-connected path to exist, there must be no colliders, as no node along the path can be an ancestor of $\mathbf{X}$ due to all incoming edges to $\mathbf{X}$ being cut in $\mathcal{M}_{\overline{\mathbf{X}}}$. Moreover, as all outgoing edges from $\mathbf{W}'$ are cut in $\mathcal{M}_{\underline{\mathbf{W}'}}$, the path cannot begin with an edge going out of $W$. Therefore, we get that the m-connected path must be of the following form: $W \leftarrow W_1 \leftarrow \cdots \leftarrow W_n \leftrightarrow R_1 \rightarrow \cdots \rightarrow R_m \rightarrow Y$ with $n, m \geq 0$ where no node along the path can be in $\mathbf{W}$; otherwise, it would either be part of $\mathbf{X}$, since we are conditioning on $\mathbf{X}$, or in $\mathbf{W}'$, in which case all of its outgoing arrows would have been removed. Since $Y$ is contained in $\mathbf{T}$, the parent of $Y$, $R_m$, along the path must be either in $\mathbf{T}$ or $\mathbf{W}$. However, as previously argued, no node along the path can be in $\mathbf{W}$; therefore, it must be in $\mathbf{T}$. This reasoning can be applied iteratively up to $R_1$, implying that $R_1$ is also in $\mathbf{T}$. Since $\mathbf{T}$ is closed under PC, the inclusion of $R_1$ in $\mathbf{T}$ implies that $W_n$ must also be in $\mathbf{T}$. Additionally, because $\mathbf{T}$ is closed under descendants, $W_{n-1}, \cdots, W_1$ must also be in $\mathbf{T}$. Consequently, $W$ must be in $\mathbf{T}$ as well. However, this leads to a contradiction, since $W$ is in $\mathbf{W}$, and $\mathbf{W}$ and $\mathbf{T}$ are disjoint by definition. Therefore, the conditional independence statement $(Y \perp\!\!\!\perp \mathbf{W}' \mid \mathbf{X})$ must hold in $\mathcal{M}_{\overline{\mathbf{X}}\underline{\mathbf{W}'}}$. $\qquad\square$

**Claim 2.** $(Y \perp\!\!\!\perp \mathbf{X}' \mid \mathbf{W})$ *holds in* $\mathcal{M}_{\overline{\mathbf{W}},\overline{\mathbf{X}'}}$ *where* $\mathbf{X}' \triangleq \mathbf{X} \setminus \mathbf{W}$.

*Proof.* Suppose this statement is *false*, i.e., $(Y \not\perp\!\!\!\perp \mathbf{X}' \mid \mathbf{W})$ holds in $\mathcal{M}_{\overline{\mathbf{W}},\overline{\mathbf{X}'}}$. Then, there exists an m-connected path from some $X \in \mathbf{X}'$ to $Y$ given $\mathbf{W}$ in $\mathcal{M}_{\overline{\mathbf{W}},\overline{\mathbf{X}'}}$. Since all edges into $\mathbf{X}'$ are removed, the path must begin with an edge going out of $X$. The path cannot contain any colliders, as no node can be an ancestor of a node in the conditioned set $\mathbf{W}$, given that all incoming edges to $\mathbf{W}$ are cut. Thus, all edges along the path must be directed, pointing to $Y$: $X \rightarrow W_1 \rightarrow \cdots \rightarrow W_n \rightarrow Y (n \geq 0)$ where no node along the path can be in $\mathbf{W}$, since we are conditioning on $\mathbf{W}$. The parent of $Y$, $W_n$, along the path must be either in $\mathbf{T}$ or $\mathbf{W}$, as $Y$ in $\mathbf{T}$. However, as previously argued, no node along the path can be included in $\mathbf{W}$, which means it must be in $\mathbf{T}$. This reasoning can be applied iteratively up to $W_1$, implying that $W_1$ is also in $\mathbf{T}$. Therefore, $X$ must be a parent of a node in $\mathbf{T}$, implying that $X$ is in $\mathbf{W}$. This leads to a contradiction for $X \in \mathbf{X} \setminus \mathbf{W}$. $\qquad\square$

We are now ready to proceed to the main proof. We will show that $\mathbf{X}$ is *not* a POMIS by proving that $\mu_{\mathbf{x}^*} \leq \mu_{\mathbf{w}^*}$ in every SCM conforming to $\mathcal{M}$. We derive that the following holds:

$$
\begin{aligned}
\mu_{\mathbf{x}^*} &= \mathbb{E}[Y \mid do(\mathbf{x}^*)] \\
&= \sum_{\mathbf{w}'} \mathbb{E}[Y \mid do(\mathbf{x}^*), \mathbf{w}']P(\mathbf{w}' \mid do(\mathbf{x}^*)) \\
&= \sum_{\mathbf{w}'} \mathbb{E}[Y \mid do(\mathbf{x}^*), do(\mathbf{w}')]P(\mathbf{w}' \mid do(\mathbf{x}^*)) && \because \textbf{Claim 1} \\
&= \sum_{\mathbf{w}'} \mathbb{E}[Y \mid do(\mathbf{x}^*[\mathbf{W}]), do(\mathbf{w}')]P(\mathbf{w}' \mid do(\mathbf{x}^*)) && \because \textbf{Claim 2} \\
&\leq \sum_{\mathbf{w}'} \mathbb{E}[Y \mid do(\mathbf{w}')]P(\mathbf{w}' \mid do(\mathbf{x}^*)) \\
&= \mathbb{E}[Y \mid do(\mathbf{w}^*)] \\
&= \mu_{\mathbf{w}^*}.
\end{aligned}
$$

Therefore, $\mathbf{X}$ is *not* a POMIS with respect to $[\![\mathcal{M}, Y]\!]$, which completes the proof.

**(If)** To prove this direction, we will show that if $\mathbf{X} = \mathsf{IB}(\mathcal{M}, Y, \mathbf{X})$, then $\mathbf{X}$ is a POMIS relative to $[\![\mathcal{M}, Y]\!]$. Suppose that $\mathbf{X} = \mathsf{IB}(\mathcal{M}, Y, \mathbf{X})$ holds. It suffices to show that there exists a causal diagram $\mathcal{G}$ such that $\mathbf{X}$ is a POMIS relative to $[\![\mathcal{G}, Y]\!]$. Consider the causal diagram $\mathcal{G}$ constructed by the following lemma:

**Lemma 34.** *Let $\mathcal{M}$ be a MAG. Let $\mathcal{G}$ be the graph resulting from the following procedure applied to $\mathcal{M}$.*

    *Step 1. For each visible edge $A \rightarrow B$ in $\mathcal{M}$, add $A \rightarrow B$ in $\mathcal{G}$.*

*Step 2. For each bidirected edge $A \leftrightarrow B$ in $\mathcal{M}$, add $A \leftrightarrow B$ in $\mathcal{G}$.*

*Step 3. For each invisible directed edge $A \to B$ in $\mathcal{M}$, if it is the unique invisible edge among directed edges outgoing from $A$ in $\mathcal{M}$, then add both a directed edge $A \to B$ and bidirected edge $A \leftrightarrow B$ to $\mathcal{G}$ .*

*Step 4. Let $\mathbf{T}_\mathcal{G} \triangleq \mathsf{MUCT}(\mathcal{G}, Y)$. Consider all nodes $A$ for which there are invisible edges outgoing from $A$ in $\mathcal{M}$.*

1. *If there exists $B \in \mathtt{Ch}(A)_\mathcal{M}$ that is contained in $\mathbf{T}_\mathcal{G}$, add both a directed edge $A \to B$ and bidirected edge $A \leftrightarrow B$, and add directed edges $A \to C$ for all $C \in \mathtt{Ch}(A)_\mathcal{M} \setminus \{B\}$.*
2. *Otherwise, if there is no intersection with $\mathbf{T}_\mathcal{G}$, add directed edges $A \to C$ for all $C \in \mathtt{Ch}(A)_\mathcal{M}$.*

*This step is repeated with the updated $\mathbf{T}_\mathcal{G} \leftarrow \mathsf{MUCT}(\mathcal{G}, Y)$ as long as $\mathcal{G}$ remains unchanged.*

*Then, the result graph $\mathcal{G}$ is a causal diagram conforming to $\mathcal{M}$.*

*Proof.* We need to show that $\mathcal{G}$ and $\mathcal{M}$ have the same ancestral relations, and the same conditional independence relations.

(① $\mathcal{G}$ **and** $\mathcal{M}$ **have the same ancestral relations).** This is evident, as each directed edge is added to $\mathcal{G}$ *if and only if* it also exists in $\mathcal{M}$.

(② $\mathcal{G}$ **and** $\mathcal{M}$ **encode the same independence relations).** The graphs $\mathcal{G}$ and $\mathcal{M}$ differ only in the bidirected edges added to $\mathcal{G}$ corresponding to invisible edges in $\mathcal{M}$. Thus, it suffices to show that these additional bidirected edges added to $\mathcal{G}$ do not encode any additional independence between variables. Therefore, we need to show that these edges do not create any new uncovered colliders.

Consider a bidirected edge $A \leftrightarrow B$ added to $\mathcal{G}$ in **Step 3**. For this added edge to create a collider, there must be either a directed edge incoming to $A$ (i.e., $C \to A \leftrightarrow B$), or bidirected edge incoming to $A$ (i.e., $C \leftrightarrow A \leftrightarrow B$) in $\mathcal{G}$. In both cases, $B$ and $C$ are adjacent in $\mathcal{M}$, since $A \to B$ is invisible in $\mathcal{M}$ by Lem. 19. Therefore, this collider at $A$ does not introduce any new independence.

Now consider a bidirected edge $A \leftrightarrow B$ added to $\mathcal{G}$ in **Step 4**. The previous argument can be reused here to argue that this edge does not encode any new independence, since we add only one bidirected among outgoing directed edges from $A$. For clarity, suppose that we have a MAG $\mathcal{M} = \langle A \to B, A \to B, A \to D \rangle$ where $B,C$, and $D$ are mutually not adjacent in $\mathcal{M}$. Adding at most one of $A \leftrightarrow C$, $A \leftrightarrow B$, or $A \leftrightarrow D$ does not introduce a new collider at $A$, thereby preserving conditional independence. $\qquad\square$

Let $\mathcal{G}$ be the causal diagram constructed following Lem. 34. We will prove that $\mathbf{X}$ is a POMIS with respect to $[\![\mathcal{G}, Y]\!]$. Let $X$ be any variable in $\mathbf{X}$. Then $X$ is a parent of some $T \in \mathsf{MUCT}(\mathcal{M}, Y, \mathbf{X})$ in $\mathcal{M}$. It suffices to show that $T \in \mathsf{MUCT}(\mathcal{G}_{\overline{\mathbf{X}}}, Y)$ since this means that $X$ is a parent of a member of $\mathsf{MUCT}(\mathcal{G}_{\overline{\mathbf{X}}}, Y)$, and is therefore in $\mathsf{IB}(\mathcal{G}_{\overline{\mathbf{X}}}, Y)$.

Let $\mathbf{T}_\mathcal{G} \triangleq \mathsf{MUCT}(\mathcal{G}, Y)$ and $\mathbf{T}_\mathcal{M} \triangleq \mathsf{MUCT}(\mathcal{M}, Y, \emptyset)$. We will show that $\mathbf{T}_\mathcal{M} \subseteq \mathbf{T}_\mathcal{G}$. Let $T$ be a node in $\mathbf{T}_\mathcal{M}$. We know such a node always exists because $Y$ is in both $\mathbf{T}_\mathcal{G}$ and $\mathbf{T}_\mathcal{M}$. Let $\mathcal{H} \triangleq \mathcal{G}[\mathtt{An}(Y)_\mathcal{G}]$ and $\mathcal{N} \triangleq \mathcal{M}[\mathtt{An}(Y)_\mathcal{M}]$. Since $\mathcal{M}$ and $\mathcal{G}$ share the same skeleton and the same ancestral relations among vertices, it follows that $\mathtt{An}(Y)_\mathcal{M} = \mathtt{An}(Y)_\mathcal{G}$, implying $\mathbf{V}(\mathcal{H}) = \mathbf{V}(\mathcal{N})$.

(① **If** $W \in \mathsf{PC}(T)_\mathcal{N}$**, then** $W \in \mathbf{T}_\mathcal{G}$**).** Suppose that another node $W$ is in the same pc-component of $T$ in $\mathcal{N}$, i.e., $W \in \mathsf{PC}(T)_\mathcal{N}$. This implies that there exists a path between $T$ and $W$ in $\mathcal{N}$ such that (i) all non-endpoint nodes along the path are colliders, and (ii) none of the edges are visible.

For all directed edges $U \to V$ along this path, if there does not exist an edge $U \to Z(\neq V)$ in $\mathcal{N}$, a bidirected edge $U \leftrightarrow V$ is added to $\mathcal{G}$ in Step 3. Consequently, $T$ and $W$ are in the same c-component in $\mathcal{H}$.

Otherwise, if there is some directed edge $U \to V$ along the path for which there exists $U \to Z(\neq V)$, then from Step 4, we know that one of these outgoing edges from $U$ will have a corresponding bidirected edge in $\mathcal{H}$ which adds $U$ to $\mathbf{T}_\mathcal{G}$. Since MUCT is closed under descendants, all descendants of $U$ are also included in MUCT as well.

This logic applies along the entire path, ensuring that $T \in \mathbf{T}_\mathcal{G} \Rightarrow W \in \mathbf{T}_\mathcal{G}$.

(② **If** $W \in \text{De}(T)_\mathcal{N}$**, then** $W \in \mathbf{T}_\mathcal{G}$**).** Now, suppose that $W$ is a descendant of $T$ in $\mathcal{N}$, i.e., $W \in \text{De}(T)_\mathcal{N}$. Then $W$ is a descendant of $T$ in $\mathcal{H}$ as well, and so we have $T \in \mathbf{T}_\mathcal{G} \Rightarrow W \in \mathbf{T}_\mathcal{G}$.

(① + ② **implies** $\mathbf{T}_\mathcal{M} \subseteq \mathbf{T}_\mathcal{G}$**).** Thus, we have shown that any node which can be shown to be in $\mathbf{T}_\mathcal{M}$ can also be shown to be in $\mathbf{T}_\mathcal{G}$, and therefore $\mathbf{T}_\mathcal{M} \subseteq \mathbf{T}_\mathcal{G}$.

It can be applied to show that $\text{MUCT}(\mathcal{M}, Y, \mathbf{X}) \subseteq \text{MUCT}(\mathcal{G}_{\overline{\mathbf{X}}}, Y)$, as we can operate over $\mathcal{M} \setminus \mathbf{X}$ and $\mathcal{G} \setminus \mathbf{X}$ instead of $\mathcal{M}$ and $\mathcal{G}$, respectively. Thus, we have that $T \in \text{MUCT}(\mathcal{M}, Y, \mathbf{X})$ implies $T \in \text{MUCT}(\mathcal{G}_{\overline{\mathbf{X}}}, Y)$. Therefore, we can conclude that $\text{IB}(\mathcal{G}_{\overline{\mathbf{X}}}, Y) = \mathbf{X}$ holds. $\qquad\square$

**Proposition 6.** *Let $\mathcal{P}$ be a PAG over $\mathbf{V}$. A set $\mathbf{X} \subseteq \mathbf{V} \setminus \{Y\}$ is a POMIS relative to $[\![\mathcal{P}, Y]\!]$ if and only if there exists a MAG $\mathcal{M}$ conforming to $\mathcal{P}$ such that $\mathbf{X}$ is a POMIS relative to $[\![\mathcal{M}, Y]\!]$.*

*Proof.* (**If**) Suppose $\mathbf{X}$ is a POMIS relative to $[\![\mathcal{M}, Y]\!]$ for some $\mathcal{M}$ conforming to $\mathcal{P}$. Then there exists an SCM $\mathcal{S}$ conforming to $\mathcal{M}$ such that $\mu_{\mathbf{x}^*} > \forall_{\mathbf{W} \in \mathbb{D}_{\mathcal{M},Y} \setminus \{\mathbf{X}\}} \mu_{\mathbf{w}^*}$. Since any SCM conforming to $\mathcal{M}$ also conforms to $\mathcal{P}$, the SCM also conforms to $\mathcal{P}$, the SCM $\mathcal{S}$ also conforms to $\mathcal{P}$, and thus $\mathbf{X}$ is a POMIS relative to $[\![\mathcal{P}, Y]\!]$.

(**Only if**) Let $\mathbf{X}$ be a POMIS relative to $[\![\mathcal{P}, Y]\!]$. Then there exists an SCM $\mathcal{S}$ conforming to $\mathcal{P}$ such that $\mu_{\mathbf{x}^*} > \forall_{\mathbf{W} \in \mathbb{D}_{\mathcal{P},Y} \setminus \{\mathbf{X}\}} \mu_{\mathbf{w}^*}$. Let $\mathcal{G}$ be the causal diagram associated with the SCM $\mathcal{S}$. Then, there exists a MAG $\mathcal{M}$ representing $\mathcal{G}$ that corresponds to $\mathcal{P}$ with $\mathbf{X}$ as a POMIS relative to $[\![\mathcal{M}, Y]\!]$, since $\mathbb{P}_{\mathcal{P},Y} \subseteq \mathbb{D}_{\mathcal{P},Y}$. This concludes the proof for this direction. $\qquad\square$

**Proposition 7.** *Let $\mathcal{Q}_\mathbf{X}$ be a PMG representing MAGs where $\mathbf{X}$ is a POMIS with respect to $Y$. Then, the following properties hold in $\mathcal{Q}_\mathbf{X}$, for every $X \in \mathbf{X}$:*

1. *Every uncovered proper possibly-directed path from $X$ to $Y$ relative to $\mathbf{X}$ ends with an arrowhead ($>$).*

2. *If $X$ is adjacent to $Y$, then the edge between $X$ and $Y$ is a directed edge ($X \to Y$).*

*Proof.* We will show that the conditions are necessary for $\mathbf{X}$ to be an MIS in the MAGs, which implies that they are also necessary for $\mathbf{X}$ to be a POMIS.

(**First condition**). For the sake of contradiction, suppose that there exists an uncovered path ending with a tail mark at $Y$ in a MAG $\mathcal{M} \in [\mathcal{Q}_\mathbf{X}]$. This implies the path must take the form $X \leftarrow \cdots \leftarrow Y$ in $\mathcal{M}$. Since $\mathbf{X}$ is an MIS relative to $[\![\mathcal{M}, Y]\!]$, there exists a directed path from $X$ to $Y$ in $\mathcal{M}$, which would introduce a directed cycle, leading to a contradiction.

(**Second condition**). We will first show $X *\!\!-\!\!* Y$ forms $X *\!\!\to Y$ in $\mathcal{Q}_\mathbf{X}$, and then demonstrate that it must be $X \to Y$ by proving that $X \leftrightarrow Y$ leads to a contradiction. For the sake of contradiction, assume that there exists $X \leftarrow Y$ in a MAG $\mathcal{M} \in [\mathcal{Q}_\mathbf{X}]$. In $\mathcal{M}$, any directed path from $X$ to $Y$ would violate the ancestral property, resulting in a contradiction. Similarly, assume that there exists $X \leftrightarrow Y$ in a MAG $\mathcal{M} \in [\mathcal{Q}_\mathbf{X}]$. This configuration would also violate the ancestral property by introducing an almost directed cycle, which leads to a contradiction. $\qquad\square$

**Proposition 8.** *For every MAG $\mathcal{M} \in [\mathcal{Q}_\mathbf{X}]$, if $\mathbf{X}$ is a POMIS relative to $[\![\mathcal{M}, Y]\!]$, then there exists a PMG $\mathcal{Q}_\mathbf{X}^i$ representing $\mathcal{M}$ such that the following conditions are satisfied:*

1. *Every circle mark around $\mathbf{X} \cup \{Y\}$ in $\mathcal{Q}_\mathbf{X}$ is oriented as either a tail ($-$) or an arrowhead ($>$) in $\mathcal{Q}_\mathbf{X}^i$ according to valid local transformations.*

2. *Every $X \in \mathbf{X}$ is an ancestor of $Y$ in $\mathcal{Q}_{\mathbf{X}}^i$.*

3. *$\mathcal{Q}_{\mathbf{X}}^i$ is closed under orientation rules.*

*Proof.* The first and third conditions are satisfied by the soundness and completeness of valid local transformations (Thm. 8). Furthermore, since $\mathbf{X}$ is a POMIS with respect to $[\![\mathcal{P}, Y]\!]$ (thus, $\mathbf{X}$ is a DMIS), the second condition is also satisfied (see Lem. 22), which completes the proof. $\qquad\square$

**Theorem 4** (Characterization of POMIS for PAGs). *A set $\mathbf{X} \subseteq \mathbf{V} \setminus \{Y\}$ is a POMIS relative to $[\![\mathcal{P}, Y]\!]$ if and only if there exists $\mathcal{Q}_{\mathbf{X}}^i$ satisfying Props. 7 and 8 such that $\mathsf{IB}(\mathcal{Q}_{\mathbf{X}}^i, Y, \mathbf{X}) = \mathbf{X}$.*

*Proof.* This follows from the result of Thm. 5. $\qquad\square$

**Theorem 5** (Soundness and completeness). *The algorithm IsPOMIS (Alg. 1) returns True if and only if there exists a MAG $\mathcal{M}$ conforming to $\mathcal{P}$ such that $\mathbf{X}$ is a POMIS relative to $[\![\mathcal{M}, Y]\!]$.*

*Proof.* (IsPOMIS returns True $\Rightarrow \exists \mathcal{G}$ **such that** $\mathsf{IB}(\mathcal{M}, Y, \mathbf{X}) = \mathbf{X}$). Suppose that IsPOMIS returns True. Then, there is a PMG $\mathcal{Q}_{\mathbf{X}}^i$ satisfying $\mathsf{IB}(\mathcal{Q}_{\mathbf{X}}^i, Y, \mathbf{X}) = \mathbf{X}$. We will demonstrate that there exists a MAG $\mathcal{M} \in [\mathcal{Q}_{\mathbf{X}}^i]$ such that $\mathsf{IB}(\mathcal{M}, Y, \mathbf{X}) = \mathbf{X}$ by constructing such a MAG. To do so, consider the following lemma:

**Lemma 35.** *Let $\mathcal{Q}_{\mathbf{X}}^i$ be a PMG in Alg. 1. Let $\mathcal{M}$ be the graph resulting from the following procedure applied to $\mathcal{Q}_{\mathbf{X}}^i$.*

*Step 1. Orient partial directed edges ($\circ\!\!\rightarrow$) as directed edges ($\rightarrow$).*

*Step 2. Consider $A *\!\!\rightarrow B$ in $\mathcal{Q}_{\mathbf{X}}^i$. Let $\mathbf{T}_{\mathcal{M}}^{\mathbf{X}} \triangleq \mathsf{MUCT}(\mathcal{M}, Y, \mathbf{X})$. If $B$ is contained in $\mathbf{T}_{\mathcal{M}}^{\mathbf{X}}$, orient the circle component including $A$ as a DAG where each circle edge involving $A$ in $\mathcal{Q}_{\mathbf{X}}^i$ corresponds to a directed edge outgoing from $A$ in $\mathcal{M}$ (i.e., $A \circ\!\!-\!\!\circ V$ corresponds to $A \rightarrow V$).*

*This step is repeated with the updated $\mathbf{T}_{\mathcal{M}}^{\mathbf{X}} \leftarrow \mathsf{MUCT}(\mathcal{M}, Y, \mathbf{X})$ as long as $\mathcal{M}$ remains unchanged.*

*Step 3. Orient remaining circle component into a DAG with no unshielded colliders.*

*Then, the resulting graph $\mathcal{M}$ is a MAG conforming to $\mathcal{Q}_{\mathbf{X}}^i$.*

*Proof.* The construction follows Lems 6 and 12, and the fact that every circle component can be oriented independently by Lem. 31. $\qquad\square$

Now, we will show that the MAG $\mathcal{M}$ constructed according to Lem. 35 satisfies $\mathsf{IB}(\mathcal{M}, Y, \mathbf{X}) = \mathbf{X}$. Let $X$ be any node in $\mathsf{IB}(\mathcal{Q}_{\mathbf{X}}^i, Y, \mathbf{X})$. Then, $X$ is a parent of some $T_X \in \mathsf{MUCT}(\mathcal{Q}_{\mathbf{X}}^i, Y, \mathbf{X})$ in $\mathcal{Q}_{\mathbf{X}}^i$. By Lem. 24, there exists an uncovered possibly-directed path $T_X \circ\!\!-\!\!\circ \cdots \circ\!\!-\!\!\circ T_X^* \overset{?}{\rightarrow} \cdot \rightarrow \cdots \rightarrow Y$. Due to the balanced property in Lem. 31 a path $X \rightarrow T_X^* \overset{?}{\rightarrow} \cdot \rightarrow \cdots \rightarrow Y$ exists in $\mathcal{Q}_{\mathbf{X}}^i$, which corresponds to $X \rightarrow T_X^* \rightarrow \cdots \rightarrow Y$ in $\mathcal{M}$ by construction (see Step 1). Therefore, we have that for any nodes $X \in \mathbf{X}$, $X$ and $T_X^*$ are included in $\mathsf{An}(Y)_{\mathcal{M}}$. Our goal is to show that $T_X^* \in \mathsf{MUCT}(\mathcal{M}, Y, \mathbf{X})$ since this means $X \in \mathsf{IB}(\mathcal{M}, Y, \mathbf{X})$.

For convenience, we denote $\mathbf{T}_{\mathcal{M}} = \mathsf{MUCT}(\mathcal{M}, Y, \mathbf{X})$ and $\mathbf{T}_{\mathcal{Q}_{\mathbf{X}}^i} = \mathsf{MUCT}(\mathcal{Q}_{\mathbf{X}}^i, Y, \mathbf{X})$. Let $\mathcal{N} \triangleq \mathcal{M}[\mathsf{An}(Y)_{\mathcal{M}}]$ and $\mathcal{H} \triangleq \mathcal{Q}_{\mathbf{X}}^i[\mathsf{PossAn}(Y)_{\mathcal{Q}_{\mathbf{X}}^i}]$. Suppose that $T$ is a node such that $T \in \mathbf{T}_{\mathcal{Q}_{\mathbf{X}}^i} \cap \mathsf{An}(Y)_{\mathcal{M}}$ and $T \in \mathbf{T}_{\mathcal{M}}$. We know such a node exists, as $Y$ is in both $\mathbf{T}_{\mathcal{M}}$ and $\mathbf{T}_{\mathcal{Q}_{\mathbf{X}}^i} \cap \mathsf{An}(Y)_{\mathcal{M}}$.

(① **If** $W \in \mathsf{PC}(T)_{\mathcal{H}[\mathsf{An}(Y)_{\mathcal{M}}]}$, **then** $W \in \mathbf{T}_{\mathcal{M}}$). Suppose that another node $W$ is in the same pc-component of $T$ in $\mathcal{H}[\mathsf{An}(Y)_{\mathcal{M}}]$. This implies that there exists a path between $T$ and $W$ such that (i) all non-endpoint nodes along the path are colliders, and (ii) none of the edges are visible, i.e., $T *\!\!\rightarrow \cdot \leftrightarrow \cdots \leftrightarrow \cdot \leftarrow\!\!* W$ in $\mathcal{H}[\mathsf{An}(Y)_{\mathcal{M}}]$.

For all edges $U \overset{?}{\to} V$ along this path, the edges correspond to directed edges $U \to V$ in $\mathcal{N}$. If there are no circle edges with $U$ in $\mathcal{H}$, the edges remain invisible in $\mathcal{N}$ since orienting a tail mark alone does not introduce any visible edges.

Otherwise, if there are any circle edges $U \circ\!\!-\!\!\circ Z$ in $\mathcal{H}$ that correspond to $U \to Z$ in $\mathcal{N}$, no additional visible edges are introduced. When the edges correspond to $U \leftarrow Z$ in $\mathcal{N}$, $U$ would already have been included in $\mathbf{T}_{\mathcal{M}}$, which in turn ensures that $V$ be included in $\mathbf{T}_{\mathcal{M}}$.

(② **If** $W \in \mathtt{PossDe}(T)_{\mathcal{H}[\mathtt{An}(Y)_{\mathcal{M}}]}$, **then** $W \in \mathbf{T}_{\mathcal{M}}$). This means that there exists an uncovered possibly-directed path from $T$ to $W$ in $\mathcal{H}[\mathtt{An}(Y)_{\mathcal{M}}]$ by Lem. 28. According to our construction, there is a node $S \in \mathbf{T}_{\mathcal{M}}$ (it could be $T$) in the same bucket as $T$ and $W$ such that all nodes in the bucket are descendants of $S$ in $\mathcal{M}$. Since $W \in \mathtt{De}(S)_{\mathcal{M}}$ and $S \in \mathbf{T}_{\mathcal{M}}$, we have $W \in \mathbf{T}_{\mathcal{M}}$.

(① + ②). Thus, we have shown that any node in $\mathbf{T}_{\mathcal{Q}^i_{\mathbf{X}}} \cap \mathtt{An}(Y)_{\mathcal{M}}$ can also be shown to be in $\mathbf{T}_{\mathcal{M}}$, and therefore we can get $T^*_X \in \mathbf{T}_{\mathcal{M}}$.

The remaining task is to prove that $\mathbf{W} \triangleq \mathsf{IB}(\mathcal{M}, Y, \mathbf{X}) \setminus \mathsf{IB}(\mathcal{Q}^i_{\mathbf{X}}, Y, \mathbf{X})$ is empty. For the sake of contradiction, consider any vertex $W \in \mathbf{W}$. Then, there exists a node $T_W \in \mathbf{T}_{\mathcal{M}}$ where $W \in \mathtt{Pa}(T_W)_{\mathcal{M}}$. Note that $T_W \in \mathbf{T}_{\mathcal{Q}^i_{\mathbf{X}}} \cap \mathtt{An}(Y)_{\mathcal{M}}$ holds (see the proof of the reverse direction). If $W \to T_W$ is invisible, then $W$ is included in $\mathbf{T}_{\mathcal{M}}$, leading to a contradiction for $W \in \mathsf{IB}(\mathcal{M}, Y, \mathbf{X})$. If $W \to T_W$ is visible in both $\mathcal{M}$ and $\mathcal{Q}^i_{\mathbf{X}}$, then we can find a visible edge $W \to T^*_W$ satisfying $W \to T^*_W \overset{?}{\to} \cdots \to Y$ in $\mathcal{Q}^i_{\mathbf{X}}$ corresponding to $W \to T^*_W \to \cdots \to Y$ in $\mathcal{M}$ by Lems 24 and 27. This implies $W \in \mathsf{IB}(\mathcal{Q}^i_{\mathbf{X}}, Y, \mathbf{X})$, resulting in a contradiction. If $W \to T_W$ appeared as an invisible edge, either $\circ\!\!-\!\!\circ$ or $\circ\!\!\to$, $W \to T^*_W$ should also appear as an invisible edge by our construction (see Step 2). Therefore, we conclude the proof of the soundness of IsPOMIS.

(IsPOMIS returns False $\Rightarrow \nexists \mathcal{M}$ **such that** $\mathsf{IB}(\mathcal{M}, Y, \mathbf{X}) = \mathbf{X}$). Suppose that $\mathbf{X}$ is a POMIS relative to $[\![\mathcal{M}, Y]\!]$. Then, we have $\mathbf{X} = \mathsf{IB}(\mathcal{M}, Y, \mathbf{X})$. Let $\mathcal{Q}^i_{\mathbf{X}}$ be a PMG representing $\mathcal{M}$. Moreover, we have that $\mathtt{An}(Y)_{\mathcal{M}} \subseteq \mathtt{PossAn}(Y)_{\mathcal{Q}^i_{\mathbf{X}}}$ holds by Lem. 28.

Let $X$ be any variable in $\mathsf{IB}(\mathcal{M}, Y, \mathbf{X})$. Then, $X$ is a parent of some $T_X \in \mathsf{MUCT}(\mathcal{M}, Y, \mathbf{X})$ in $\mathcal{M}$. Furthermore, this appears in $\mathcal{Q}^i_{\mathbf{X}}$ by the construction of IsPOMIS in Alg. 1 (outgoing edges from $X$ are determined in $\mathcal{Q}^i_{\mathbf{X}}$). By Lem. 24, there exists an uncovered possibly-directed path $T_X \circ\!\!-\!\!\circ \cdots T'_X \overset{?}{\to} \cdot \to \cdots \to Y$ in $\mathcal{Q}^i_{\mathbf{X}}$. Due to Lems 20 and 31, the path $X \to T^*_X \to \cdot \to \cdots \to Y$ exists in $\mathcal{Q}^i_{\mathbf{X}}$. Now we will show that $T^*_X \in \mathsf{MUCT}(\mathcal{Q}^i_{\mathbf{X}}, Y, \mathbf{X})$ since this implies $X \in \mathsf{IB}(\mathcal{Q}^i_{\mathbf{X}}, Y, \mathbf{X})$.

Let $\mathbf{T}_{\mathcal{M}} \triangleq \mathsf{MUCT}(\mathcal{M}, Y, \mathbf{X})$ and $\mathbf{T}_{\mathcal{Q}^i_{\mathbf{X}}} \triangleq \mathsf{MUCT}(\mathcal{Q}^i_{\mathbf{X}}, Y, \mathbf{X})$. Let $\mathcal{N} \triangleq \mathcal{M}[\mathtt{An}(Y)_{\mathcal{M}}]$ and $\mathcal{H} \triangleq \mathcal{Q}^i_{\mathbf{X}}[\mathtt{PossAn}(Y)_{\mathcal{Q}^i_{\mathbf{X}}}]$. Suppose that $T$ is a node satisfying $T \in \mathbf{T}_{\mathcal{Q}^i_{\mathbf{X}}} \cap \mathtt{An}(Y)_{\mathcal{M}}$ and $T \in \mathbf{T}_{\mathcal{M}}$. We know such a node exists since $Y$ is in both $\mathbf{T}_{\mathcal{M}}$ and $\mathbf{T}_{\mathcal{Q}^i_{\mathbf{X}}} \cap \mathtt{An}(Y)_{\mathcal{M}}$.

(**If** $W \in \mathbf{T}_{\mathcal{M}}$, **then** $W \in \mathbf{T}_{\mathcal{Q}^i_{\mathbf{X}}} \cap \mathtt{An}(Y)_{\mathcal{M}}$). Since any invisible edges in $\mathcal{M}$ correspond to invisible ones in $\mathcal{Q}^i_{\mathbf{X}}$, we have $W \in \mathsf{PC}_{\mathcal{N}}(T)$ implies $W \in \mathbf{T}_{\mathcal{Q}^i_{\mathbf{X}}} \cap \mathtt{An}(Y)_{\mathcal{M}}$ according to Lem. 30. Furthermore, we know that $W \in \mathtt{De}(T)_{\mathcal{N}}$ implies $W \in \mathtt{PossDe}(T)_{\mathcal{H}[\mathtt{An}(Y)_{\mathcal{M}}]}$ by Lem. 28. Therefore, we get that $W \in \mathbf{T}_{\mathcal{Q}^i_{\mathbf{X}}} \cap \mathtt{An}(Y)_{\mathcal{M}}$. Thus, we have shown that any node in $\mathbf{T}_{\mathcal{M}}$ can also be shown to be in $\mathbf{T}_{\mathcal{Q}^i_{\mathbf{X}}} \cap \mathtt{An}(Y)_{\mathcal{M}}$, and therefore $T^*_X \in \mathbf{T}_{\mathcal{Q}^i_{\mathbf{X}}}$.

The remaining task is to prove that $\mathbf{W} \triangleq \mathsf{IB}(\mathcal{Q}^i_{\mathbf{X}}, Y, \mathbf{X}) \setminus \mathsf{IB}(\mathcal{M}, Y, \mathbf{X})$ is empty. For the sake of contradiction, consider any vertex $W \in \mathbf{W}$. Then, there exists a node $T_W \in \mathbf{T}_{\mathcal{Q}^i_{\mathbf{X}}}$ where $W \in \mathtt{Pa}(T_W)_{\mathcal{Q}^i_{\mathbf{X}}}$. If $W \to T_W$ is invisible in $\mathcal{Q}^i_{\mathbf{X}}$, then $W$ is included in $\mathbf{T}_{\mathcal{Q}^i_{\mathbf{X}}}$, leading to a contradiction for $W \in \mathsf{IB}(\mathcal{Q}^i_{\mathbf{X}}, Y, \mathbf{X})$. If $W \to T_W$ is visible in $\mathcal{Q}^i_{\mathbf{X}}$, it is also visible in $\mathcal{M}$, and we can find a visible edge $W \to T^*_W$ satisfying $W \to T^*_W \to \cdots \to Y$ by Lems 24 and 27. This implies $W \in \mathsf{IB}(\mathcal{M}, Y, \mathbf{X})$, resulting in a contradiction. Therefore, we conclude the proof of the completeness of IsPOMIS. $\qquad\square$

## Broader Impact Statement

This work addresses a structured causal bandit framework that leverages causal knowledge from a Markov equivalence class represented by a PAG. This approach has potential applications in practical settings such as personalized healthcare, adaptive education, and resource-constrained recommendation systems, where a decision-maker aims to make optimal decisions without assuming causal sufficiency (i.e., the absence of unobserved variables), an assumption that is often unrealistic in practice. Therefore, this study takes a step toward the practical application of the framework. However, improper specification of causal structures may lead to misleading conclusions and biased decisions; thus, careful validation and domain-specific causal modeling are essential prior to deployment in high-stakes environments.

