# OpenReview forum: "Structural Causal Bandits under Markov Equivalence"
_NeurIPS.cc/2025/Conference — NeurIPS 2025 poster_

### Official Review · Reviewer_nSMs · 2025-06-26

**Clarity:** 3
**Significance:** 2
**Originality:** 2
**Rating:** 5
**Confidence:** 3

**Summary:**

This paper proposes a general framework for bandit learning when the causal structure of the data generating process is only known up the Markov equivalence class of partial ancestral graphs (PAG) and associated maximum ancestral graphs (MAG). The aim of the work are to extend prior research on the structured causal bandit (SCB) problem to practical applications involving learning from observational data, where causal relations can only be known up to PAG and MAG. The paper contributes by developing the theory for learning the sets of interventions worth exploring in this problem setting, and then devising an algorithm to discern whether such a discovered set is possibly optimal.

**Questions:**

Experimental evaluations:
1. Do you have any evidence or references that 5'000 and 10'000 trials are enough?
2. Are the number of nodes considered relevant for empirical problems? I would expect that practical problems can involve much large graphs (and intervention sets) than those considered here. How would the two proposed methods scale computationally with the size of a graphs and intervention sets? A comment on this would help readers understand the applicability of the work to real data.

I ask for this because, as far as I know, causal discovery is still a research area primarily of theoretical interest. For applied sciences where causal inference methodology is critical and appreciated, like epidemiology, political science, and economics, causal discovery algorithms have yet to really demonstrate a practical usefulness.

This is not to say that the authors work is intrinsically less valuable, but because the problem being studied is explicitly motivated by a lack  of knowledge of the true graph in "practical contexts", it would, at the minimum, be useful to understand when also the authors approach of working with Markov Equivalence classes would provide any tangible progress in real-world settings. Some comments and discussions would be appreciated,

**Ethical Concerns:**

["NO or VERY MINOR ethics concerns only"]

**Final Justification:**

I updated my score from Bordeline Accept to Accept in light of the authors response to my review.

**Limitations:**

Appendix F provides some discussion of limitations and opportunities for future work. The paper could benefit from further discussing the problem that, in practice, the SCM is essentially never known outside of deterministic DGPs or trivial settings. The authors kind of touch on this by noting that assumed SCMs may not fully capture real world systems (first paragraph) and that their work assumes knowledge of the PAG (second paragraph).

However, the main issue underlying the uptake of SCMs for research in applied sciences is that we do not know the true SCM to begin with, and learning it via causal discovery algorithms, even if only up to a PAG or MAG, is to my knowledge not very reliable because causal discovery algorithms tend not to be robust against sampling variability, measurement errors, and other practical problems related to statistical uncertainty in empirical data. Currently, the paper appears to "abstract away" this problem, which is fine for theoretical research. If the work is to be positioned as useful for practice, however, then it should be acknowledged and carefully discussed.

**Paper Formatting Concerns:**

None that I could see.

**Quality:**

3

**Strengths And Weaknesses:**

Strengths:
1. The problem is well-defined and the authors clearly address the problem being studied.
2. The results appear to be complete and novel.

Weaknesses:
3. The paper is positioned as addressing a problem for practical applications, yet does not fully engage with what (I understand) is the main hindrance to the uptake of the class of methods being studied. This is a limitation to its significance.
4. The experimental evaluation does not seem comprehensive. This affects the quality a bit.
5. The writing is technically sound but at times very dense, negatively affecting clarity. I am referring to Sec. 3-4 in particular. The paper could be improved by simplifying the exposition, if possible.
6. The work is incremental, in the sense that it takes a known results and extends them by relaxing an assumption. Please correct me if this is the incorrect interpretation of what the paper does. If my understanding is right, then originality is limited.

Please see Questions for details on weaknesses.

---

> ### Author Rebuttal · Authors · 2025-07-30
>
> > 3. The paper is positioned as addressing a problem for practical applications, yet does not fully engage with what (I understand) is the main hindrance to the uptake of the class of methods being studied. This is a limitation to its significance.
> >
>
> A. We deeply appreciate for your valuable feedback. We will revise this point to better underscore the position of our paper. To clarify, our work aims to provide a theoretical guarantee for the *a priori* elimination of suboptimal arms by leveraging structural knowledge encoded in a PAG, rather than a fully specified causal diagram. This is motivated by the fact that, in the absence of the *causal sufficiency* assumption (i.e., in the presence of unobserved confounders), only a PAG can be obtained from observational data. Accordingly, our framework offers a practically meaningful guideline for action space pruning under more commonly encountered assumptions. This directly addresses a key barrier to the practical uptake of causal bandit methods—-namely, the reliance on a fully specified causal diagram, which is rarely available. By operating under the weaker assumption of partial causal knowledge, our approach significantly broadens the applicability and accessibility of these methods.
>
>
>
> > 5. The writing is technically sound but at times very dense, negatively affecting clarity. I am referring to Sec. 3-4 in particular. The paper could be improved by simplifying the exposition, if possible.
> >
>
> A. We have made efforts to follow the notation used in prior work as closely as possible, particularly those by [Wang et al., 2022, 2023ab, 2024ab], [Zhang 2008ab], and [Jaber et al., 2018, 2022]. That said, we acknowledge that the exposition in Sections 3–4 can be dense, and we will make further efforts to improve clarity, making use of the extra page available in the final version.
>
>
>
> > 6. The work is incremental, in the sense that it takes a known results and extends them by relaxing an assumption. Please correct me if this is the incorrect interpretation of what the paper does. If my understanding is right, then originality is limited.
> >
>
> A. We are happy to emphasize this point. At first glance, one might view our work as a natural extension of existing methods from causal diagrams to PAGs—-for example, by replacing ancestral relations with “possibly ancestral” ones, or substituting the standard do-calculus for causal diagrams [Pearl, 1995] with the do-calculus for PAGs [Jaber at el., 2022]. However, as the saying goes, “*the devil is in the details.”,* extending structural causal bandits from causal diagrams to PAGs presents several nontrivial challenges:
>
> 1. As discussed in Section 3.1 (MIS for PAGs and Its Possible Vacuousness), the standard notion of MIS does not behave as expected in PAGs, necessitating the introduction of *Definitely MIS* (DMIS).
> 2. The naive intuition of simply replacing deterministic relations with “possible” ones fails to capture the correct characterization of POMISs in PAGs. We refer to corresponding challenge in the section “Discussion on Partial Mixed Graphs Obtained from Local Transformation” in Appendix E.2.
> 3. The edges in a PAG are governed by structural constraints and logical dependencies. For example, in the structure $A *\to B \circ  - *C$ , one can deduce $A \ast\to C$ (*balanced property*; shown in Lemma 4). Furthermore, any circle components (composed only with circle edges $\circ - \circ$ ) in a PAG are chordal graph.  These structural entanglements pose significant challenges when proving the soundness and completeness of graphical characterizations in PAGs.
> 4. To obtain solid theoretical result, we need to extend the known result to Partial Mixed Graphs (PMGs) incorporating local transformation (e.g., visibility, partial ancestral relation, balanced properties for the PMGs). We built this point in the “Auxiliary Results” section in Appendix G.
>
> We hope the reviewer appreciates that these challenges partly explain the length and complexity of the proofs provided in our Appendix G and H.
>
>
>
> > Do you have any evidence or references that 5'000 and 10'000 trials are enough?
> >
>
> A. The number of trials is selected such that the cumulative regret with respect to POMIS stabilizes across 1000 repeated runs. Our experimental setup closely follows those of [Lee and Bareinboim, 2018] and [Wei et al., 2023].
>
>
>
> > 4. The experimental evaluation does not seem comprehensive. This affects the quality a bit.
> >
>
> > Are the number of nodes considered relevant for empirical problems? I would expect that practical problems can involve much large graphs (and intervention sets) than those considered here. I ask for this because, as far as I know, causal discovery is still a research area primarily of theoretical interest. For applied sciences where causal inference methodology is critical and appreciated, like epidemiology, political science, and economics, causal discovery algorithms have yet to really demonstrate a practical usefulness.
> >
>
> A. We appreciate your concern regarding the scope of the experimental evaluation. However, we would like to clarify that our work does not focus on causal discovery; rather, it assumes a given PAG as input and centers on the theoretical developments in graphical characterizations. Specifically, our contributions lie in proving the soundness and completeness of the proposed algorithm for identifying POMISs from a PAG. Given this theoretical orientation, the number of nodes does not affect the validity of our results.
>
> For the experiments, we deliberately chose representative PAGs that facilitate reader understanding. The underlying SCMs were constructed using a randomized yet principled procedure to ensure diversity and fairness. We believe these instances serve as meaningful test cases that reflect the theoretical nature of our work, rather than aiming for exhaustive empirical benchmarking.
>
>
>
> > How would the two proposed methods scale computationally with the size of a graphs and intervention sets? A comment on this would help readers understand the applicability of the work to real data.
> >
>
> A. Thank you for your comment about complexity. Identifying all POMIS sets requires checking all subsets of $\mathbf{V} \setminus \lbrace Y \rbrace$ using IsPOMIS (Alg. 1), and thus the size of the search space grows exponentially. However, since all non-DMIS sets are filtered out, the enumeration process effectively depends only on the number of DMISs, i.e. $\mathcal{O}(2^d)$ where $d$ denotes the number of DMISs.
>
> Moreover, although [Lee and Bareinboim, 2018] provided an efficient algorithm for enumerating all POMISs in a causal diagram by leveraging a topological order between $Y$ and other variables, such an approach is *not* applicable to PAGs, where the topological order is not determined; this fundamental challenge with respect to *undetermined* topological order in PAGs lead to exponential complexity. We would like to note that discussion on complexity is shown in footnote 4 in page 8, and further detailed discussions are addressed in the “Discussion on Complexity of Enumerating POMISs for PAG.” section in Appendix E.2.
>
>
>
>
> > However, the main issue underlying the uptake of SCMs for research in applied sciences is that we do not know the true SCM to begin with, and learning it via causal discovery algorithms, even if only up to a PAG or MAG, is to my knowledge not very reliable because causal discovery algorithms tend not to be robust against sampling variability, measurement errors, and other practical problems related to statistical uncertainty in empirical data. Currently, the paper appears to "abstract away" this problem, which is fine for theoretical research. If the work is to be positioned as useful for practice, however, then it should be acknowledged and carefully discussed.
> >
>
> A. We appreciate your thoughtful comment. Our work does not focus on causal discovery itself, and thus assumes access to a correct PAG as a starting point—-an assumption commonly made in this line of research (e.g., adjustment in PAGs [Wang et al., 2023a], causal effect identification, and do-calculus for PAGs [Jaber et al., 2022]). However, we fully agree that in practical applications, the reliability of causal discovery algorithms is limited due to statistical uncertainty, measurement noise, and sampling variability. Since our method builds upon a given PAG, we acknowledge that this limitation deserves more explicit discussion.
>
> While we referred this point in Lines 1593—1595 of the section Broader Impact Statement—-“Improper specification of causal structures may lead to misleading conclusions and biased decisions; thus, careful validation and domain-specific causal modeling are essential prior to deployment in high-stakes environments”—-we agree that a more direct discussion of this point would strengthen the paper. In the camera ready version, we will add a dedicated note in the limitations section to clarify that the reliability of the input PAG is critical for safe and effective application, and that care must be taken. Thank you again for your constructive suggestions. Your feedback will strengthen our work.

---

> > ### Comment · Reviewer_nSMs · 2025-08-02
> >
> > I have read the authors response and appreciate their thorough explanationas. It addressed my most pressing questions. I will increase my rating of the submission by one point as a result.

---

> > > ### Author Response · Authors · 2025-08-06
> > >
> > > We sincerely appreciate your pointed comments, which provided an opportunity to further clarify our key contributions. Your insightful feedback will be reflected in the final version of the paper.

---

### Official Review · Reviewer_FQk8 · 2025-06-30

**Clarity:** 3
**Significance:** 2
**Originality:** 2
**Rating:** 4
**Confidence:** 4

**Summary:**

This paper studies the causal bandit problem under do interventions and unrestricted structural causal models. In this setting, many multi-node interventions (i.e., arms of the causal bandit) are equivalent since do interventions break the causal paths that start upstream and pass through the intervened node to affect the reward. This notion is previously formalized under (possibly-optimal) minimal intervention sets (PO)MISs, for causally sufficient models. The whole focus of this paper is on generalizing this concept to ancestral graphs. Specifically, characterizing and determining the conditions of a set being POMIS given a partial ancestral graph.

**Questions:**

**Questions about the setting and positioning of the paper**:
1. MIS-POMIS notions are strictly for causal bandits with *do-hard* interventions. Many papers on causal bandits, including some of the first ones, indeed use do interventions. However, many of the more recent papers, including some of the papers that are covered in the related work discussion, focus on ``soft intervention’’. To clarify, in the soft intervention case, the downstream effect of an intervention is not cancelled out by the other interventions on the causal path to the reward node. In the next version of the paper, I strongly suggest that soft interventions to also be discussed (at least briefly), and the distinction/positioning of the paper should be clearer.
2. Causal bandits with unknown graphs: Discussion on this subject is incomplete. To my knowledge, Bilodeau et al. (2022) (which is already cited), de Kroon et al. (2022), and Malek et al. (2023) have some results for unknown graph structures as well. More recently, Yan et al. (2024b) studied unknown causal graphs with soft interventions (under a linear model).

This is not really a weakness, but more of a comment/question. The theoretical nature of the paper is quite involved. I’m mostly familiar with this line of work, and still, I found it difficult to follow since it requires calling many tools and notions from the literature---which are inevitably mostly relegated to the appendix. In this regard, I wonder whether a journal would have been a more appropriate venue for this paper to be hosted, reviewed more carefully, etc.

**Experiments**:
As discussed in Appendix F, one limitation of the current work is that it still requires the partial ancestral graph as input. It would have been interesting to see how robust the proposed algorithm (and the gains w.r.t. the baselines) is under slight perturbations of the input PAG, e.g., a couple of misspecified edges, which is a realistic setting for practical usage. For instance, Yan et al. (2024a), Peng et al. (2025), and Varici et al. (2023) demonstrate some empirical results for robustness and/or misspecifications.

**Minor points:**
* For example, Rule 3 of do-calculus for MAGs is given in the Appendix, but its understanding is crucial for the main body. In particular, the possible vacuousness of MIS for PAGs is explained through this rule. Hence, it’d be better to make the rule clear in the main paper. This is related to my broader point that, in my opinion, it’s difficult to fit the content of this paper into a 9-page conference format.
* Footnote 4: PAGs have “partially ordered” topological orders, right?

**References**
- Malek, A., Aglietti, V., & Chiappa, S. (2023). Additive causal bandits with unknown graph. ICML 2023
- De Kroon, A., Mooij, J., & Belgrave, D. (2022). Causal bandits without prior knowledge using separating sets. CLeaR 2022
- Yan, Z., Mukherjee, A., Varıcı, B., & Tajer, A. (2024a). Robust causal bandits for linear models. IEEE Journal on Selected Areas in Information Theory.
- Yan, Z., & Tajer, A. (2024b). Linear Causal Bandits: Unknown Graph and Soft Interventions.  NeurIPS 2024
- Peng, C., Zhang, D., & Mitra, U. (2025). Asymmetric graph error control with low complexity in causal bandits. IEEE Transactions on Signal Processing.

**Ethical Concerns:**

["NO or VERY MINOR ethics concerns only"]

**Final Justification:**

As I stated in my initial review, the paper has a clear (albeit niche) problem and solves it rigorously. I find this problem/solution possibly interesting to the causal bandits community, which is a clear positive factor in my evaluation. On the other hand, technical novelty is limited (not at the level of 'very clear accept' threshold for me). Also, the revision should carefully address the valid concerns of reviewer UG9x regarding the utility of the proposed approach compared to Elahi et al. (2024). To be clear, I understand, settings are different (offline vs. online causal discovery stages), though the utility of the current setting (offline discovery, taking PAG as input) is not very clearly conveyed, beyond the appreciation of the technical problem.

Overall, I remain slightly positive about this paper, and maintain my score (4).

**Limitations:**

Yes.

**Paper Formatting Concerns:**

No.

**Quality:**

3

**Strengths And Weaknesses:**

**Strengths:** The paper has a clear problem---generalizing (possibly-optimal) minimal intervention sets (PO)MISs, which were previously proposed and studied under causal sufficiency, to ancestral graphs. This is a niche yet interesting problem that is unexplored to my knowledge. While the paper does not introduce new tools for solving this problem (if I'm not missing), still, the solution steps and progression of the results are rigorous.

**Weaknesses:** While related work discussion is mostly good, I think the positioning of the paper is missing some important clarifications on the intervention model (and relatedly, some recent references are missing). See my detailed comments below the questions.

---

> ### Author Rebuttal · Authors · 2025-07-30
>
> > MIS-POMIS notions are strictly for causal bandits with *do-hard* interventions. Many papers on causal bandits, including some of the first ones, indeed use do interventions. However, many of the more recent papers, including some of the papers that are covered in the related work discussion, focus on ``soft intervention’’. To clarify, in the soft intervention case, the downstream effect of an intervention is not cancelled out by the other interventions on the causal path to the reward node. In the next version of the paper, I strongly suggest that soft interventions to also be discussed (at least briefly), and the distinction/positioning of the paper should be clearer.
> >
>
> > Yan et al. (2024b) studied unknown causal graphs with soft interventions (under a linear model).
> >
>
> A. We sincerely appreciate for pointing this out. We would like to refer to the sentence in Lines 44–45: “[Lee and Bareinboim, 2020] and [Everitt et al., 2021] established SCM-MAB with stochastic policies, and [Carey et al., 2024] studied the completeness of its graphical characterization.” To elaborate further, [Lee and Bareinboim, 2020] introduced the concept of *Possibly Optimal Mixed Policy Scope* (POMPS), a soft-intervention counterpart to POMIS, along with its graphical characterization. Similarly, [Everitt et al., 2021] proposed a related framework in the context of the *influence diagrams* literature. Subsequently, [Carey et al., 2024] studied the completeness of POMPS characterization, although this still remains an open problem.
>
> We will revise this part to improve clarity in related works (Appendix A) of the final version with additional reference [Yan et al., 2024b]. Thank you again for your insightful comment.
>
> - Yan, Z., and Tajer, A., Linear Causal Bandits: Unknown Graph and Soft Interventions., NeurIPS (2024b).
>
> > Causal bandits with unknown graphs: Discussion on this subject is incomplete. To my knowledge, Bilodeau et al. (2022) (which is already cited), de Kroon et al. (2022), and Malek et al. (2023) have some results for unknown graph structures as well. More recently,
> >
>
> A. [Bilodeau et al., 2022] and [de Kroon et al., 2022] assumed *causal sufficiency*---that is, the absence of unobserved confounders and thus no bidirected edges in the graph. While [Malek et al., 2023] also provided some results for settings with unknown graph structures, the authors initially highlight the challenge posed by the exponentially large number of arms in causal bandit problems under unknown graphs, and they assumed no confounding exists between the reward variable $Y$ and its ancestors. We will also append the works into related works (Appendix A), discussing on comparison with this line of work.
>
> - De Kroon, et al., Causal bandits without prior knowledge using separating sets., CLeaR (2022).
> - Malek, A., et al., Additive causal bandits with unknown graph., ICML (2023).
>
> > This is not really a weakness, but more of a comment/question. The theoretical nature of the paper is quite involved. I’m mostly familiar with this line of work, and still, I found it difficult to follow since it requires calling many tools and notions from the literature---which are inevitably mostly relegated to the appendix. In this regard, I wonder whether a journal would have been a more appropriate venue for this paper to be hosted, reviewed more carefully, etc.
> >
>
> A. We appreciate your thoughtful comment. We fully agree that the theoretical nature of the paper demands a substantial background and inevitably involves many technical tools, some of which are deferred to the Appendix due to space constraints.
>
> We chose to submit this work to NeurIPS because we believe the conference offers a valuable venue for early dissemination and active discussion with a community of experts in related areas, including causal inference [Jaber et al., 2022], causal bandits [Lee and Bareinboim 2018, 2020; Wei et al., 2023; Elahi et al., 2024], and graphical models [Wang et al., 2022]. In particular, there might be audiences who are actively seeking the solutions addressed in our work, and we hope to contribute meaningfully to these ongoing conversations.
>
> > It would have been interesting to see how robust the proposed algorithm (and the gains w.r.t. the baselines) is under slight perturbations of the input PAG, e.g., a couple of misspecified edges, which is a realistic setting for practical usage.
> >
>
> A. This is an insightful suggestion. We agree that it is an important direction. One possible approach could involve checking the compatibility of observed data with the given PAG while pulling POMIS arms, thereby identifying inconsistencies that may hint at structural misspecification. This idea resonates with hybrid strategies that combine exploration for model validation (e.g., $\epsilon$-greedy mechanisms) with regret minimization via UCB or Thompson Sampling. Alternatively, one could consider scenarios where multiple PAG candidates are available. In such cases, pulling POMISs from each candidate—possibly with appropriate weighting on overlapping arms—might allow the agent to gradually place higher confidence on the true PAG. However, this strategy leans more toward *causal discovery* than regret minimization and falls outside the primary scope of our contribution.
>
> We would like to note that our algorithm relies entirely on the partial information encoded in a PAG when determining POMISs. Importantly, the edges in a PAG are not independent; rather, they are governed by structural constraints and logical dependencies. For example, in the structure $A *\to B \circ - *C$ , one can deduce $A \ast\to C$ (*balanced property*; shown in Lemma 4). Due to the structured entanglements, it is indeed difficult to expect that computing POMISs on an incorrect PAG would lead to robust results. We will include this limitation into the Limitation section in the final version.
>
> > For example, Rule 3 of do-calculus for MAGs is given in the Appendix, but its understanding is crucial for the main body. In particular, the possible vacuousness of MIS for PAGs is explained through this rule. Hence, it’d be better to make the rule clear in the main paper.
> >
>
> A. This is an excellent suggestion. We will add do-calculus for MAGs and PAGs (Theorem 6 in Appendix B.1) into main body, utilizing an extra page in the final version.
>
> > Footnote 4: PAGs have “partially ordered” topological orders, right?
> >
>
> A. Yes, that is correct. PAGs have partial topological order among circle components (Lemma 17 in Appendix B.2.3).

---

> ### Comment · Reviewer_FQk8 · 2025-08-04
>
> Thank you for the rebuttal. I have also read the other reviews and authors' rebuttals. I maintain my moderately positive judgement and score.
>
> Two minor clarifications for my earlier comments:
> 1. **PAG misspecification**: I understand that the focus of paper entirely disregards causal discovery component. My suggestion for considering misspecified PAGs is more on the empirical side. As also mentioned by Reviewer nSMs, one repeated talking point of the paper is “practical motivation”. In this sense, it’s fair to assume that algorithm will almost never have access to the *true* PAG. Because, to my knowledge, existing causal discovery algorithms all are far from perfect in non-trivial real-data benchmarks. This would be especially true for constrained-based algorithms like FCI, where the exact PAG recovery would require many empirical conditional-independence test to work perfectly. Due to this reason, it’s of great practical interest to study (at least experimentally) how minimal intervention sets (and in turn, empirical regret) are affected given PAG misspecification.
> 2. **Soft interventions**: Thank you for the references and discussion. I just want to note that I was referring to a slightly different formulation, as covered in Yan et al. and others. To be clear, in the fully unconstrained soft intervention setting, none of the intervention sets would be redundant, which makes a graphical characterization impossible. Thus, I appreciate the references and the valuable discussion, but I just note that POMPS characterization is not what I was referring to.

---

> > ### Author Response · Authors · 2025-08-06
> >
> > We appreciate your valuable feedback. We deeply agree with your comments regarding the importance of causal discovery and the references to soft interventions. These points will strengthen the final version.

---

### Official Review · Reviewer_UG9x · 2025-07-01

**Clarity:** 3
**Significance:** 1
**Originality:** 3
**Rating:** 2
**Confidence:** 3

**Summary:**

The paper addresses a multi-armed bandit setting in which actions and rewards form a causal graph. The authors consider a scenario where the causal graph is known only up to its Markov equivalence class, in contrast to prior work that assumes the graph is fully known. The goal of the paper is to preprocess and identify the actions (i.e., interventions) that could be optimal given information. This approach can significantly reduce the action space.

**Questions:**

1- Could you elaborate on the point noted in the section above?

2- As you mentioned in the related work, Elahi et al. [2024] consider the causal bandit problem with an unknown graph. How does your approach (i.e., finding the POMIS set in the MAG and running KL-UCB) compare to theirs, both theoretically and practically?

3- Can you provide an analysis of how much the action space is reduced? For example, could you quantify this for a specific graph structure if a general result is not possible?

**Ethical Concerns:**

["NO or VERY MINOR ethics concerns only"]

**Final Justification:**

The paper aims to characterize the best possible optimal arms in PAGs. By identifying this set of possible arms, the arm set size can be significantly reduced. However, the main issues with the paper are:
i) they do not provide any non-trivial algorithm for the online phase,
ii) they do not present any regret bound comparable to previous work, and
iii) more generally, there is no clear justification for why this pruning in the offline phase is necessary. There might be alternative approaches that could lead to better regret. For example, ideas similar to those proposed by Elahi et al. [2024].

I discussed these concerns with the authors, but their responses did not convince me.

**Limitations:**

Yes.

**Quality:**

3

**Strengths And Weaknesses:**

The paper presents solid theoretical results and provides a graphical characterization of potentially optimal actions within Markov equivalence classes. However, I have some concerns about the effectiveness of the proposed method in reducing cumulative regret. While I agree that the action space can be significantly reduced, this does not necessarily lead to optimal regret performance. For example, since arbitrary interventions are allowed, why not attempt to dynamically learn the graph structure while optimizing regret? In that case, the actions removed during the preprocessing phase might actually be beneficial. Alternatively, is it possible to formally show that the removed actions are uninformative?

---

> ### Author Rebuttal · Authors · 2025-07-30
>
> > While I agree that the action space can be significantly reduced, this does not necessarily lead to optimal regret performance. For example, since arbitrary interventions are allowed, why not attempt to dynamically learn the graph structure while optimizing regret? In that case, the actions removed during the preprocessing phase might actually be beneficial.
> >
>
> A. We are grateful for sharing your insightful feedback. Discovering graphical structures is a core line of research in the causal inference literature, referred to *causal discovery.* To clarify our focus, we need to provide representative studies that aim to causal discovery from interventions.
>
> **Offline causal discovery from interventions.** In the offline or non-adaptive setting, interventions are predetermined before algorithm execution. [Hauser and Bühlmann, 2012] studied the problem of learning graph structures from interventions under the assumption of no unobserved confounders, while [Kocaoglu et al., 2017] explored experimental design for learning causal diagrams from interventions. Recently, [Zhou et al., 2025] investigated learning PAGs from interventions.
>
> **Online causal discovery from interventions.** While those offline causal discovery researches require access to infinite interventional samples, there has been intensive works that *adaptively* selects interventions from an online learning. [Squireset al., 2020; Choo and Shiragur, 2023] applied interventions sequentially, with adaptively chosen targets at each step, still necessitating access to interventional distributions (i.e., an infinite number of interventional samples). Although [Greenewald et al. 2019;  Elahi et al., 2024b] worked with finite interventions, it is applicable only when the underlying causal structure has no unobserved confounders. Notably, designing adaptive discovery algorithms that work with finite interventions *and* allow for unobserved confounders remains an open problem.
>
> |  | cover unobserved confounder | online discovery | Interventional sample efficiency |
> | --- | --- | --- | --- |
> | [1] Hauser and Bühlmann, 2012 | x | x | x |
> | [2] Kocaoglu et al., 2017 | o | x | x |
> | [3] Greenewald et al. 2019 | x | o | o |
> | [4] Squires et al., 2020 | x | o | x |
> | [5] Choo and Shiragur, 2023 | x | o | x |
> | [6] Elahi et al., 2024b | x | o | o |
> | [7] Zhou et al., 2025 | o | x | x |
>
> It may possible to incorporate online causal discovery into the decision-making process.  For example, at each step, an agent may choose interventions aimed at improving structural knowledge, while also expecting that those arms could be valuable for minimizing regret. However, designing algorithms that effectively balance exploration for structure learning and for regret minimization poses substantial additional challenges, as these two objectives—-structure discovery and regret minimization—are not naturally aligned.
>
> While we acknowledge this as an interesting direction, our main contribution lies in providing a sound and complete characterization of optimal action selection given a PAG, which is obtained in the offline phase without interventions. We believe this foundation can complement future work on integrating online structure refinement where beneficial.
>
> - [1] Hauser, Alain, and Peter Bühlmann., Characterization and greedy learning of interventional Markov equivalence classes of directed acyclic graphs., JMLR (2012)
> - [2] Kocaoglu, Murat, Karthikeyan Shanmugam, and Elias Bareinboim., Experimental design for learning causal graphs with latent variables., NeurIPS (2017).
> - [3] Greenewald, K., et al., Sample efficient active learning of causal trees., NeurIPS (2019).
> - [4] Squires, C., et al., Active structure learning of causal dags via directed clique trees., NeurIPS (2020).
> - [5] Choo, D. and Shiragur, K. Adaptivity complexity for causal graph discovery. arXiv preprint (2023).
> - [6] Muhammad Qasim Elahi, et al., Adaptive Online Experimental Design for Causal Discovery., ICML(2024b).
> - [7] Zhou, Zihan, Muhammad Qasim Elahi, and Murat Kocaoglu., Characterization and Learning of Causal Graphs from Hard Interventions., arXiv preprint (2025).
>
> > Alternatively, is it possible to formally show that the removed actions are uninformative?
> >
>
> A. From the perspective of regret minimization, arms that are not POMIS are clearly uninformative by the definition of POMIS (Definition 4). Accordingly, our algorithm (Alg. 1) ensures that only suboptimal or redundant arms are pruned.
>
>
> > As you mentioned in the related work, Elahi et al. [2024] consider the causal bandit problem with an unknown graph. How does your approach (i.e., finding the POMIS set in the MAG and drunning KL-UCB) compare to theirs, both theoretically and practically?
> >
>
> A. We deeply appreciate your pointed comment, which helps us highlight an important distinction between our work and [Elahi et al., 2024].
>
> **Theoretical comparison.** As you noted, suboptimal arms (i.e., non-POMIS arms) can be useful for learning the underlying causal graph structure—-a direction explored by [Elahi et al., 2024]. To briefly summarize their approach:
>
> (1) The first phase learns the induced subgraph of the ancestors of the reward node $Y$ in an online manner, by interventions over $\mathbf{V} \setminus \lbrace Y \rbrace$; (2) Using the learned graph, they identify POMISs following the method of [Lee and Bareinboim, 2018]; (3) Finally, they run standard UCB with the identified POMISs.
>
> Their key contribution lies in (1) learning the graph until it is sufficient to identify POMISs; particularly uncovering ancestral relations and latent confounders using interventions. However, they acknowledge that sample-efficient learning of causal diagrams with latent confounders without any assumptions is challenging. To address this, their method assumes certain testable conditions, such as probability gaps among variables.
>
> In contrast, our work does not focus on causal discovery, but instead assumes access to a PAG (e.g., obtained via FCI algorithm [Spirtes et al., 2001b; Zhang, 2008a]) and aims to identify POMISs directly from a PAG input. That is, given a PAG from purely observational data, our algorithm prunes suboptimal arms *a priori*, without requiring any interventional data for causal discovery.
>
> In summary, while [Elahi et al., 2024] focus on discovering a subgraph sufficient to identify POMISs, our work addresses the problem of identifying POMISs given a PAG.
>
> **Practical comparison.** From a practical perspective, interventional (experimental) data is often more costly and risk-prone than observational data. This suggests that our approach first discovering a PAG from observational data and then identifying POMISs may offer substantial advantages in resource-constrained or high-risk domains, compared to methods that rely on extensive intervention for graph discovery.
>
> > Can you provide an analysis of how much the action space is reduced? For example, could you quantify this for a specific graph structure if a general result is not possible?
> >
>
> A. Thank you for the opportunity to emphasize how significantly the action space can be reduced. We first note that, in general, there is no known method to precisely quantify the number of arms based solely on the graph structure. However, in specific cases---such as when (1) the system is assumed to be Markovian or when there are no unobserved confounders between $Y$ and its ancestors; and (2) given the corresponding causal diagram---it is known that the set of parents of $Y$ is the only POMIS (Proposition 2 and Corollary 3 in [Lee and Bareinboim, 2018]. In such cases, the number of arms reduces to the cardinality of $Pa(Y)$.
>
> Since the number of POMIS arms depends on the graphical instance, we provide the example our paper addressed. Let us consider the PAG shown in Fig. 9. Given the PAG with 10 nodes, an agent can consider interventions on arbitrary combinations of nodes $\mathbf{V} \setminus \lbrace Y \rbrace$.  The number of possible intervention sets is $2^9 = 512$ (i.e., the number of subsets of $\mathbf{V} \setminus \lbrace Y \rbrace$), and the total number of corresponding arms is $\sum_{i=1}^9
> \binom{9}{i}K^i = (K+1)^9 = 19683$ with binary domains $K=2$.  The number of POMISs is 8, with a total of 54 corresponding arms. Therefore, it suffices to consider only $\frac{54}{19 683} \approx 0.27 $\% of the total arms. The complete counts of intervention sets and their corresponding arms, for both the main and additional experiments, are provided in Table 2 in Appendix D.
>
> |  | Task1 (Fig. 5a) | Taks2 (Fig. 8) | Taks3 (Fig. 9) | Task4 (Fig. 11a) | Task5 (Fig. 11b) | Task6 (Fig. 11c) |
> | --- | --- | --- | --- | --- | --- | --- |
> | POMIS arms | **19 (23.5%)** | **89 (36.6%)** | **54 (0.27%)** | **15 (6.17%)** | **81 (3.7%)** | **231 (10.7%)** |
> | DMIS arms | 25 | 195 | 2025 | 57 | 189 | 1755 |
> | Total arms | 81 | 243 | 19683 | 243 | 2187 | 2187 |
>
> where the percentages indicate the ratio $\frac{\text{The number of POMIS arms}}{\text{The number of Total arms}} \times 100$, and tasks 4—6 are in Fig. 11 in Appendix D. All POMISs of our experiments can be found in pages 25-26.

---

> > ### Comment · Reviewer_UG9x · 2025-08-02
> >
> > Thanks to the authors for their response, and specifically for illustrating the reduction of the action space across different graphs. However, I still have concerns about the first two questions.
> >
> > **Regarding the causal discovery literature**
> >
> > Thank you for reviewing the relevant literature, most of which I was already familiar with. It would be helpful to include some of these works in the "Related Work" section of the paper.
> >
> > > our main contribution lies in providing a sound and complete characterization of optimal action selection given a PAG, which is obtained in the offline phase without interventions.
> >
> > This is exactly my concern: why should this be done in the offline phase? I agree that it reduces the action space, but is this approach optimal? If the goal is to use the UCB-based method and disregard the causal structure thereafter, your approach makes sense. However, I’m not convinced this is the only viable strategy.
> >
> > > From the perspective of regret minimization, arms that are not POMIS are clearly uninformative by the definition of POMIS (Definition 4). Accordingly, our algorithm (Alg. 1) ensures that only suboptimal or redundant arms are pruned
> >
> >
> > I understand that your algorithm prunes suboptimal actions, but my point is that a suboptimal action might still be informative for learning the causal structure more efficiently—even if it temporarily increases regret. If you disagree, I would appreciate a demonstration or explanation. This would support the case for safely using your algorithm to prune actions in the offline phase and then focusing on others in the online phase.
> >
> >
> > **Regarding the comparison to Elahi et al. [2024]**
> >
> > Thank you for comparing your method to Elahi et al. [2024]. I agree that their algorithm operates in an online fashion and aims to minimize regret. However, in the end, the objective is to use the causal structure of actions to minimize regret. My question remains: how does your algorithm (possibly combined with an online phase) compare to Elahi et al. [2024] in terms of the regret upper bound? I find it hard to evaluate the effectiveness of your approach without empirical or theoretical comparisons.
> >
> >
> > **Regarding costly interventions**
> >
> > I agree that interventions are costly in many applications. However, in an online setting, interventions occur in every round. So I don't think it's entirely accurate to say that Elahi et al. [2024] rely on costly interventions while your method does not, simply because you don't enter an online phase.

---

> > > ### Author Response · Authors · 2025-08-04
> > >
> > > > Regarding causal discovery
> > > >
> > >
> > > A. Our algorithm extracts the relevant information that can be obtained from a PAG, *prior to* any interaction. Beyond this point, no additional causal knowledge is utilized in online interaction—-this is evident from the fact that we simply plug our results into a standard independent-arm bandit algorithms, as in [Lee and Bareinboim, 2018; Elahi et al., 2024].
> > >
> > > **Optimal strategy.** In the case of structural causal bandits, arms may be correlated, and each intervention can provide some information about others. [Wei et al., 2023] proposed an approach that leverages these correlations to accelerate learning. However, a key limitation of this line of work is that it assumes access to the true causal diagram, and requires full parameterization of the underlying SCMs [Zhang et al., 2022], which often incurs significant computational overhead and limits its scalability. As noted by [Elahi et al, 2024], this approach is infeasible for larger or denser causal graphs, and exploiting them effectively remains an open problem.
> > >
> > > To summarize, while we make full use of the information immediately obtainable from the PAG structure in the offline phase, we believe there is room to develop optimal strategies that incorporate online updates, which remains a promising direction for future work.
> > >
> > > - Zhang et al., Partial Counterfactual Identification from Observational and Experimental Data, ICML (2022)
> > >
> > > **Online PAG update.** [Wang et al., 2023b] adaptively refine a PAG by resolving circle marks through targeted interventions. In this sense, interventions on nodes involved in circle marks can be useful for structural refinement. However, as noted in [Wang et al., 2024], obtaining a closed-form characterization of the number of MAGs compatible with a PAG—-given a particular orientation of a circle mark—- remains an open problem, implying that determining which circle marks to prioritize for learning is itself a challenging problem. As such, designing reliable algorithms that exploit structural uncertainty during learning involves solving nontrivial structure learning problems—-closely related to causal discovery—-and remains an active area of research.
> > >
> > > > Regarding the comparison to Elahi et al., [2024]
> > > >
> > >
> > > A. Since both our method and that of [Elahi et al., 2024] ultimately reduce the problem to standard bandits over a set of POMISs, their regret bounds depend critically on the size of the resulting POMIS set. Suppose we plug in the standard UCB algorithm in both cases. Then,
> > >
> > > (1) For [Elahi et al., 2024], the regret is upper-bounded as:
> > >
> > >  $Reg_T \leq Kn \max(\frac{8}{\epsilon^2},\frac{8}{\gamma^2}) \log{\frac{4n^2K^2}{\delta}}+\frac{8}{\epsilon^2}\log{\frac{4nK^2}{\delta}}+8\alpha d_{\max}(KA \vert An(Y)\vert+\max(B,C))\log(\vert An(Y)\vert) + c\sum_{\mathbf{x} \in \mathcal{I}\_{\mathcal{G}}}\Delta_{\mathbf{x}}(1+ \frac{\log T}{\Delta^2_{\mathbf{x}}})$
> > >
> > > where $n$ is the number of nodes, $K$ cardinarity, $\Delta_{\mathbf{x}}$ suboptimality gap of $do(\mathbf{x})$, $d_{\max}$ a constant greater than the maximal in-degree in the true causal diagram $\mathcal{G}$ with $c >0$. The first three terms arise from the regret incurred during the online causal discovery phase. The final term corresponds to the standard UCB regret after identifying the POMISs from the discovered $\mathcal{G}$ (i.e., $\mathcal{I}_{\mathcal{G}}$). See Appendix A.13 of [Elahi et al., 2024] for further details.
> > >
> > > (2) For ours: $Reg_T \leq c\sum_{\mathbf{x} \in \mathcal{I}\_{\mathcal{P}}}\Delta_{\mathbf{x}}(1+ \frac{\log T}{\Delta^2_{\mathbf{x}}})$
> > >
> > > which depends only on the size of POMIS from PAG $\mathcal{I}_{\mathcal{P}}$. Since our approach avoids online discovery, our regret does not include the additional discovery terms. However, since PAGs contain structural uncertainty (e.g., circle marks), the number of POMISs from a PAG is typically larger than that from a causal diagram. As a result, the UCB regret term (the last term in (1)) is usually smaller than our total regret (2). Therefore, while a direct comparison is difficult due to the different settings, we note that the trade-off is between eliminating online discovery cost and accepting a larger initial action space.
> > >
> > > > Regarding costly interventions
> > > >
> > >
> > > A. What we intended to emphasize is that [Elahi et al., 2024] performs discovery through interventions during the online phase, which requires actively exploring the entire action space and may incur a large number of costly trials. On the other hand, our method assumes access to the true PAG inferred from prior observation and leverages it to prune the action space before any online interaction begins. We do not claim that our approach is universally superior to theirs. Rather, our point is that their method is fully online and involves costly interventions throughout, while ours exploits prior graphical knowledge (PAG) in an offline manner. We will add all of these valuable discussions into the final version.

---

> ### Author Response · Authors · 2025-08-08
>
> Dear Reviewer UG9x
>
> We would like to kindly ask whether our response has addressed your concerns. In particular, we have carefully addressed your comments regarding adaptive causal discovery and have highlighted that our work focuses on pruning the action space in an offline manner, using only a PAG. In addition, we have further clarified the distinction from [Elahi et al., 2024]’s fully online approach through a comparison of the regret bounds.
>
> We sincerely hope that our clarifications were helpful in resolving your concerns and look forward to your response.
>
> Best,
>
> Authors 26106

---

> > ### Comment · Reviewer_UG9x · 2025-08-08
> >
> > Dear Authors,
> > Thank you for your response. I have read it and I do not have any further questions.

---

### Official Review · Reviewer_QaCD · 2025-07-02

**Clarity:** 2
**Significance:** 3
**Originality:** 3
**Rating:** 5
**Confidence:** 1

**Summary:**

The paper considers structural causal bandits.  It relaxes the assumption that a structural causal model must be provided to the agent apriori. Instead, it assumes that a partial ancestor graph is known.  The main contribution is an analysis with a corresponding algorithm to identify possibly optimal minimal intervention sets (POMISs).  This reduces the number of arms that need to be explored to find the the optimal arm.

**Questions:**

* Is there a way to explain the approach so that the average ML researcher without much causality background could apply the proposed approach?
* What are the MIS and POMIS in each task and how is the count derived for the number of arms to be explored in each approach in the experiments?

**Ethical Concerns:**

["NO or VERY MINOR ethics concerns only"]

**Final Justification:**

My evaluation is really a guess since I lack familiarity with the topic.  While I recommend acceptance, I will not object to rejection based on review by UG9x.

**Limitations:**

The paper describes limitations in the appendix.

**Paper Formatting Concerns:**

No concern

**Quality:**

4

**Strengths And Weaknesses:**

Let me start by acknowledging that I am not an expert in structural causal bandits.  I lack background in causality to fully understand the details of this paper.  Hence, this evaluation is really a guess.

Strengths:
* Strong theoretical contribution
* The first approach (to my knowledge) to identify POMISs
* A practical approach to reduce the amount of exploration in structural causal bandits

Weaknesses:
* Since most ML practitioners may not be familiar with partial ancestral graphs (PAGs), it is not clear that practitioners will be able to provide a PAG as input.  This may or may not be a weakness of the approach if practitioners need to be educated.  Nevertheless, specifying a PAG should require less knowledge than specifying a structural causal model.
* I found the paper difficult to read because of the high number of definitions, propositions and theorems before we get to the main contribution.  Again, I acknowledge that I don't have sufficient background in causality.  However, suppose that a practitioner wanted to use your approach.  Is there a way to explain the approach so that the average ML researcher without much causality background could apply the proposed approach?

---

> ### Author Rebuttal · Authors · 2025-07-30
>
> > Since most ML practitioners may not be familiar with partial ancestral graphs (PAGs), it is not clear that practitioners will be able to provide a PAG as input. This may or may not be a weakness of the approach if practitioners need to be educated. Nevertheless, specifying a PAG should require less knowledge than specifying a structural causal model
> >
>
> > suppose that a practitioner wanted to use your approach. Is there a way to explain the approach so that the average ML researcher without much causality background could apply the proposed approach?
> >
>
> A. Thank you for prompting us to provide a brief description of our framework’s workflow. We believe that our approach can be made accessible to practitioners with limited background in causality.
>
> The Fast Causal Inference (FCI) algorithm [Spirtes et al., 2001b, Zhang 2008a] is a well-established causal structure learning method that infers a PAG from purely observational data using conditional independence tests among variables (Practitioners can use the python package *causal-learn* [Zheng et al., 2024] to run FCI and learn a PAG). Therefore, if a practitioner wishes to perform causal decision-making, they can first (1) apply FCI to obtain a PAG, and then (2) use our proposed method to prune the action space accordingly. This abstracted procedure facilitates causal decision-making without requiring full knowledge of the underlying causal diagram.
>
> - Zheng, Yujia, et al., Causal-learn: Causal discovery in python., JMLR (2024)
>
> > What are the MIS and POMIS in each task and how is the count derived for the number of arms to be explored in each approach in the experiments?
> >
>
> A. We appreciate your comment for helping us highlight this important point. Let us consider the PAG shown in Fig. 9. Given the PAG with 10 nodes, an agent can consider interventions on arbitrary combinations of nodes $\mathbf{V} \setminus \lbrace Y \rbrace$.  The number of possible intervention sets is $2^9 = 512$ (i.e., the number of subsets of $\mathbf{V} \setminus \lbrace Y \rbrace$), and the total number of corresponding arms is $\sum_{i=1}^9
> \binom{9}{i}K^i = (K+1)^9 = 19683$ with binary domains $K=2$.
>
> In this example, there are 152 MISs in total (identical to DMIS in this example). We present only the POMIS sets. The collection of POMISs (8 sets) is $\lbrace \lbrace E \rbrace, \lbrace E, F \rbrace, \lbrace E, H \rbrace, \lbrace E, I \rbrace, \lbrace E, F, I \rbrace, \lbrace E, F, H \rbrace, \lbrace E, H, I\rbrace, \lbrace E, F, H, I\rbrace \rbrace$ with a total of $(54 = 2 + 4\ast 3 + 8\ast 3+ 16)$ corresponding arms. Therefore, it suffices to consider only $\frac{54}{19 683} \approx 0.27$% of the total arms. The complete counts of intervention sets and their corresponding arms, for both the main and additional experiments, are provided in Table 2 in Appendix D.
>
> |  | Task1 (Fig. 5a) | Taks2 (Fig. 8) | Taks3 (Fig. 9) | Task4 (Fig. 11a) | Task5 (Fig. 11b) | Task6 (Fig. 11c) |
> | --- | --- | --- | --- | --- | --- | --- |
> | POMIS arms | **19 (23.5%)** | **89 (36.6%)** | **54 (0.27%)** | **15 (6.17%)** | **81 (3.7%)** | **231 (10.7%)** |
> | DMIS arms | 25 | 195 | 2025 | 57 | 189 | 1755 |
> | Total arms | 81 | 243 | 19683 | 243 | 2187 | 2187 |
>
> where the percentages indicate the ratio $\frac{\text{The number of POMIS arms}}{\text{The number of Total arms}} \times 100$, and tasks 4—6 are in Fig. 11 in Appendix D.  All POMISs of our experiments can be found in pages 25-26.

---

> > ### Comment · Reviewer_QaCD · 2025-08-06
> > **response to rebuttal**
> >
> > Thank you for the explanation.  I have no further question.

---

> > > ### Author Response · Authors · 2025-08-06
> > >
> > > We appreciate your prompt once again, which allowed us to provide a brief description of the overall framework’s workflow and to highlight how many arms are removed by our method.

---

### Note · Authors · 2025-08-12

Dear reviewers `QaCD` , `FQq8` , `nSMs` , and `UG9x` .

We sincerely appreciate all reviewers for taking the time to review our paper. Your insightful suggestions and discussions, which may address questions that future readers might have, will improve our work.

We assure you that suggestions made during the rebuttal period will be incorporated into the final version. Specifically,

(1) We will improve a discussion on the assumption of having access to the true PAG from the perspective of causal discovery literature, as this can be challenging when working with real-world data.

(2) We will add a discussion on future work involving adaptive causal discovery from interventions in the online setting. While balancing exploration for causal discovery and for regret minimization poses substantial additional challenges—as these two objectives, structure discovery and regret minimization, are not naturally aligned—we recognize that this is an important direction for the literature to pursue.

(3) We will include detailed comparison with [Elahi et al., 2024] in terms of both the overall framework and regret bounds, to clarify the distinction between the two approaches: [Elahi et al., 2024] adopts a fully online strategy that discovers the structure from interventions and then interacts, whereas our approach relies on offline discovery PAG (e.g., FCI) followed by interaction.

We once again thank the reviewers for their time and valuable feedback.

Best regards,

Authors 26106

---

### Decision · Program_Chairs · 2025-09-17

**Decision:**

Accept (poster)

**Comment:**

The paper presents a study on stochastic causal bandit with latent variables, and it starts with a Markov equivalent class. Addressing causal bandits with latent variables and unknown causal graph structure is an important topic, and the reviewers acknowledge the importance and appreciate that the authors provide a solid algorithmic contribution with theoretical analysis. After the rebuttal and discussion period, one major remaining concern is that the work requires a preprocessing period to obtain the Markov equivalent class, while some other studies do not need this preprocessing period and is purely online processing, e.g. [Elahi et al., 2024] .

After reading the review comments and discussion, I feel that the technical contribution of the paper overall is solid, and outweigh this concern of preprocessing. In some scenarios, preprocessing data to obtain Markov equivalent class may be necessary anyway and may already have been done, and thus some method utilizing the Markov equivalent class could be reasonable and worth exploring.

Therefore, I would like to recommend the paper for acceptance, but indeed the paper is more or less at the boundary of NeurIPS acceptance bar. I hope that the authors could provide a thorough revision addressing all concerns from the reviewers to improve the quality of the paper, and provide a detailed discussion on the reliance of Markov equivalent class.